



# Automating Finite Element Methods for Geodynamics via Firedrake

D. Rhodri Davies[1], Stephan C. Kramer[2], Sia Ghelichkhan[1], and Angus Gibson[1]

[1]Research School of Earth Sciences, The Australian National University, Canberra, ACT, Australia.
[2]Department of Earth Science and Engineering, Imperial College London, London, UK.

**Correspondence:** Rhodri Davies (Rhodri.Davies@anu.edu.au)

**Abstract.** Firedrake is an automated system for solving partial differential equations using the finite element method. By applying sophisticated performance optimisations through automatic code-generation techniques, it provides a means to create accurate, efficient, flexible, easily extensible, scalable, transparent and reproducible research software, that is ideally suited to simulating a wide-range of problems in geophysical fluid dynamics. Here, we demonstrate the applicability of Firedrake for geodynamical simulation, with a focus on mantle dynamics. The accuracy and efficiency of the approach is confirmed via comparisons against a suite of analytical and benchmark cases of systematically increasing complexity, whilst parallel scalability is demonstrated up to 12288 compute cores, where the problem size and the number of processing cores are simultaneously increased. In addition, Firedrake's flexibility is highlighted via straightforward application to different physical (e.g. complex nonlinear rheologies, compressibility) and geometrical (2-D and 3-D Cartesian and spherical domains) scenarios. Finally, a representative simulation of global mantle convection is examined, which incorporates 230 Myr of plate motion history as a kinematic surface boundary condition, confirming Firedrake's suitability for addressing research problems at the frontiers of global mantle dynamics research.

## 1 Introduction

Since the advent of plate tectonic theory, there has been a long and successful history of research software development within the geodynamics community. The earliest modelling tools provided fundamental new insight into the process of mantle convection, its sensitivity to variations in viscosity, and its role in controlling Earth's surface plate motions and heat transport (e.g. McKenzie, 1969; Minear and Toksoz, 1970; Torrance and Turcotte, 1971; McKenzie et al., 1973). Although transformative at the time, computational and algorithmic limitations dictated that these tools were restricted to a simplified approximation of the underlying physics and, excluding some notable exceptions (e.g. Baumgardner, 1985; Glatzmaier, 1988), to 2-D Cartesian geometries. They were specifically designed to address targeted scientific questions. As such, they offered limited flexibility, were not easily extensible, and were not portable across different platforms. Furthermore, since they were often developed for use by one or two expert practitioners, they were poorly documented: details of the implementation could only be determined by analysing the underlying code, which was often a non-trivial and specialised task.

Growing computational resources and significant theoretical and algorithmic advances have since underpinned the development of more advanced research software, which incorporate, for example, better approximations to the fundamental physical principles, including compressibility (e.g. Jarvis and McKenzie, 1980; Bercovici et al., 1992; Tackley, 1996; Bunge et al., 1997;





Gassmoller et al., 2020), mineralogical phase transformations (e.g. Tackley et al., 1993; Nakagawa et al., 2009), multi-phase flow (e.g. Katz and Weatherley, 2012; Wilson et al., 2014), variable and nonlinear rheologies (e.g. Moresi and Solomatov, 1995; Bunge et al., 1996; Trompert and Hansen, 1998; Tackley, 2000; Moresi et al., 2002; Stadler et al., 2010; Alisic et al.,

2011; Le Voci et al., 2014; Garel et al., 2014), and feedbacks between chemical heterogeneity and buoyancy (e.g. van Keken, 1997; Tackley and Xie, 2002; Davies et al., 2012). In addition, these more recent tools can often be applied in more representative 2-D cylindrical and/or 3-D spherical geometries (e.g. Baumgardner, 1985; Bercovici et al., 1989; Jarvis, 1993; Bunge et al., 1997; van Keken and Ballentine, 1998; Zhong et al., 2000, 2008; Tackley, 2008; Wolstencroft et al., 2009; Stadler et al., 2010; Davies et al., 2013). The user-base of these tools has rapidly increased, with software development teams emerging

to enhance their applicability and ensure their ongoing functionality. These teams have done so by adopting best-practices in modern software development, including version control, unit and regression testing across a range of platforms, and validation of model predictions against a suite of analytical and benchmark solutions (e.g. Blankenbach et al., 1989; Busse et al., 1994; King et al., 2009; Tosi et al., 2015; Kramer et al., 2021a).

Nonetheless, given rapid and ongoing improvements in algorithmic design and software engineering, alongside the develop-

ment of robust and flexible scientific computing libraries that provide access to much of the low-level numerical functionality required by geodynamical models, a next-generation of open-source and community driven geodynamical research software has emerged, exploiting developments from the forefront of computational engineering. This includes ASPECT (e.g. Kronbichler et al., 2012; Heister et al., 2017; Bangerth et al., 2020), built on the deal.II (Bangerth et al., 2007), p4est (Burstedde et al., 2011) and Trilinos (Heroux et al., 2005; Trilinos Project Team) libraries, Fluidity (e.g. Davies et al., 2011; Kramer et al.,

2012, 2021a, b), which is underpinned by the PETSc (Balay et al., 1997, 2021a, b) and Spud libraries (Ham et al., 2009), Underworld2 (e.g. Moresi et al., 2007; Beucher et al., 2019), core aspects of which are built on the St Germain (Quenette et al., 2007) and PETSc libraries, and TerraFERMA (Wilson et al., 2017), which has foundations in the FEniCS (Logg et al., 2012; Alnes et al., 2014), PETSc and Spud libraries. By building on existing computational libraries that are highly-efficient, extensively tested and validated, modern geodynamical research software is becoming increasingly reliable and reproducible. Its

modular design also facilitates the addition of new features and provides a degree of confidence about the validity of previous developments, as evidenced by growth in the use and applicability of ASPECT over recent years.

However, even with these modern research software frameworks, some fundamental development decisions, such as the core physical equations, numerical approximations and general solution strategy, have been integrated into the basic building blocks of the code. Whilst there remains some flexibility within the context of a single problem, modifications to include

different physical approximations or components, which can affect nonlinear coupling and associated solution strategies, often require extensive and time-consuming development and testing, using either separate code forks or increasingly complex options systems. This makes reproducibility of a given simulation difficult, resulting in a lack of transparency – even with detailed documentation, specific details of the implementation are sometimes only available by reading the code itself, which, as noted previously, is non-trivial, particularly across different forks or with increasing code complexity (Wilson et al., 2017).

This makes scientific studies into the influence of different physical or geometrical scenarios, using a consistent code-base, extremely challenging. Those software frameworks that try to maintain some degree of flexibility often do so at the compromise



of performance: the flexibility to configure different equations, numerical discretisations and solver strategies, in different dimensions and geometries, requires implementation compromises in the choice of optimal algorithms and specific low-level optimisations for all possible configurations.

A challenge that remains central to research software development in geodynamics, therefore, is the need to provide accurate, efficient, flexible, easily extensible, scalable, transparent and reproducible research software that can be applied to simulating a wide-range of scenarios, including problems in different geometries and those incorporating different approximations of the underlying physics (e.g. Wilson et al., 2017). However, this requires a large time commitment and knowledge that spans several academic disciplines. Arriving at a physical description of a complex system, such as global mantle convection, demands

expertise in geology, geophysics, geochemistry, fluid mechanics and rheology. Discretising the governing partial differential equations (PDEs) to produce a suitable numerical scheme, requires proficiency in mathematical analysis, whilst its translation into efficient code for massively parallel systems demands advanced knowledge in low-level code optimisation and computer architectures (e.g. Rathgeber et al., 2016). The consequence of this is that the development of research software for geodynamics has now become a multi-disciplinary effort and its design must enable scientists across several disciplines to collaborate

effectively, without requiring each of them to comprehend all aspects of the system.

Key to achieving this is to abstract, automate, and compose the various processes involved in numerically solving the PDEs governing a specific problem (e.g. Logg et al., 2012; Alnes et al., 2014; Rathgeber et al., 2016; Wilson et al., 2017), to enable a separation of concerns between developing a technique and employing it. As such, software projects involving automatic code generation have become increasingly popular, as these help to separate different aspects of development. Such an approach

facilitates collaboration between computational engineers with expertise in hardware and software, computer scientists and applied mathematicians with expertise in numerical algorithms, and domain specific scientists, such as geodynamicists.

In this study, we introduce Firedrake to the geodynamical modelling community: a next-generation automated system for solving PDEs using the finite element method (e.g. Rathgeber et al., 2016; Gibson et al., 2019). As we will show, the finite element method is well-suited to automatic code-generation techniques: a weak formulation of the governing PDEs, together

with a mesh, initial and boundary conditions, and appropriate discrete function spaces, is sufficient to fully represent the problem. The purpose of this manuscript is to demonstrate the applicability of Firedrake for geodynamical simulation, whilst also highlighting its advantages over existing geodynamical research software. We do so via comparisons against a suite of analytical and benchmark cases of systematically increasing complexity.

The remainder of the manuscript is structured as follows. In Section 2, we provide a background to the Firedrake project

and the various dependencies of its software stack. In Section 3 we introduce the equations governing mantle convection which will be central to the examples developed herein, followed, in Section 4, by a description of their discretisation via the finite element method and the associated solution strategies. In Section 5 we introduce a series of benchmark cases in Cartesian and spherical geometries. These are commonly examined within the geodynamical modelling community, and we describe the steps involved with setting up these cases in Firedrake. Parallel performance is analysed in Section 6, with a representative

example of global mantle convection described and analysed in Section 7. The latter case confirms Firedrake's suitability for



addressing research problems at the frontiers of global mantle dynamics research. Other components of Firedrake, which have not been showcased in this manuscript but may be beneficial to various future research endeavours, are discussed in Section 8.

## 2 Firedrake

The Firedrake project is an automated system for solving partial differential equations using the finite element method (e.g.
Rathgeber et al., 2016). Using a high-level language that reflects the mathematical description of the governing equations (e.g. Alnes et al., 2014), the user specifies the finite element problem symbolically. The high-performance implementation of assembly operations for the discrete operators is then generated 'automatically' by a sequence of specialised compiler passes that apply symbolic mathematical transformations to the input equations to ultimately produce C (and C++) code (Rathgeber et al., 2016; Homolya et al., 2018). Firedrake compiles and executes this code to create linear or nonlinear systems, which are
solved by PETSc (Balay et al., 1997, 2021b, a). As stated by Rathgeber et al. (2016), in comparison with conventional finite element libraries, and even more so with handwritten code, Firedrake provides a higher productivity mechanism for solving finite element problems whilst simultaneously applying sophisticated performance optimisations that few users would have the resources to code by hand.

Firedrake builds on the concepts and some of the code of the FEniCS project (e.g. Logg et al., 2012), particularly its
representation of variational problems via the Unified Form Language (UFL) (Alnes et al., 2014). We note that the applicability of FEniCS for geodynamical problems has already been demonstrated (e.g. Vynnytska et al., 2013; Wilson et al., 2017). Both frameworks have the goal of saving users from manually writing low-level code for assembling the systems of equations that discretise their model physics. An important architectural difference is that while FEniCS has components written in C++ and Python, Firedrake is completely written in Python, including its run-time environment (it is only the automatically generated
assembly code that is in C/C++, although it does leverage the PETSc library, written in C, to solve the assembled systems, albeit through its Python interface). This provides a highly flexible user interface with ease of introspection of data structures. We note that the Python environment also allows deploying of hand written C kernels should the need arise to perform discrete mesh-based operations that cannot be expressed in the finite element framework, such as sophisticated slope limiters or bespoke sub-grid physics.

Firedrake offers several highly-desirable features rendering it well-suited to problems in geophysical fluid dynamics. As will be illustrated through a series of examples below, of particular importance in the context of this manuscript are Firedrake's support for a range of different finite-element discretisations, including a highly efficient implementation of those based on extruded meshes, programmable nonlinear solvers and composable operator aware solver preconditioners. As the importance of reproducibility in the computational geosciences is increasingly recognized, we note that Firedrake integrates with Zenodo and
GitHub to provide users with the ability to generate a set of DOIs corresponding to the exact set of Firedrake components used to conduct a particular simulation, in full compliance with FAIR (Findable, Accessible, Interoperable, Reusable) principles.





## 2.1 Dependencies

Firedrake treats finite element problems as a composition of several abstract processes, using separate packages for each. The framework imposes a clear separation of concerns between the definition of the problem (UFL, Firedrake Language), the
generation of computational kernels used to assemble the coefficients of the discrete equations (TSFC, FInAT), the parallel execution of this kernel (PyOP2) over a given mesh topology (DMPlex), and the solution of the resulting linear or nonlinear systems (PETSc). These layers allow various types of optimisation to be applied at different stages of the solution process. The key components of this software stack are next described.

1. Unified Form Language (UFL) – as we will see in the examples below, a core part of finite element problems is the
specification of the weak form of the governing PDEs. UFL, a domain-specific symbolic language with well-defined and mathematically consistent semantics that is embedded in Python, provides an elegant solution to this problem. It was pioneered by the FEniCS project (Logg et al., 2012), although Firedrake has added several extensions.

2. Firedrake Language – in addition to the weak form of the PDEs, finite element problems require the user to select appropriate finite elements, specify the mesh to be employed, set field values for initial and boundary conditions and
specify the sequence in which solves occur. Firedrake implements its own language for these tasks, which was designed to be to a large extent compatible with DOLFIN (Logg et al., 2012), the runtime API of the FEniCS project. We note that Firedrake implements various extensions to DOLFIN, whilst some features of DOLFIN are not supported by Firedrake.

3. FInAT (Kirby and Mitchell, 2019) – incorporates all information required to evaluate the basis functions of the different finite element families supported by Firedrake. In earlier versions of Firedrake this was done through tabulation of the
basis functions evaluated at Gauss points (FIAT: Kirby, 2004). FInAT, however, provides this information to the form compiler as a combination of symbolic expressions and numerical values, allowing for further optimisations. FInAT allows Firedrake to support a wide-range of finite elements, including continuous, discontinuous, H(div) and H(curl) discretisations, and elements with continuous derivatives such as the Argyris and Bell elements.

4. Two-Stage Form Compiler (TSFC) – a form compiler takes a high-level description of the weak form of PDEs (here
in UFL) and produces low-level code that carries out the finite element assembly. Firedrake uses TSFC, which was developed specifically for the Firedrake project (Homolya et al., 2018), to generate its local assembly kernels. TSFC invokes two stages, where in the first stage UFL is translated to an intermediate symbolic tensor algebra language, before translating this into assembly kernels written in C. In comparison with the form compilers of FEniCS (FFC and UFLACS), TSFC aims to maintain the algebraic structure of the input expression for longer, which opens up additional
opportunities for optimisation.

5. PyOP2 – a key component of Firedrake's software stack is PyOP2, a high-level framework that optimises the parallel execution of computational kernels on unstructured meshes (Rathgeber et al., 2012; Markall et al., 2013). Where the local assembly kernels generated by TSFC calculate the values of a local tensor from local input tensors, all associated with





the degrees of freedom of a single element, PyOP2 wraps this code in an additional layer responsible for the extraction
and addition of these local tensors out of/into global structures such as vectors and sparse matrices. It is also responsible
for the maintenance of halo layers, the overlapping regions in a parallel decomposed problem. PyOP2 allows for a clean
separation of concerns between the specification of the local kernel functions, in which the numerics of the method are
encoded, and their efficient parallel execution. More generally, this separation of concerns is the key novel abstraction
that underlies the design of the Firedrake system.

6. DMPlex – PyOP2 has no concept of the topological construction of a mesh. Firedrake derives the required maps through
DMPlex, a data management abstraction that represents unstructured mesh data, which is part of the PETSc project
(Knepley and Karpeev, 2009). This allows Firedrake to leverage the DMPlex partitioning and data migration interfaces to
perform domain decomposition at run-time, whilst supporting multiple mesh file formats. Moreover, Firedrake reorders
mesh entities to ensure computational efficiency (Lange et al., 2016).

7. Linear and nonlinear solvers – Firedrake passes solver problems on to PETSc (Balay et al., 1997, 2021a, b), a well-
established, high-performance solver library that provides access to several of its own and third-party implementations of
solver algorithms. The Python interface to PETSc (Dalcin et al., 2011) makes integration with Firedrake straightforward.
We note that employing PETSc for both its solver library and for DMPlex has the additional advantage that the set of
library dependencies required by Firedrake is kept small (Rathgeber et al., 2016).

## 3  Governing Equations

Our focus here is on mantle convection, the slow creeping motion of Earth's mantle over geological timescales. The equations
governing mantle convection are derived from the conservation laws of mass, momentum and energy. The simplest mathe-
matical formulation assumes incompressibility and the Boussinesq approximation (McKenzie et al., 1973), under which the
non–dimensional momentum and continuity equations are given by:

$$\nabla \cdot \bar{\bar{\sigma}} + Ra_0 T \hat{\boldsymbol{k}} = 0, \tag{1}$$

$$\nabla \cdot \boldsymbol{u} = 0, \tag{2}$$

where $\bar{\bar{\sigma}}$ is the stress tensor, $\boldsymbol{u}$ is the velocity and $T$ temperature. $\hat{\boldsymbol{k}}$ is the unit vector in the direction opposite to gravity and
$Ra_0$ denotes the Rayleigh number, a dimensionless number that quantifies the vigor of convection:

$$Ra_0 = \frac{\rho_0 \alpha \Delta T g d^3}{\mu_0 \kappa}. \tag{3}$$

Here, $\rho_0$ denotes reference density, $\alpha$ the thermal expansion coefficient, $\Delta T$ the characteristic temperature change across the
domain, $g$ the gravitational acceleration, $d$ the characteristic length, $\mu_0$ the reference dynamic viscosity and $\kappa$ the thermal
diffusivity. Note that the above non–dimensional equations are obtained through the following characteristic scales: length $d$;
time $d^2 / \kappa$; and temperature $\Delta T$.



When simulating incompressible flow, the full stress tensor, $\bar{\bar{\sigma}}$, is decomposed into deviatoric and volumetric components:

$$\bar{\bar{\sigma}} = \bar{\bar{\tau}} - pI, \tag{4}$$

where $\bar{\bar{\tau}}$ is the deviatoric stress tensor, $p$ is dynamic pressure and $I$ is the identity matrix. Substituting Eq. (4) into Eq. (1) and utilizing the constituative relation

$$\bar{\bar{\tau}} = 2\mu\dot{\epsilon} = 2\mu\,\mathrm{sym}(\nabla\boldsymbol{u}) = \mu\left[\nabla\boldsymbol{u} + (\nabla\boldsymbol{u})^T\right], \tag{5}$$

which relates the deviatoric stress tensor, $\bar{\bar{\tau}}$, to the strain-rate tensor, $\dot{\epsilon} = \mathrm{sym}(\nabla\boldsymbol{u})$, yields:

$$\nabla \cdot \mu\left[\nabla\boldsymbol{u} + (\nabla\boldsymbol{u})^T\right] - \nabla p + Ra_0 T\hat{\boldsymbol{k}} = 0. \tag{6}$$

The viscous flow problem can thus be posed in terms of pressure, $p$, velocity, $\boldsymbol{u}$, and temperature, $T$. The evolution of the thermal field is controlled by an advection–diffusion equation:

$$\frac{\partial T}{\partial t} + \boldsymbol{u} \cdot \nabla T - \nabla \cdot (\kappa\nabla T) = 0 \tag{7}$$

These governing equations are sufficient to solve for the three unknowns, together with adequate boundary and initial conditions.

## 4 Finite Element Discretisation and Solution Strategy

For the derivation of the finite element discretisation of Equations (6), (2), and (7) we start by writing these in their weak form. We select appropriate function spaces V, W, and Q that contain, respectively, the solution fields for velocity $\boldsymbol{u}$, pressure $p$, and temperature $T$, and also contain the test functions $\boldsymbol{v}, w$ and $q$. The weak form is then obtained by multiplying these equations with the test functions and integrating over the domain $\Omega$,

$$\int_\Omega (\nabla\boldsymbol{v}) : \mu\left[\nabla\mathbf{u} + (\nabla\mathbf{u})^T\right] \, \mathrm{d}\boldsymbol{x} + \int_\Omega \boldsymbol{v} \cdot \nabla p \, \mathrm{d}\boldsymbol{x} - \int_\Omega Ra_0 T\boldsymbol{v} \cdot \hat{\boldsymbol{k}} \, \mathrm{d}\boldsymbol{x} = 0 \text{ for all } \boldsymbol{v} \in V, \tag{8}$$

$$-\int_\Omega (\nabla w) \cdot \mathbf{u} \, \mathrm{d}\boldsymbol{x} = 0 \text{ for all } w \in W, \tag{9}$$

$$\int_\Omega q\frac{\partial T}{\partial t} \, \mathrm{d}\boldsymbol{x} + \int_\Omega q\boldsymbol{u} \cdot \nabla T \, \mathrm{d}\boldsymbol{x} + \int_\Omega (\nabla q) \cdot (\kappa\nabla T) \, \mathrm{d}\boldsymbol{x} = 0 \text{ for all } q \in Q. \tag{10}$$

Note that we have integrated by parts the viscosity term in (6), the divergence term in (2), and the diffusion term in (7), but have omitted the corresponding boundary terms, which will be considered in the following section.

Equations (8-10) are a more general, mathematically rigorous representation of the continuous PDEs in strong form (Equations 6, 2 and 7), provided suitable function spaces with sufficient regularity are chosen (see, for example Zienkiewicz and Taylor, 1991; Elman et al., 2005). Galerkin finite element discretisation proceeds by restricting these function spaces to



finite-dimensional subspaces. These are typically constructed by dividing the domain into cells or elements, and restricting

to piecewise polynomial subspaces with various continuity requirements between cells. In all examples presented in this paper, we use Continuous Galerkin (CG) finite elements, specifically the Q2-Q1 element pair for velocity and pressure and the Q2 discretisation for temperature.

We note that there are many other choices of finite element function spaces available in Firedrake, although they are not considered herein (see Gibson et al., 2019, for an overview). All that is required for their implementation is that a basis can be

found for the function space such that each solution can be written as a linear combination of basis functions. For example, if we have a basis $\phi_i$ of the finite dimensional function space $Q_h$ of temperature solutions, then we can write each temperature solution as

$$T(\boldsymbol{x}) = \sum_i T_i \phi_i(\boldsymbol{x}) \tag{11}$$

where $T_i$ represents the coefficients that we can collect into a discrete solution vector $\underline{T}$. Using a Lagrangian polynomial basis

the coefficients $T_i$ correspond to values at the nodes, where each node $i$ is associated with one basis function $\phi_i$, but this is not generally true for other choices of finite element bases.

In curved domains, boundaries can only be approximated with a finite number of triangles, tetrahedrals, quadrilaterals or hexahedrals. In a sense, this can be seen as a piecewise linear (or bi/tri-linear) approximation where the domain is approximated by straight lines (edges) between vertices. A more accurate representation of the domain is obtained by allowing higher order

polynomials that describe the physical embedding of the element within the domain. A typical choice is to use a so-called isoparametric representation in which the polynomial order of the embedding is the same as that of the discretised functions that are solved for.

Finally, we note that it is common to use a subscript $_h$ for the discrete, finite-dimensional function subspaces and $\Omega_h$ for the discretised approximation by the mesh of the domain $\Omega$, but since the remainder of this manuscript focusses on the details and

implementation of this discretisation, we simply drop the $_h$ subscripts from here on.

### 4.1 Boundary conditions

In the Cartesian examples considered below, zero-slip and free-slip boundary conditions for (8) and (9) are imposed through strong Dirichlet boundary conditions for velocity $\boldsymbol{u}$. This is achieved by restricting the velocity function space $V$ to a subspace $V_0$ of vector functions for which all components (zero-slip) or only the normal component (free-slip) are zero at the boundary.

Since this restriction also applies to the test functions $\boldsymbol{v}$, the weak form only needs to be satisfied for all test functions $\boldsymbol{v} \in V_0$ that satisfy the homogeneous boundary conditions. Therefore, the omitted boundary integral

$$-\int_{\partial\Omega} \boldsymbol{v} \cdot \left( \mu \left[ \nabla \boldsymbol{u} + (\nabla \boldsymbol{u})^T \right] \right) \cdot \boldsymbol{n} \, \mathrm{d}\boldsymbol{s} \tag{12}$$

that was required to obtain the integrated by parts viscosity term in Equation (8), automatically vanishes for zero-slip boundary conditions as $\mathbf{v} = \mathbf{0}$ at the domain boundary, $\partial\Omega$. In the case of a free-slip boundary condition for which the tangential com-

ponents of $\boldsymbol{v}$ are non-zero, the boundary term does not vanish, but by omitting that term in (8) we weakly impose a zero shear



stress condition. The boundary term obtained by integrating the divergence term in (2) by parts,

$$\int_{\partial\Omega} w\boldsymbol{n} \cdot \boldsymbol{u} \, \mathrm{d}\boldsymbol{s}, \tag{13}$$

vanishes for both zero-slip and free-slip boundary conditions because of the no-outflow boundary condition.

Similarly, in the examples presented below, we impose strong Dirichlet boundary conditions for temperature at the top and
bottom boundaries of our domain. The test functions are restricted to $Q_0$ which consists of temperature functions that satisfy homogeneous boundary conditions at these boundaries, and thus

$$\int_{\partial\Omega} q\boldsymbol{n} \cdot \kappa\nabla T \, \mathrm{d}\boldsymbol{s}, \tag{14}$$

the boundary term associated with integrating by parts of the diffusion term, vanishes. In Cartesian domains the boundary term does not vanish for the lateral boundaries, but by omitting this term from (10) we weakly impose a homogeneous Neumann
(zero-flux) boundary condition at these boundaries. The temperature solution itself is found in $Q_0 + \{T_{\mathrm{inhom}}\}$ where $T_{\mathrm{inhom}}$ is any representative temperature function that satisfies the required inhomogenous boundary conditions.

In curved domains, such as the 2-D cylindrical and 3-D spherical cases examined below, imposing free-slip boundary conditions is complicated by the fact that it is not straightforward to decompose the degrees of freedom of the velocity space $V$ into tangential and lateral components for many finite element discretisations. For Lagrangian based discretisations we could
define normal vectors at the Lagrangian nodes on the surface and decompose accordingly, but these normal vectors would have to be averaged due to the piecewise approximation of the curved surface. To avoid such complications for our examples in cylindrical and spherical geometries, we employ a symmetric Nitsche penalty method (Nitsche, 1971) where the velocity space is not restricted and, thus, retains all discrete solutions with a non-zero normal component. This entails adding the following three surface integrals to Equation (8):

$$-\int_{\partial\Omega} \boldsymbol{v} \cdot \boldsymbol{n}\, \boldsymbol{n} \cdot \left(\mu\left[\nabla\boldsymbol{u} + (\nabla\boldsymbol{u})^T\right]\right) \cdot \boldsymbol{n} \, \mathrm{d}\boldsymbol{s} - \int_{\partial\Omega} \boldsymbol{n} \cdot \left(\mu\left[\nabla\boldsymbol{v} + (\nabla\boldsymbol{v})^T\right]\right) \cdot \boldsymbol{n}\, \boldsymbol{u} \cdot \boldsymbol{n} \, \mathrm{d}\boldsymbol{s} + \int_{\partial\Omega} C_{\mathrm{Nitsche}}\, \mu\boldsymbol{v} \cdot \boldsymbol{n}\, \boldsymbol{u} \cdot \boldsymbol{n} \, \mathrm{d}\boldsymbol{s} \quad . \tag{15}$$

The first of these corresponds to the normal component of Equation (12) associated with integration by parts of the viscosity term. The tangential component, as before, is omitted and weakly imposes a zero shear stress condition. The second term ensures symmetry of Equation (8) with respect to $\boldsymbol{u}$ and $\boldsymbol{v}$. The third term penalizes the normal component of $\boldsymbol{u}$ and involves a penalty parameter $C_{\mathrm{Nitsche}} > 0$ that should be sufficiently large to ensure coercivity of $F_{\mathrm{Stokes}}$ as a bilinear form in $\boldsymbol{u}$ and
$\boldsymbol{v}$. Lower bounds for $C_{\mathrm{Nitsche},f}$ on each face $f$ can be derived for simplicial (Shahbazi, 2005) and quadrilateral/hexahedral (Hillewaert, 2013) meshes, respectively:

Triangular ($d = 2$) / Tetrahedral ($d = 3$) meshes:
$$C_{\mathrm{Nitsche},f} > C_{\mathrm{ip}}\frac{p(p+d-1)}{d}\frac{A_f}{V_{c_f}}, \tag{16}$$

Quadrilateral/Hexahedral meshes:
$$C_{\mathrm{Nitsche},f} > C_{\mathrm{ip}}(p+1)^2\frac{A_f}{V_{c_f}}, \tag{17}$$

where $A_f$ is the facet area of face $f$, $V_{c_f}$ the cell volume of the adjacent cell $c_f$, and $p$ is the polynomial degree of the velocity
discretisation. Here, we introduce an additional factor, $C_{\mathrm{ip}}$, to account for spatial variance of the viscosity $\mu$ in the adjacent cell,





and domain curvature, which are not taken into account in the standard lower bounds (using $C_{\text{ip}} = 1$). In all free-slip cylindrical and spherical examples presented below, we use $C_{\text{ip}} = 100$.

## 4.2  Temporal discretisation and solution process for temperature

For temporal integration, we apply a simple $\theta$ scheme to the energy equation (10):

$$F_{\text{energy}}(q; T^{n+1}) := \int_\Omega q \frac{T^{n+1} - T^n}{\Delta t} \, \mathrm{d}\boldsymbol{x} + \int_\Omega q\boldsymbol{u} \cdot \nabla T^{n+\theta} \, \mathrm{d}\boldsymbol{x} + \int_\Omega (\nabla q) \cdot \left( \kappa \nabla T^{n+\theta} \right) \, \mathrm{d}\boldsymbol{x} = 0 \text{ for all } q \in Q, \tag{18}$$

where

$$T^{n+\theta} = \theta T^{n+1} + (1 - \theta) T^n \tag{19}$$

is interpolated between the temperature solutions $T^n$ and $T^{n+1}$ at the beginning and end of the $n + 1$-th time step using a parameter $0 \leq \theta \leq 1$. In all examples that follow, we use a Crank-Nicholson scheme, where $\theta = 0.5$. To simplify we will solve for velocity and pressure, $\boldsymbol{u}$ and $p$, in a separate step before solving for the new temperature $T^{n+1}$.

Because $F_{\text{energy}}$ is linear in $q$, if we expand the test function $q$ as a linear combination of basis functions $\phi_i$ of **Q**

$$F_{\text{energy}}(q; T^{n+1}) = F_{\text{energy}}(\sum_i q_i \phi_i; T^{n+1}) = \sum_i q_i F_{\text{energy}}(\phi_i; T^{n+1}) =: \sum_i q_i \underline{\text{F}}(T^{n+1})_i, \tag{20}$$

where $\underline{\text{F}}(T^{n+1})$ is the vector with coefficients $F_{\text{energy}}(\phi_i; T^{n+1})$ (i.e. the energy equation tested with the basis functions $\phi_i$). Thus, to satisfy Equation (18) we need to solve for a temperature $T$ for which the entire vector $\underline{\text{F}}(T^{n+1})$ is zero.

In the general, nonlinear case, this can be solved using a Newton solver, but here the system of equations $\underline{\text{F}}(T^{n+1})$ is also linear in $T^{n+1}$ and, accordingly, if we also expand the temperature with respect to the same basis: $T^{n+1} = \sum_j T_j^{n+1} \phi_j$ where we store the coefficients $T_j^{n+1}$ in a vector $\underline{\text{T}}$, we can write it in the usual form as a linear system of equations

$$A\underline{\text{T}} = \underline{\text{b}}, \tag{21}$$

with $A$ the matrix that represents the Jacobian $\frac{\partial F}{\partial T}$ with respect to the basis $\phi_i$, and the right-hand side vector $\underline{\text{b}}$ containing all terms in (18) that do not depend on $T^{n+1}$, specifically:

$$A_{ij} = \frac{\partial F_{\text{energy}}(\phi_i; T^{n+1})}{\partial T_j^{n+1}} = \int_\Omega \phi_i \frac{\phi_j}{\Delta t} \, \mathrm{d}\boldsymbol{x} + \int_\Omega \phi_i \boldsymbol{u} \cdot \theta \nabla \phi_j \, \mathrm{d}\boldsymbol{x} + \int_\Omega (\nabla \phi_i) \cdot (\kappa \theta \nabla \phi_j) \, \mathrm{d}\boldsymbol{x} = 0 \tag{22}$$

$$\underline{\text{b}}_j = F_{\text{energy}}(\phi_i; 0) = \int_\Omega \phi_i \frac{T^n}{\Delta t} \, \mathrm{d}\boldsymbol{x} - \int_\Omega \phi_i \boldsymbol{u} \cdot (1 - \theta) \nabla T^n \, \mathrm{d}\boldsymbol{x} - \int_\Omega (\nabla \phi_i) \cdot (\kappa (1 - \theta) \nabla T^n) \, \mathrm{d}\boldsymbol{x} = 0 \tag{23}$$

In the nonlinear case, every Newton iteration requires the solution of such a linear system with a Jacobian matrix $A_{ij} = \partial F_{\text{energy}} / \partial T_j^{n+1}$ and right-hand side vector based on the residual $\underline{\text{b}}_i = F_{\text{energy}}(\phi_i, T^{n+1})$ that both need to be reassembled every iteration as $T^{n+1}$ is iteratively improved. For the 2-D cases presented in this paper, this asymmetric linear system is solved with a direct solver, and in 3-D using a combination of the GMRES Krylov subspace method with a symmetric SOR (SSOR) preconditioner.





### 4.3 Solving for velocity and pressure

In a separate step, we solve Equations (8) and (9) for velocity and pressure. Since these weak equations need to hold for all test
functions $\boldsymbol{v} \in V$ and $w \in W$ we can equivalently write, using a single residual functional $F_{\text{Stokes}}$:

$$F_{\text{Stokes}}(\boldsymbol{v}, w; \boldsymbol{u}, p) = \int_{\Omega} (\nabla \boldsymbol{v}) : \mu \left[ \nabla \boldsymbol{u} + (\nabla \boldsymbol{u})^T \right] \, \mathrm{d}\boldsymbol{x} + \int_{\Omega} \boldsymbol{v} \cdot \nabla p \, \mathrm{d}\boldsymbol{x}$$

$$- \int_{\Omega} Ra_0 T \boldsymbol{v} \cdot \hat{\boldsymbol{k}} \, \mathrm{d}\boldsymbol{x} + \int_{\Omega} (\nabla w) \cdot \boldsymbol{u} \, \mathrm{d}\boldsymbol{x} = 0 \text{ for all } \boldsymbol{v} \in V, w \in W, \quad (24)$$

where we have multiplied the continuity equation with $-1$ to ensure symmetry between the $\nabla p$ and $\nabla \cdot u$ terms. This combined
weak form that we simultaneously solve for a velocity $u \in V$ and pressure $p \in W$ is referred to as a *mixed problem*, and the
combined solution $(u, p)$ is said to be found in the *mixed function space* $V \oplus W$.

As before, we expand the discrete solutions $\boldsymbol{u}$ and $p$, and test functions $\boldsymbol{v}$ and $w$ in terms of basis functions for $V$ and $W$

$$\boldsymbol{u} = \sum_i u_i \boldsymbol{\psi}_i, \qquad\qquad \boldsymbol{v} = \sum_i v_i \boldsymbol{\psi}_i, \qquad\qquad \text{span}\{\boldsymbol{\psi}_i\} = V \qquad (25)$$

$$p = \sum_k p_k \chi_k, \qquad\qquad w = \sum_k w_k \chi_k, \qquad\qquad \text{span}\{\chi_k\} = W \qquad (26)$$

For isoviscous cases, where $F_{\text{Stokes}}$ is linear in $\boldsymbol{u}$ and $p$, we then derive a linear system of the following form

$$\begin{pmatrix} K & G \\ G^T & 0 \end{pmatrix} \begin{pmatrix} \underline{u} \\ \underline{p} \end{pmatrix} = \begin{pmatrix} \underline{f} \\ \underline{0} \end{pmatrix} \qquad (27)$$

where

$$K_{ij} = \frac{\partial F_{\text{Stokes}}(\boldsymbol{\psi}_i, 0; \boldsymbol{u}, p)}{\partial \boldsymbol{u}_j} = \int_{\Omega} (\nabla \boldsymbol{\psi}_i) : \mu \left[ \nabla \boldsymbol{\psi}_j + (\nabla \boldsymbol{\psi}_j)^T \right] \, \mathrm{d}\boldsymbol{x} \qquad (28)$$

$$G_{ik} = \frac{\partial F_{\text{Stokes}}(\boldsymbol{\psi}_i, 0; \boldsymbol{u}, p)}{\partial p_k} = \int_{\Omega} \boldsymbol{\psi}_i \cdot \nabla \chi_k \, \mathrm{d}\boldsymbol{x} = -\frac{\partial F_{\text{Stokes}}(0, \chi_k; \boldsymbol{u}, p)}{\partial \boldsymbol{u}_i} \qquad (29)$$

$$\underline{f}_i = Ra_0 \int_{\Omega} T \boldsymbol{\psi}_i \cdot \hat{\boldsymbol{k}} \, \mathrm{d}\boldsymbol{x} \qquad (30)$$

For cases with more general rheologies, in particular those with a strain-rate dependent viscosity, the system $\underline{F}_{\text{Stokes}}(\boldsymbol{u}, p) = \underline{0}$
is nonlinear and can be solved using Newton's method. This requires the solution in every Newton iteration of a linear system
of the same form as in Equation (27) but with an additional term in $K$ associated with $\partial \mu / \partial \boldsymbol{u}$.

There is a wide literature on iterative methods for solving saddle point systems of the form in Equation (27). For an overview
of the methods commonly used in geodynamics, see May and Moresi (2008). Here we employ the Schur complement approach,
where pressure $\underline{p}$ is determined by solving

$$G^T K^{-1} G \underline{p} = G^T K^{-1} \underline{f} \quad . \qquad (31)$$





It should be noted that $K^{-1}$ is not assembled explicitly. Rather, in a first step we obtain $\underline{y} = K^{-1}\underline{f}$ by solving $K\underline{y} = \underline{f}$ so that we can construct the right-hand side of the equation. We subsequently apply an iterative method to the linear system as a whole,

in which each iteration requires matrix-vector multiplication with the matrix $G^T K^{-1} G$ that again involves the solution of a linear system with matrix $K$. In addition to this matrix-vector multiplication we also need a suitable preconditioner. Here we follow the inverse scaled-mass matrix approach which uses the following approximation

$$G^T K^{-1} G \approx M, \quad M_{ij} = \int_\Omega \mu \psi_i \psi_j \tag{32}$$

Finally, after solving Equation (31) for $\underline{p}$, we obtain $\underline{u}$ in a final solve $K\underline{u} = \underline{f} - G\underline{p}$.

Since this solution process involves multiple solves with the matrix $K$, we also need an efficient algorithm to solve that system. For this, we combine the conjugate gradient method with an algebraic multigrid approach, specifically the Geometric Algebraic Multigrid (GAMG) method implemented in PETSc (Balay et al., 1997, 2021a, b).

Depending on boundary conditions, the linearised Stokes system admits a number of null modes. In the absence of open boundaries, which is the case for all cases examined here, the pressure admits a constant null mode, where any arbitrary

constant can be added to the pressure solution and remain a valid solution to the equations. In addition, the cylindrical and spherical cases with free-slip boundary conditions at both boundaries examined in Section 5, admit, respectively, one and three independent rotational null modes in velocity. These null modes result in indefinite matrices and preconditioned iterative methods typically require the null vectors to be provided so that they can be projected out during iteration.

In the absence of any Dirichlet conditions on velocity, the nullspace of the velocity block $K$ also consists of a further

two independent translational modes in 2D, and three in 3D. Even if, as for the cases here, the boundary conditions do not admit all rotational and translational modes, these solutions are still associated with low energy modes of the matrix, and some multigrid methods use this information to improve their performance by ensuring that these near-nullspace modes are accurately represented at the coarser levels (Vanek et al., 1996). We make use of this in several of the examples considered below.

**5 Examples: Benchmark Cases and Validation**

Firedrake provides a complete framework for solving finite element problems, highlighted in this section through a series of examples. We start in Section 5.1 with the most basic problem – isoviscous, incompressible convection, in an enclosed 2-D Cartesian box – and systematically build complexity, initially moving into more realistic physical approximations (Section 5.2) and, subsequently, geometries that are more representative of Earth's mantle (Section 5.3).

**5.1 Basic Example: 2-D Convection in a Square Box**

A simple 2-D square convection problem, from Blankenbach et al. (1989), for execution in Firedrake, is displayed in Listing 1. The problem is incompressible, isoviscous, heated from below and cooled from above, with closed, free-slip boundaries, on a unit square mesh. Solutions are obtained by solving the Stokes equations for velocity and pressure, alongside the energy





```python
from firedrake import *

# Mesh – use a built in meshing function:
mesh = UnitSquareMesh(40, 40, quadrilateral=True)

# Function spaces:
V = VectorFunctionSpace(mesh, "CG", 2) # Velocity function space (vector)
W = FunctionSpace(mesh, "CG", 1) # Pressure function space (scalar)
Q = FunctionSpace(mesh, "CG", 2) # Temperature function space (scalar)
Z = MixedFunctionSpace([V, W]) # Mixed function space

# Test functions and functions to hold solutions:
v, w = TestFunctions(Z)
q = TestFunction(Q)
z = Function(Z)
u, p = split(z) # Returns symbolic UFL expression for u and p
Told, Tnew = Function(Q, name="OldTemp"), Function(Q, name="NewTemp")
Ttheta = 0.5 * Tnew + 0.5 * Told # Temporal discretisation through Crank-Nicholson

# Initialise temperature field:
X = SpatialCoordinate(mesh)
Told.interpolate(1.0 - X[1] + 0.05 * cos(pi * X[0]) * sin(pi * X[1]))
Tnew.assign(Told)

# Important constants:
Ra = Constant(1e4) # Rayleigh number
mu = Constant(1.0) # Viscosity – constant for this isoviscous case
kappa = Constant(1.0) # Thermal diffusivity
delta_t = Constant(1e-4) # Time-step
k = Constant((0, 1)) # Unit vector (in direction opposite to gravity)

# Stokes equations in UFL form:
stress = 2 * mu * sym(grad(u))
F_stokes = inner(grad(v), stress) * dx + dot(v, grad(p)) * dx - (dot(v, k) * Ra * Ttheta) * dx
F_stokes += dot(grad(w), u) * dx  # Continuity equation
# Energy equation in UFL form:
F_energy = q * (Tnew - Told) / delta_t * dx + q * dot(u, grad(Ttheta)) * dx + dot(grad(q), kappa *
    grad(Ttheta)) * dx

# Set up boundary conditions and deal with nullspaces:
bcvx, bcvy = DirichletBC(Z.sub(0).sub(0), 0, (1, 2)), DirichletBC(Z.sub(0).sub(1), 0, (3, 4))
bctb, bctt = DirichletBC(Q, 1.0, 3), DirichletBC(Q, 0.0, 4)
p_nullspace = MixedVectorSpaceBasis(Z, [Z.sub(0), VectorSpaceBasis(constant=True)])

# Initialise output:
output_file = File('output.pvd') # Create output file
u, p = z.split()
u.rename("Velocity"), p.rename("Pressure")

# Solver dictionary:
solver_parameters = {
    "mat_type": "aij",
    "snes_type": "ksponly",
    "ksp_type": "preonly",
    "pc_type": "lu",
    "pc_factor_mat_solver_type": "mumps",
}

# Setup problem and solver objects so we can reuse (cache) solver setup
stokes_problem = NonlinearVariationalProblem(F_stokes, z, bcs=[bcvx, bcvy])
stokes_solver = NonlinearVariationalSolver(stokes_problem, solver_parameters=solver_parameters,
    nullspace=p_nullspace, transpose_nullspace=p_nullspace)
energy_problem = NonlinearVariationalProblem(F_energy, Tnew, bcs=[bctb, bctt])
energy_solver = NonlinearVariationalSolver(energy_problem, solver_parameters=solver_parameters)

for timestep in range(0, 1000):  # Perform time loop for 1000 steps
    if timestep % 10 == 0:
        output_file.write(u, p, Tnew)
    stokes_solver.solve()
    energy_solver.solve()
    Told.assign(Tnew)
```

**Listing 1.** Firedrake code required to reproduce 2-D Cartesian incompressible isoviscous benchmark cases from Blankenbach et al. (1989).

equation for temperature. The initial temperature distribution is prescribed as follows:

$$T(x,y) = (1 - y) + A\cos(\pi x)\sin(\pi y),$$ (33)





where $A = 0.05$ is the amplitude of the initial perturbation.

We have set up the problem using a bilinear quadrilateral element pair (Q2-Q1) for velocity and pressure, with Q2 elements for temperature. Firedrake user code is written in Python, so the first step, illustrated on line 1 of Listing 1, is to import the Firedrake module. We next need a mesh: for simple domains such as the unit square, Firedrake provides built-in meshing

functions. As such, line 4 defines the mesh, with 40 quadrilateral elements in x and y directions. We also need function spaces, which is achieved by associating the mesh with the relevant finite element on lines 7-9: $V$, $W$ and $Q$ are symbolic variables representing function spaces. They also contain the function space's computational implementation, recording the association of degrees of freedom with the mesh and pointing to the finite element basis. The user does not usually need to pay any attention to this: the function space just behaves as a mathematical object (Rathgeber et al., 2016). Function spaces can be combined

in the natural way to create mixed function spaces, as we do on line 10, combining the velocity and pressure function spaces to form a function space for the mixed Stokes problem, $Z$. Note that although we use continuous Lagrange elements (CG) in all examples presented herein, Firedrake offers a range of different options, including discontinuous elements (DG). Test functions, $\boldsymbol{v}$, $w$ and $q$ are subsequently defined (lines 13-14) and we also specify functions to hold our solutions (lines 15-18): $z$ in the mixed function space, noting that a symbolic representation of the two parts – velocity and pressure – is obtained with

`split` on line 16, and $T_{\text{old}}$ and $T_{\text{new}}$ (line 17), required for the Crank-Nicholson scheme used for temporal discretisation in our energy equation (see Equations 18 and 19 in Section 4.2), where $T_\theta$ is defined on line 18.

We obtain symbolic expressions for coordinates in the physical mesh (line 21) and subsequently use these to initialize the old temperature field, via Equation (33), on line 22. This is where Firedrake transforms a symbolic operation into a numerical computation for the first time: the `interpolate` method generates C code that evaluates this expression at the nodes of $T_{\text{old}}$,

and immediately executes it to populate the values of $T_{\text{old}}$. We initialize $T_{\text{new}}$ with the values of $T_{\text{old}}$, on line 23, via the `assign` function. Important constants in this problem (Rayleigh Number, $Ra$; viscosity, $\mu$; thermal diffusivity, $\kappa$), in addition to the constant timestep ($\Delta_t$) and unit vector ($k$), are defined on lines 26-30. We note that viscosity could also be a `Function`, if we wanted spatial variation.

We are now in a position to define the variational problems expressed in Equations (24) and (18). Although in this test case

the problems are linear, we maintain the more general nonlinear residual form $F_{\text{Stokes}}(\boldsymbol{v}, \boldsymbol{u}) = 0$ and $F_{\text{energy}}(q, T) = 0$, to allow for straightforward extension to nonlinear problems below. The symbolic expressions for $F_{\text{Stokes}}$ and $F_{\text{Energy}}$ in the UFL are given on lines 33-37: the resemblance to the mathematical formulation is immediately apparent. Integration over the domain is indicated by multiplication with `dx`.

Strong Dirichlet boundary conditions for velocity (bcvx, bcvy) and temperature (bctb, bctt) are specified on lines 40-41. A

Dirichlet boundary condition is created by constructing a Python `DirichletBC` object, where the user must provide the function space the condition applies to, the value, and the part of the mesh at which it applies. The latter uses integer mesh markers which are commonly used by mesh generation software to tag entities of meshes. Boundaries are automatically tagged by the built-in meshes supported by Firedrake. For the `UnitSquareMesh` being used here, tag 1 corresponds to the plane $x = 0$; 2 to $x = 1$; 3 to $y = 0$; and 4 to $y = 1$. Note how boundary conditions are being applied to the velocity part of the mixed finite

element space $Z$, indicated by `Z.sub(0)`. Within `Z.sub(0)` we can further subdivide into `Z.sub(0).sub(0)` and `Z.sub(0).sub`



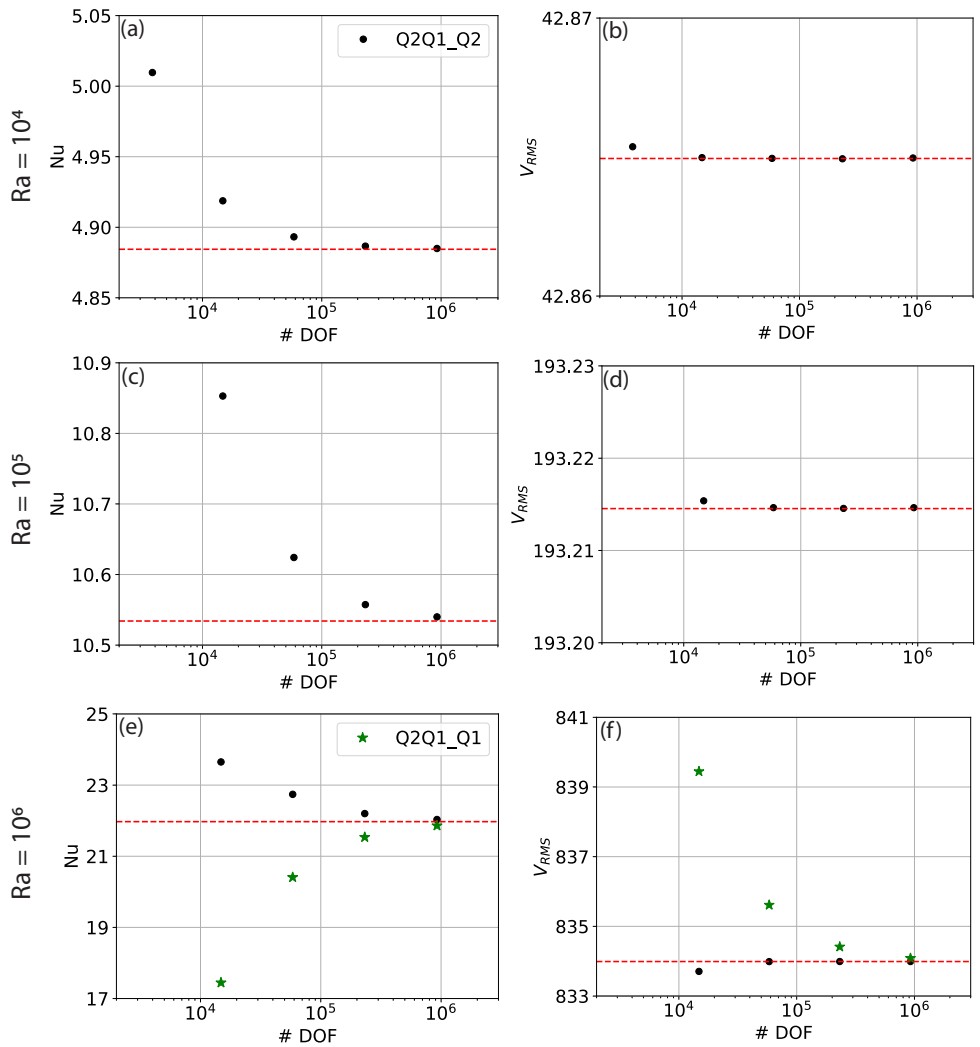

**Figure 1.** Results from 2-D incompressible isoviscous square convection benchmark cases: (a) Nusselt number versus number of pressure and velocity degrees of freedom (DOF), at $Ra = 1 \times 10^4$ (Case 1a - Blankenbach et al., 1989), for a series of uniform, structured meshes; (b) RMS velocity versus number of pressure and velocity DOF, at $Ra = 1 \times 10^4$; (c, d) as in panels a and b, but at $Ra = 1 \times 10^5$ (Case 1b - Blankenbach et al., 1989); (e, f) at $Ra = 1 \times 10^6$ (Case 1c - Blankenbach et al., 1989). Benchmark values are denoted by dashed red lines. In panels e and f, we also display results from simulations where temperature is represented through a Q1 discretisation (Q2Q1_Q1), for comparison with our standard Q2 temperature discretisations (Q2Q1_Q2).

(1) to apply boundary conditions to the $x$ and $y$ components of the velocity field only. To apply conditions to the pressure space, we would use `z.sub(1)`. This problem has a constant pressure nullspace and we must ensure that our solver removes this space. To do so, we build a nullspace object on line 42, which will subsequently be passed to the solver, and PETSc will seek a solution in the space orthogonal to the provided nullspace.



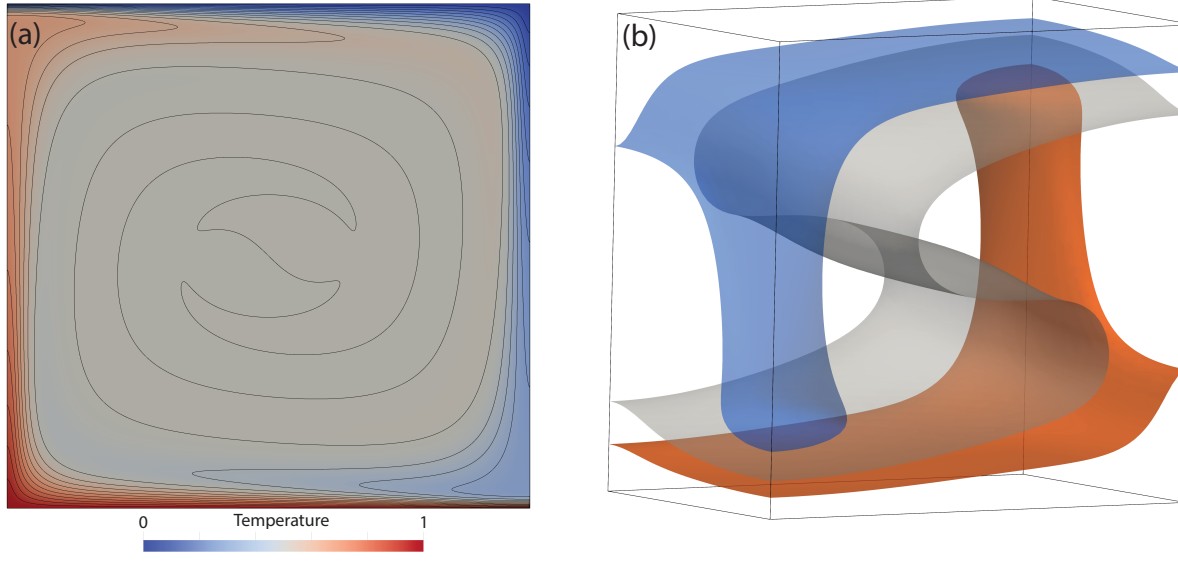

**Figure 2.** Final steady-state temperature field, in 2-D and 3-D, from Firedrake simulations, designed to match: (a) Case 1a from (Blankenbach et al., 1989), with contours spanning temperatures of 0 to 1, at 0.05 intervals; (b) Case 1a from (Busse et al., 1994), with transparent isosurfaces plotted at $T = 0.3$, $0.5$ and $0.7$.

We finally come to solving the variational problem, with problems and solver objects created on lines 59–62. We pass in the residual functions $F_{\text{Stokes}}$ and $F_{\text{Energy}}$, solution fields (z, $T_{\text{new}}$), boundary conditions and, for the Stokes system, the nullspace object. Solution of the two variational problems is undertaken by the PETSc library (Balay et al., 1997), guided by the solver parameters specified on lines 50–56 (see Balay et al., 2021a, b, for comprehensive documentation of all PETSc options). The first option on line 51, instructs the Jacobian to be assembled in PETSc's default `aij` sparse matrix type. Although the Stokes and

energy problem in this example are linear, for consistency with latter cases, we use Firedrake's `NonlinearVariationalSolver` which makes use of PETSc's Scalable Nonlinear Equations Solvers (SNES) interface. However, since we do not actually need a nonlinear solver for this case, we choose the `ksponly` method on line 52 indicating that only a single linear solve needs to be performed. The linear solvers are configured through PETSc's Krylov Subspace (KSP) interface, where we can request a direct solver by choosing the `preonly` KSP method, in combination with `lu` as the 'preconditioner' (PC) type (lines 53–54). The

specific implementation of the LU-decomposition based direct solver is selected on line 55, as the MUMPS library (Amestoy et al., 2001, 2019). As we shall see through subsequent examples, the solution process is fully programmable, enabling the creation of sophisticated solvers by combining multiple layers of Krylov methods and preconditioners (Kirby and Mitchell, 2018).

    The time-loop is initiated on line 64, with the Stokes system solved on line 67 and the energy equation on line 68. These

`solve` calls once again convert symbolic mathematics into computation. The linear systems for both problems are based on the Jacobian matrix, and a right-hand side vector based on the residual, as indicated in Equations (21), (22) and (23) for the energy





equation, and Equations (27), (28), (29) and (30) for the Stokes equation. Note, however, that the symbolic expression for the Jacobian is derived automatically in UFL. Firedrake's TSFC (Homolya et al., 2018) subsequently converts the UFL into highly optimised assembly code, which is then executed to create the matrix and vectors, with the resulting system passed to PETSc

for solution. Finally, we note that output is written on line 66, to a $.pvd$ file, initialised on line 45, for visualisation in software such as ParaView (e.g. Ahrens et al., 2005).

In $< 70$ lines of Python, we are able to produce a model that can be executed and quantitatively compared with benchmark results from Blankenbach et al. (1989). To do so, we have computed the RMS velocity andsurface Nusselt number at a range of different mesh resolutions and Rayleigh numbers, with results presented in Figure 1. Results converge towards the benchmark

solutions, with increasing resolution. The final steady-state temperature field, at $Ra = 1 \times 10^6$, is illustrated in Figure 2(a).

To further highlight the flexibility of Firedrake, we have also simulated these cases using a Q1 discretisation for the temperature field. The modifications necessary are minimal: on line 9, the polynomial order is changed from 2 to 1. Results, at $Ra = 1 \times 10^6$, are presented in Figure 1(e,f), again converging towards benchmark values with increasing resolution. We find that, as expected, a Q2 temperature discretisation leads to more accurate results, although results converge towards the bench-

mark solutions from different directions. For the remainder of the examples considered herein, we use a Q2 discretisation for temperature.

## 5.2 Extension: more realistic physics

We next highlight the ease at which simulations can be updated to incorporate more realistic physical approximations. We first account for compressibility, under the Anelastic Liquid Approximation (e.g. Schubert et al., 2001), simulating a well-

established benchmark case from King et al. (2009) (Section 5.2.1). We subsequently focus on a case with a more Earth-like approximation of the rheology (Section 5.2.2), simulating another well-established benchmark case from Tosi et al. (2015). All cases are set up in an enclosed 2-D Cartesian box with free-slip boundary conditions, with the required changes discussed relative to the base case presented in Section 5.1.

### 5.2.1 Compressibility

The governing equations applicable for compressible mantle convection, under the Anelastic Liquid Approximation (ALA), are presented in Appendix A (based on, for example, Schubert et al., 2001). Their weak forms are derived by multiplying these equations with appropriate test functions and integrating over the domain, as we did with their incompressible counterparts in Section 4. They differ appreciably from the incompressible approximations that have been utilised thus far, with important updates to all three governing equations. Despite this, the changes required to incorporate these equations, within UFL and

Firedrake, are minimal.

Although King et al. (2009) examined a number of cases, we focus on one illustrative example here, at $Ra = 10^5$ and a dissipation number $Di = 0.5$. This allows us to demonstrate the ease at which these cases can be configured within Firedrake. The required changes, relative to the base case, are displayed in Listing 2. They can be summarised as follows:

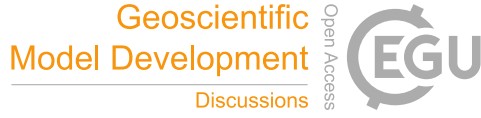

```
# Additional constants and definition of compressible reference state:
Ra = Constant(1e5) # Rayleigh number
Di = Constant(0.5) # Dissipation number
T0 = Constant(0.091) # Non-dimensional surface temperature
tcond = Constant(1.0) # Thermal conductivity
rho_0, alpha, cpr, cvr, gruneisen = 1.0, 1.0, 1.0, 1.0, 1.0
rhobar = Function(Q, name="CompRefDensity").interpolate(rho_0 * exp(((1.0 - X[1]) * Di) / alpha))
8  Tbar = Function(Q, name="CompRefTemperature").interpolate(T0 * exp((1.0 - X[1]) * Di) - T0)
alphabar = Function(Q, name="IsobaricThermalExpansivity").assign(1.0)
cpbar = Function(Q, name="IsobaricSpecificHeatCapacity").assign(1.0)
chibar = Function(Q, name="IsothermalBulkModulus").assign(1.0)
FullT = Function(Q, name="FullTemperature").assign(Tnew+Tbar)
--------------------------------------------------------------------------------
# Equations in UFL:
I = Identity(2)
stress = 2 * mu * sym(grad(u)) - 2./3.*I*mu*div(u)
F_stokes = inner(grad(v), stress) * dx + dot(v, grad(p)) * dx - (dot(v,k) * (Ra * Ttheta * rhobar
       * alphabar - (Di/gruneisen) * (cpr/cvr)*rhobar*chibar*p) * dx)
F_stokes += dot(grad(w), rhobar*u) * dx # Mass conservation
F_energy = q * rhobar * cpbar * ((Tnew - Told) / delta_t) * dx + q * rhobar * cpbar * dot(u, grad(
       Ttheta)) * dx + dot(grad(q), tcond * grad(Tbar + Ttheta)) * dx + q * (alphabar * rhobar * Di *
       u[1] * Ttheta) * dx  - q * ( (Di/Ra) * inner(stress, grad(u)) ) * dx
--------------------------------------------------------------------------------
# Temperature boundary conditions:
bctb, bctt = DirichletBC(Q, 1.0 - (T0*exp(Di) - T0), bottom_id), DirichletBC(Q, 0.0, top_id)
--------------------------------------------------------------------------------
# Pressure nullspace:
stokes_solver = NonlinearVariationalSolver(stokes_problem, solver_parameters=solver_parameters,
       transpose_nullspace=p_nullspace)
```

**Listing 2.** Difference in Firedrake code required to reproduce compressible ALA cases from King et al. (2009) relative to our base case.

1. Definition and initialisation of additional constants and the 1-D reference state, derived here via an Adams-Williamson equation of state (lines 1-12). In this benchmark example, several of the key constants and parameters required for compressible convection are assigned values of 1 and could be removed. However, to ensure consistency between the governing equations presented in Appendix A and the UFL, we chose not to omit these constants in Listing 2.

2. The UFL for the momentum, mass conservation and energy equations is updated, emphasising once again the resemblance to the mathematical formulation (lines 16-20). The key changes are as follows: (i) the stress tensor is updated to account for a non-zero velocity divergence (line 17), where `Identity` represents a unit matrix of a given size (2 in this case) and `div` represents the symbolic divergence of a field; (ii) the Stokes equations are further modified to account for dynamic pressure's influence on buoyancy (final term on line 18); (iii) the mass conservation equation includes the depth-dependent reference density, $\bar{\rho}$ (line 19); and (iv) the energy equation is updated to incorporate adiabatic heating and viscous dissipation terms (final 2 terms on line 20).

3. Temperature boundary conditions are updated, noting that we are solving for deviatoric temperature rather than the full temperature, which also includes the reference state.

4. In our Stokes solver, we only specify the `transpose_nullspace` option (as opposed to both `nullspace` and `transpose_nullspace` options for our base case): the incorporation of dynamic pressure's impact on buoyancy implies that the (right-hand side) pressure nullspace is no longer the same as the (left-hand side) transpose nullspace. The transpose nullspace remains the same space of constant pressure solutions, and is used to project out these modes from the initial residual vector to ensure



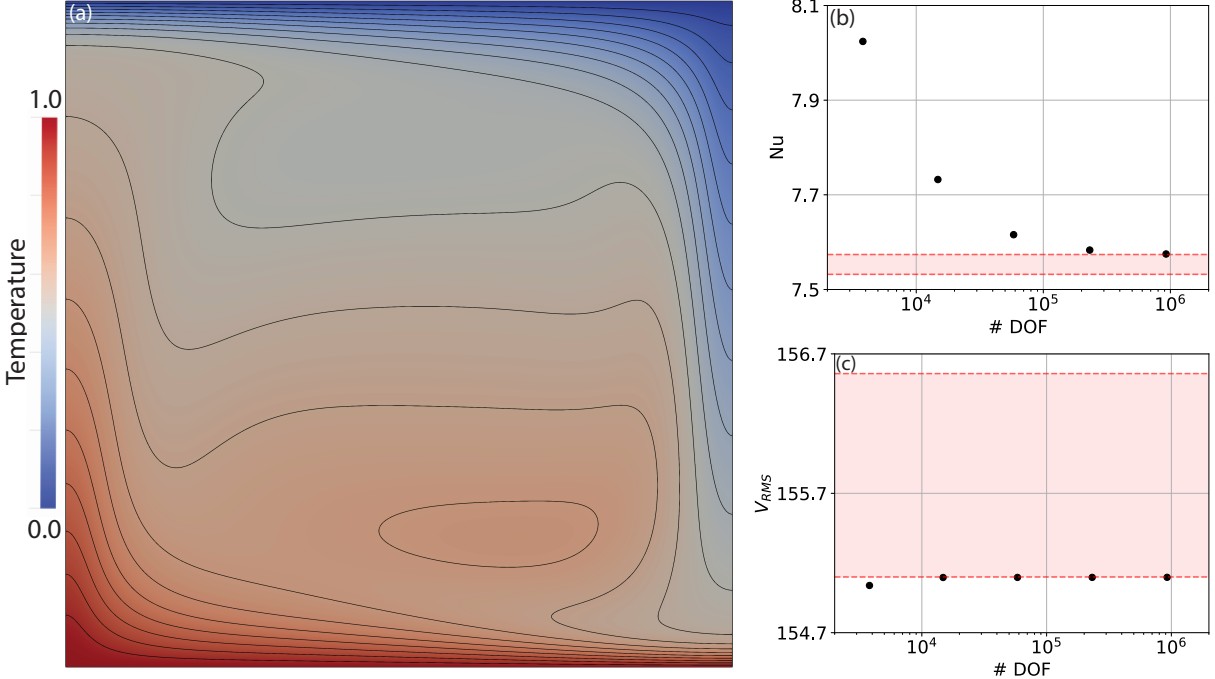

**Figure 3.** Results from Firedrake simulations configured to reproduce 2-D compressible benchmark case from King et al. (2009) at $Ra = 10^5$ and $Di = 0.5$: (a) final steady-state (full) temperature field, with contours spanning temperatures of 0 to 1, at 0.05 intervals; (b) Nusselt number versus number of pressure and velocity DOF, for a series of uniform, structured meshes; (c) RMS velocity versus number of pressure and velocity DOF. The range of solutions provided by different codes in the King et al. (2009) benchmark study are bounded by dashed red lines.

that the linear system is well-posed. The right-hand side nullspace now consists of different modes, which can be found through integration. However, this nullspace is only required for iterative linear solvers in which the modes are projected out from the solution vector at each iteration to prevent its unbounded growth.

470 We note that in setting up the Stokes solver as we have, we incorporate the pressure effect on buoyancy implicitly, as advocated by Leng and Zhong (2008). As this term depends on the pressure that we are solving for, an extra term is required in addition to the pressure gradient matrix G in the Jacobian matrix in Equation (27). The inclusion of $\bar{\rho}$ in the continuity constraint also means that this term is no longer simply represented by the transpose of G. Such changes are automatically incorporated by Firedrake, highlighting a major benefit of the automatic assembly approach that is utilised. To ensure the validity of our approach, we have computed the RMS velocity and Nusselt number at a range of different mesh resolutions, for direct comparison

475 with King et al. (2009), with results presented in Figure 3, alongside the final steady-state (full) temperature field. As expected, results converge towards the benchmark solutions, with increasing resolution, demonstrating the applicability and accuracy of Firedrake for compressible simulations of this nature.



```
# Stokes solver dictionary:
stokes_solver_parameters = {
"mat_type": "aij",
"snes_type": "newtonls",
"snes_linesearch_type": "l2",
"snes_max_it": 100,
"snes_atol": 1e-10,
"ksp_type": "preonly",
"pc_type": "lu",
"pc_factor_mat_solver_type": "mumps",
}
# Energy solver dictionary:
energy_solver_parameters = {
"mat_type": "aij",
"snes_type": "ksponly",
"ksp_type": "preonly",
"pc_type": "lu",
"pc_factor_mat_solver_type": "mumps",
}
----------------------------------------------------------------------------------------
# Viscosity calculation and Rayleigh number:
Ra = Constant(100.) # Rayleigh number
gamma_T, gamma_Z = Constant(ln(10**5)), Constant(ln(10))
mu_star, sigma_y = Constant(0.001), Constant(1.0)
epsilon = 0.5 * (grad(u)+transpose(grad(u))) # strain-rate
epsii = sqrt(inner(epsilon,epsilon) + 1e-20) # 2nd invariant (with tolerance to ensure stability)
mu_lin = exp(-gamma_T*Tnew + gamma_Z*(1 - X[1]))
mu_plast = mu_star + (sigma_y / epsii)
mu = (2. * mu_lin * mu_plast) / (mu_lin + mu_plast)
----------------------------------------------------------------------------------------
# Updated solver:
stokes_solver = NonlinearVariationalSolver(stokes_problem, solver_parameters=
        stokes_solver_parameters, nullspace=p_nullspace, transpose_nullspace=p_nullspace)
energy_solver = NonlinearVariationalSolver(energy_problem, solver_parameters=
        energy_solver_parameters)
```

**Listing 3.** Difference in Firedrake code required to reproduce viscoplastic rheology cases from Tosi et al. (2015) relative to our base case.

### 5.2.2 Viscoplastic Rheology

To illustrate the changes necessary to incorporate a viscoplastic rheology, which is more representative of deformation within

Earth's mantle and lithosphere, we examine a case from Tosi et al. (2015), a benchmark study intended to form a straightforward extension to Blankenbach et al. (1989). Indeed, aside from the viscosity and reference Rayleigh Number ($Ra_0 = 10^2$), all other aspects of this case are identical to the case presented in Section 5.1. The viscosity field, $\mu$, is calculated as the harmonic mean between a linear component, $\mu_{\text{lin}}$ and a nonlinear, plastic component, $\mu_{\text{plast}}$, which is dependent on the strain-rate, as follows:

$$\mu(T, z, \dot{\epsilon}) = 2\Big(\frac{1}{\mu_{\text{lin}(T,z)}} + \frac{1}{\mu_{\text{plast}(\dot{\epsilon})}}\Big)^{-1}. \tag{34}$$

The linear part is given by an Arrhenius law (the so-called Frank-Kamenetskii approximation):

$$\mu_{\text{lin}(T,z)} = \exp(-\gamma_T T + \gamma_z z), \tag{35}$$

where $\gamma_T = \ln(\Delta\mu_T)$ and $\gamma_z = \ln(\Delta\mu_z)$ are parameters controlling the total viscosity contrast due to temperature and depth, respectively. The nonlinear component is given by:

$$\mu_{\text{plast}}(\dot{\epsilon}) = \mu^\star + \frac{\sigma_y}{\sqrt{\dot{\epsilon} : \dot{\epsilon}}} \tag{36}$$





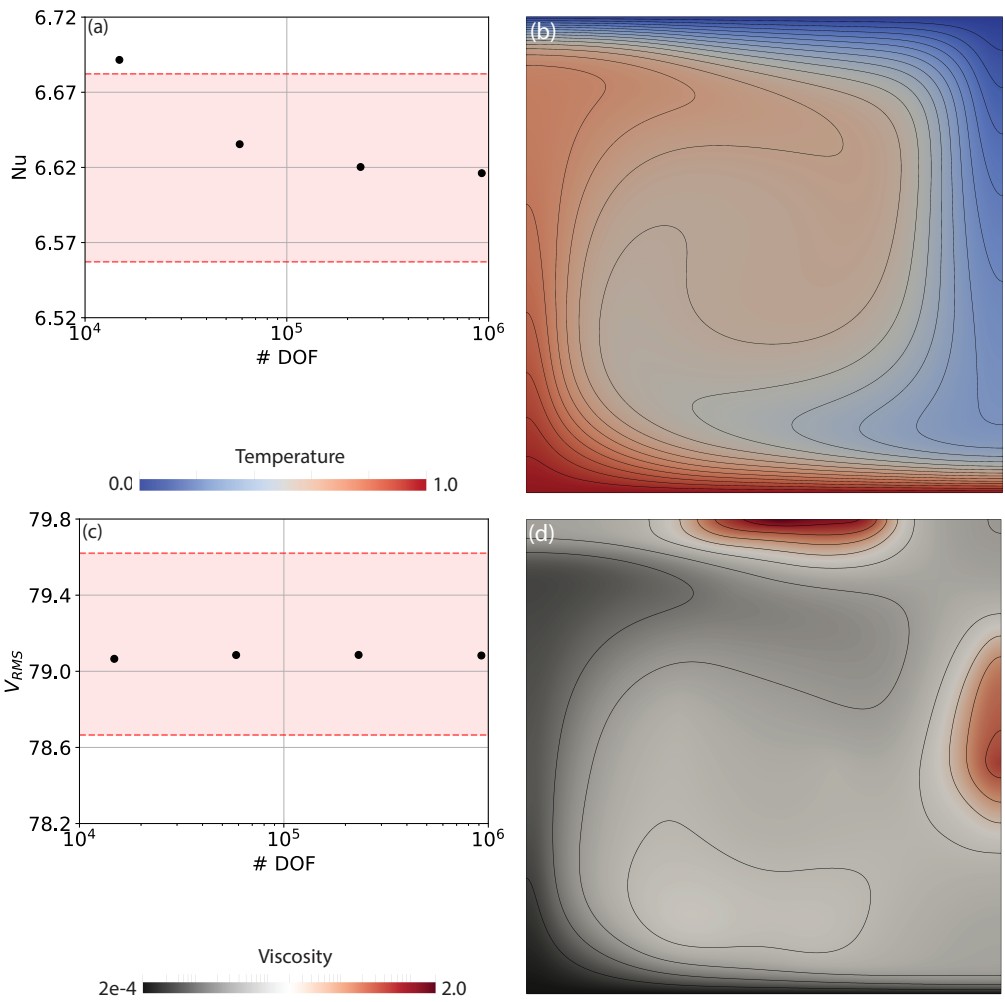

**Figure 4.** Results from 2-D benchmark case from Tosi et al. (2015), with a viscoplastic rheology, at $Ra_0 = 10^2$: (a) Nusselt number versus number of pressure and velocity DOF, for a series of uniform, structured meshes; (b) final steady-state temperature field, with contours spanning temperatures of 0 to 1, at 0.05 intervals; (c) RMS velocity versus number of pressure and velocity DOF; (d) final steady-state viscosity field (note logarithmic scale). In panels a and c, the range of solutions provided by different codes in the Tosi et al. (2015) benchmark study are bounded by dashed red lines.

where $\mu^\star$ is a constant representing the effective viscosity at high stresses and $\sigma_y$ is the yield stress. The denominator of the second term in Equation (36) represents the second invariant of the strain-rate tensor. The viscoplastic flow law (Eq. 34) leads to linear viscous deformation at low stresses and plastic deformation at stresses that exceed $\sigma_y$, with the decrease in viscosity limited by the choice of $\mu^\star$.





Although Tosi et al. (2015) examined a number of cases, we focus on one here (Case 4: $Ra_0 = 10^2$, $\Delta\mu_T = 10^5$, $\Delta\mu_y = 10$

and $\mu^\star = 10^{-3}$), which allows us to demonstrate how a temperature-, depth- and strain-rate dependent viscosity is incorporated within Firedrake. The changes required to simulate this case, relative to our base case, are displayed in Listing 3. These are:

1. Linear solver options are no longer applicable, given the dependence of of viscosity on the flow field, through the strain-rate. Accordingly, the solver dictionary is updated to account for the nonlinear nature of our Stokes system (lines 2-11). For the first time, we fully-exploit the SNES, using a setup based on Newton's method (`"snes_type": "newtonls"`)

with a secant line search over the L2-norm of the function (`"snes_linesearch_type": "l2"`). As we target a steady-state solution, an absolute tolerance is specified for our nonlinear solver (`"snes_atol": 1e-10`).

2. Solver options differ between the (nonlinear) Stokes and (linear) energy systems. As such, a separate solver dictionary is specified for solution of the energy equation (lines 13-20). Consistent with our base case, we use a direct solver for solution of the energy equation, based on the Mumps library.

3. Viscosity is calculated as a function of temperature, depth ($\mu_{\mathrm{lin}}$ - line 29) and strain-rate ($\mu_{\mathrm{plast}}$ - line 30), using constants specified on lines 25-26. Linear and nonlinear components are subsequently combined via a harmonic mean (line 31).

4. Updated solver dictionaries are incorporated into their respective solvers on lines 35 and 36, noting that for this case both the nullspace and transpose_nullspace options are provided for the Stokes system, consistent with the base case.

We note that even though the UFL for the Stokes and energy systems remains identical to our base case, assembly of addi-

tional terms in the Jacobian, associated with the nonlinearity in this system, is once again handled automatically by Firedrake. To compare our results to those of Tosi et al. (2015) we have computed the RMS velocity and Nusselt number at a range of different mesh resolutions. These are presented in Figure 4 and, once again, results converge towards the benchmark solutions, with increasing resolution. Final steady-state temperature and viscosity fields are also illustrated to allow for straightforward comparison with those presented by Tosi et al. (2015), illustrating that viscosity varies by roughly four orders of magnitude

across the computational domain.

Taken together, our compressible and viscoplastic rheology results demonstrate the accuracy and applicability of Firedrake for problems incorporating a range of different approximations to the underlying physics. They have allowed us to illustrate Firedrake's flexibility: by leveraging UFL and PETSc, the framework is easily extensible, allowing for straightforward application to scenarios involving different physical approximations, even if they require distinct solution strategies.

**5.3 Extension: Dimensions and Geometry**

In this section we highlight the ease at which simulations can be examined in different dimensions and geometries, by modifying our basic 2-D case. We primarily simulate benchmark cases that are well-known within the geodynamical community, initially matching the steady-state, isoviscous simulation of Busse et al. (1994) in a 3-D Cartesian domain. There is currently no published community benchmark for simulations in the 2-D cylindrical domain. As such, we next compare results for

an isoviscous, steady-state case, in a 2-D cylindrical domain, with those of the Fluidity computational modelling framework





(Davies et al., 2011), which has been carefully validated against the extensive set of analytical solutions introduced by Kramer et al. (2021a), in both cylindrical and spherical geometries. Finally, we analyze an isoviscous 3-D spherical benchmark case from Zhong et al. (2008). Once again, the changes required to run these cases are discussed relative to our base case (Section 5.1), unless noted otherwise.

### 5.3.1   3-D Cartesian Domain

We first examine and validate our setup in a 3-D Cartesian domain, for a steady-state, isoviscous case – specifically Case 1a from Busse et al. (1994). The domain is a box of dimensions $1.0079 \times 0.6283 \times 1$. The initial temperature distribution, chosen to produce a single ascending and descending flow, at $x = y = 0$ and $(x = 1.0079, y = 0.6283)$, respectively, is prescribed as:

$$T(x,y,z) = \left[ \frac{\mathrm{erf}(4(1-z)) + \mathrm{erf}(-4z) + 1}{2} \right] + A[\cos(\pi x/1.0079) + \cos(\pi y/0.6283)] \sin(\pi z), \tag{37}$$

where $A = 0.2$ is the amplitude of the initial perturbation. Boundary conditions for temperature are T = 0 at the surface (z = 1) and T = 1 at the base (z = 0), with insulating (homogeneous Neumann) sidewalls. No-slip velocity boundary conditions are specified at the top surface and base of the domain, with free-slip boundary conditions on all sidewalls. The Rayleigh number $Ra = 3 \times 10^4$.

In comparison to Listing 1, the changes required to simulate this case, using trilinear (Q2-Q1) elements for velocity and pressure, are minimal. The key differences, summarised in Listing 4, are:

1. The creation of the underlying mesh (lines 1-5), which we generate by extruding a 2-D quadrilateral mesh in the $z$-direction to a layered 3-D hexahedral mesh. Our final mesh has $20 \times 12 \times 20$ elements, in $x$-, $y$- and $z$-directions, respectively. For extruded meshes, top and bottom boundaries are tagged by `top` and `bottom`, respectively, whilst boundary markers from the base mesh can be used to set boundary conditions on the relevant side of the extruded mesh. We note that Firedrake exploits the regularity of extruded meshes to enhance performance.

2. Specification of the initial condition for temperature, following Equation (37), updated values for $Ra$, and definition of the 3-D unit vector (lines 9-11).

3. The inclusion of Python dictionaries that define iterative solver parameters for the Stokes and energy systems (lines 15-47). Although direct solves provide robust performance in the 2-D cases examined above, in 3-D the computational (CPU and memory) requirements quickly become intractable. PETSc's `fieldsplit pc_type` provides a class of preconditioners for mixed problems that allows one to apply different preconditioners to different blocks of the system. Here we configure the Schur complement approach as described in Section 4.3.

   The `fieldsplit_0` entries configure solver options for the first of these blocks, the K matrix. The linear systems associated with this matrix are solved using a combination of the Conjugate Gradient method (`cg`, line 23) and an algebraic multigrid preconditioner (`gamg`, line 27). We also specify two options (`gamg_threshold` and `gamg_square_graph`) that


```python
# Mesh Generation:
a, b, c, nx, ny, nz = 1.0079, 0.6283, 1.0, 20, int(0.6283/1.0 * 20), 20
mesh2d = RectangleMesh(nx, ny, a, b, quadrilateral=True) # Rectangular 2D mesh
mesh = ExtrudedMesh(mesh2d, nz)
bottom_id, top_id, left_id, right_id, front_id, back_id = "bottom", "top", 1, 2, 3, 4

# -------------------------------------------------------------------------------
# Initial condition and constants:
Told.interpolate(0.5*(erf((1-X[2])*4)+erf(-X[2]*4)+1) + 0.2*(cos(pi*X[0]/a)+cos(pi*X[1]/b))*sin(pi
    *X[2]))
Ra = Constant(3e4) # Rayleigh number
k = Constant((0, 0, 1)) # Unit vector (in direction opposite to gravity).

# -------------------------------------------------------------------------------
# Stokes Equation Solver Parameters:
stokes_solver_parameters = {
    "mat_type": "matfree",
    "snes_type": "ksponly",
    "ksp_type": "preonly",
    "pc_type": "fieldsplit",
    "pc_fieldsplit_type": "schur",
    "pc_fieldsplit_schur_type": "full",
    "fieldsplit_0": {
        "ksp_type": "cg",
        "ksp_rtol": 1e-7,
        "pc_type": "python",
        "pc_python_type": "firedrake.AssembledPC",
        "assembled_pc_type": "gamg",
        "assembled_pc_gamg_threshold": 0.01,
        "assembled_pc_gamg_square_graph": 100,
    },
    "fieldsplit_1": {
        "ksp_type": "fgmres",
        "ksp_rtol": 1e-6,
        "pc_type": "python",
        "pc_python_type": "firedrake.MassInvPC",
        "Mp_ksp_rtol": 1e-5,
        "Mp_ksp_type": "cg",
        "Mp_pc_type": "sor",
    } }

# Energy Equation Solver Parameters:
energy_solver_parameters = {
    "mat_type": "aij",
    "snes_type": "ksponly",
    "ksp_type": "gmres",
    "ksp_rtol": 1e-7,
    "pc_type": "sor", }

# -------------------------------------------------------------------------------
# Set up boundary conditions:
bcvfb = DirichletBC(Z.sub(0).sub(1), 0, (front_id, back_id))
bcvlr = DirichletBC(Z.sub(0).sub(0), 0, (left_id, right_id))
bcvbt = DirichletBC(Z.sub(0), 0, (bot_id,top_id))
bctb, bctt = DirichletBC(Q, 1.0, bot_id), DirichletBC(Q, 0.0, top_id)

# -------------------------------------------------------------------------------
# Generating near_nullspaces for GAMG:
x_rotV = Function(V).interpolate(as_vector((0, X[2], -X[1])))
y_rotV = Function(V).interpolate(as_vector((-X[2], 0, X[0])))
z_rotV = Function(V).interpolate(as_vector((-X[1], X[0], 0)))
nns_x = Function(V).interpolate(Constant([1., 0., 0.]))
nns_y = Function(V).interpolate(Constant([0., 1., 0.]))
nns_z = Function(V).interpolate(Constant([0., 0., 1.]))
V_near_nullspace = VectorSpaceBasis([nns_x, nns_y, nns_z, x_rotV, y_rotV, z_rotV])
V_near_nullspace.orthonormalize()
Z_near_nullspace = MixedVectorSpaceBasis(Z, [V_near_nullspace, Z.sub(1)])

# -------------------------------------------------------------------------------
# Updated solve setup:
stokes_problem = NonlinearVariationalProblem(F_stokes, z, bcs=[bcvbt, bcvfb, bcvlr])
stokes_solver = NonlinearVariationalSolver(stokes_problem, solver_parameters=
    stokes_solver_parameters, appctx={"mu": mu}, nullspace=p_nullspace, transpose_nullspace=
    p_nullspace, near_nullspace=Z_near_nullspace)
energy_problem = NonlinearVariationalProblem(F_energy, Tnew, bcs=[bctb, bctt])
energy_solver = NonlinearVariationalSolver(energy_problem, solver_parameters=
    energy_solver_parameters)
```

**Listing 4.** Changes required to reproduce a 3-D Cartesian case from Busse et al. (1994) relative to Listing 1.





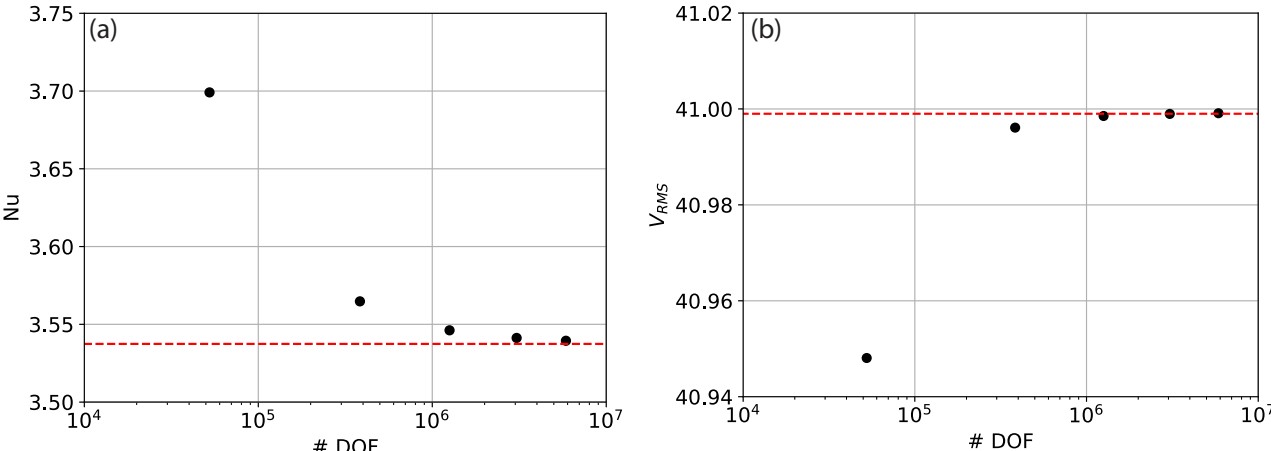

**Figure 5.** Results from 3-D isoviscous simulations in Firedrake, configured to reproduce benchmark results from Case 1a of Busse et al. (1994): (a) Nusselt number vs. number of pressure and velocity degrees of freedom (DOF), at $Ra = 3 \times 10^4$ , for a series of uniform, structured meshes; (b) RMS velocity vs. number of pressure and velocity DOF. Benchmark values are denoted by dashed red lines.

control the aggregation method (coarsening strategy) in the GAMG preconditioner, which balance the multigrid effectiveness (convergence rate) with coarse grid complexity (cost per iteration) (Balay et al., 2021a).

The `fieldsplit_1` entries contain solver options for the Schur complement solve itself. As explained in Section 4.3 we do not have explicit access to the Schur complement matrix, $G^T K^{-1} G$, but can compute its action on any vector, at the cost of a `fieldsplit_0` solve with the K matrix, which is sufficient to solve the system using a Krylov method. However, for preconditioning, we do need access to the values of the matrix or its approximation. For this purpose we approximate the Schur complement matrix with a mass matrix scaled by viscosity, which is implemented in `MassInvPC` (line 35) with the viscosity provided through the optional `appctx` argument on line 71. This is a simple example of Firedrake's powerful programmable preconditioner interface which, in turn, connects with the Python preconditioner interface of PETSc (line 34). In more complex cases the user can specify their own linear operator in UFL that approximates the true linear operator but is easier to invert. The `MassInvPC` preconditioner step itself is performed through a linear solve with the approximate matrix with options prefixed with `Mp_` to specify a Conjugate Gradient solver with symmetric SOR (SSOR) preconditioning (lines 36-38). Note that PETSc's `sor` preconditioner type, specified on line 38, defaults to the symmetric SOR variant. Since this preconditioner step now involves an iterative solve, the Krylov method used for the Schur complement needs to be of flexible type, and we specify flexible GMRES (`fgmres`) on line 32.

Specification of the matrix type `matfree` (line 16) for the combined system ensures that we do not explicitly assemble its associated sparse matrix, instead computing the matrix-vector multiplications required by the Krylov iterations as they arise. Again, for preconditioning in the K-matrix solve we need access to matrix values, which is achieved using `AssembledPC`. This explicitly assembles the K-matrix by extracting relevant terms from the `F_Stokes` form.





Finally, the energy solve is performed through a combination of the GMRES (`gmres`) Krylov method and SSOR precon-
ditioning (lines 42-47). For all iterative solves we specify a convergence criterion based on the relative reduction of the
preconditioned residual (`ksp_rtol`: lines 24, 33, 36 and 46).

    4. Velocity boundary conditions, which must be specified along all 6 faces, are modified on lines 51-53, with temperature
boundary conditions specified on line 54.

5. Generating near-nullspace information for the GAMG preconditioner (lines 58-66), consisting of three rotational (x_rotV,
y_rotV, z_rotV) and three translational (nns_x, nns_y, nns_z) modes, as outlined in Section 4.3. These are combined in
the mixed function space on line 66.

    6. Updating of the Stokes problem (line 70) to account for additional boundary conditions, and the Stokes solver (line 71)
to include the near nullspace options defined above, in addition to the optional `appctx` keyword argument that passes the
viscosity through to our `MassInvPC` Schur complement preconditioner. Energy solver options are also updated relative to
our base case (lines 72-73), using the dictionary created on lines 42-47.

    Our model results can be validated against those of Busse et al. (1994). As with our previous examples, we compute the
Nusselt number and RMS velocity at a range of different mesh resolutions, with results presented in Figure 5. We find that
results converge towards the benchmark solutions, with increasing resolution, as expected. The final steady state temperature
field is illustrated in Figure 2(b).

### 5.3.2    2-D Cylindrical Shell Domain

We next examine simulations in a 2-D cylindrical domain. The domain is defined by the radii of the inner ($r_{\min}$) and outer ($r_{\max}$)
boundaries. These are chosen such that the non–dimensional depth of the mantle, $z = r_{\max} - r_{\min} = 1$, and the ratio of the inner
and outer radii, $f = r_{\min}/r_{\max} = 0.55$, thus approximating the ratio between the radii of Earth's surface and core-mantle-
boundary (CMB). Specifically, we set $r_{\min} = 1.22$ and $r_{\max} = 2.22$. The initial temperature distribution, chosen to produce 4
equidistant plumes, is prescribed as:

$$T(x,y) = (r_{\max} - r) + A\cos(4 \ \mathrm{atan2}(y,x))\sin(r - r_{\min})\pi) \tag{38}$$

where $A = 0.02$ is the amplitude of the initial perturbation. Boundary conditions for temperature are $T = 0$ at the surface ($r_{\max}$)
and $T = 1$ at the base ($r_{\min}$). Free-slip velocity boundary conditions are specified on both boundaries, which we incorporate
weakly through the Nitsche approximation (see Section 4.1). The Rayleigh number $Ra = 1 \times 10^5$.

    With a free-slip boundary condition on both boundaries, one can add an arbitrary rotation of the form $(-y,x) = r\hat{\theta}$ to the
velocity solution (i.e. this case incorporates a velocity nullspace, as well as a pressure nullspace). As noted in Section 4,
these lead to null-modes (eigenvectors) for the linear system, rendering the resulting matrix singular. In preconditioned Krylov
methods these null-modes must be subtracted from the approximate solution at every iteration (Kramer et al., 2021a), which
we illustrate through this example. The key changes required to simulate this case, displayed in Listing 5, are:





1. Mesh generation: we generate a circular manifold mesh (with 256 elements in this example) and extrude in the radial direction, using the optional keyword argument `extrusion_type`, forming 64 layers (lines 2-5). To better represent the curvature of the domain and ensure accuracy of our quadratic representation of velocity, we approximate the curved cylindrical domain quadratically, using the optional keyword argument `degree= 2` (see Section 4 for further detail).

2. The unit vector, $k$, points radially, in the direction opposite to gravity, as defined on line 11. The temperature field is initialised using Equation (38) on line 12.

3. Boundary conditions are no longer aligned with Cartesian directions. We use the Nitsche method (see Section 4.1) to impose our free-slip boundary conditions weakly (lines 15-27). The fudge factor in the interior penalty term is set to 100 on line 16, with Nitsche-related contributions to the UFL added on lines 24-27. Note that for extruded meshes in
Firedrake, `ds_tb` denotes an integral over both the top and bottom surfaces of the mesh (`ds_t` and `ds_b` denote integrals

```
# Mesh Generation:
rmin, rmax, ncells, nlayers = 1.22, 2.22, 256, 64
mesh1d = CircleManifoldMesh(ncells, radius=rmin, degree=2)
mesh = ExtrudedMesh(mesh1d, layers=nlayers, extrusion_type="radial")
bottom_id, top_id = "bottom", "top"
------------------------------------------------------------------------
# Constants, unit vector, initial condition
Ra = Constant(1e5)
r = sqrt(X[0]**2 + X[1]**2)
k = as_vector((X[0], X[1])) / r
Told.interpolate(rmax-r + 0.02*cos(4.*atan_2(X[1],X[0]))*sin((r-rmin)*pi))
------------------------------------------------------------------------
# UFL for Stokes equations incorporating Nitsche:
C_ip = Constant(100.0) # Fudge factor for interior penalty term used in weak imposition of BCs
p_ip = 2 # Maximum polynomial degree of the _gradient_ of velocity
# Stokes equations in UFL form:
stress = 2 * mu * sym(grad(u))
F_stokes = inner(grad(v), stress) * dx + dot(v, grad(p)) * dx - (dot(v, k) * Ra * Ttheta) * dx
F_stokes += dot(grad(w), u) * dx  # Continuity equation
# nitsche free-slip BCs
F_stokes += -dot(v, n) * dot(dot(n, stress), n) * ds_tb
F_stokes += -dot(u, n) * dot(dot(n, 2 * mu * sym(grad(v))), n) * ds_tb
F_stokes += C_ip * mu * (p_ip + 1)**2 * FacetArea(mesh) / CellVolume(mesh) * dot(u, n) * dot(v, n)
        * ds_tb
------------------------------------------------------------------------
# Nullspaces and near-nullspaces:
x_rotV = Function(V).interpolate(as_vector((-X[1], X[0])))
V_nullspace = VectorSpaceBasis([x_rotV])
V_nullspace.orthonormalize()
p_nullspace = VectorSpaceBasis(constant=True) # Constant nullspace for pressure n
Z_nullspace = MixedVectorSpaceBasis(Z, [V_nullspace, p_nullspace]) # Setting mixed nullspace
# Generating near_nullspaces for GAMG:
nns_x  = Function(V).interpolate(Constant([1., 0.]))
nns_y  = Function(V).interpolate(Constant([0., 1.]))
V_near_nullspace = VectorSpaceBasis([nns_x, nns_y, x_rotV])
V_near_nullspace.orthonormalize()
Z_near_nullspace = MixedVectorSpaceBasis(Z, [V_near_nullspace, Z.sub(1)])
------------------------------------------------------------------------
# Updated solve calls:
stokes_problem = NonlinearVariationalProblem(F_stokes, z) # velocity BC's handled through Nitsche
stokes_solver = NonlinearVariationalSolver(stokes_problem, solver_parameters=
        stokes_solver_parameters, appctx={"mu": mu}, nullspace=Z_nullspace, transpose_nullspace=
        Z_nullspace, near_nullspace=Z_near_nullspace)
```

**Listing 5.** Difference in Firedrake code required to reproduce isoviscous case in a 2-D cylindrical domain.

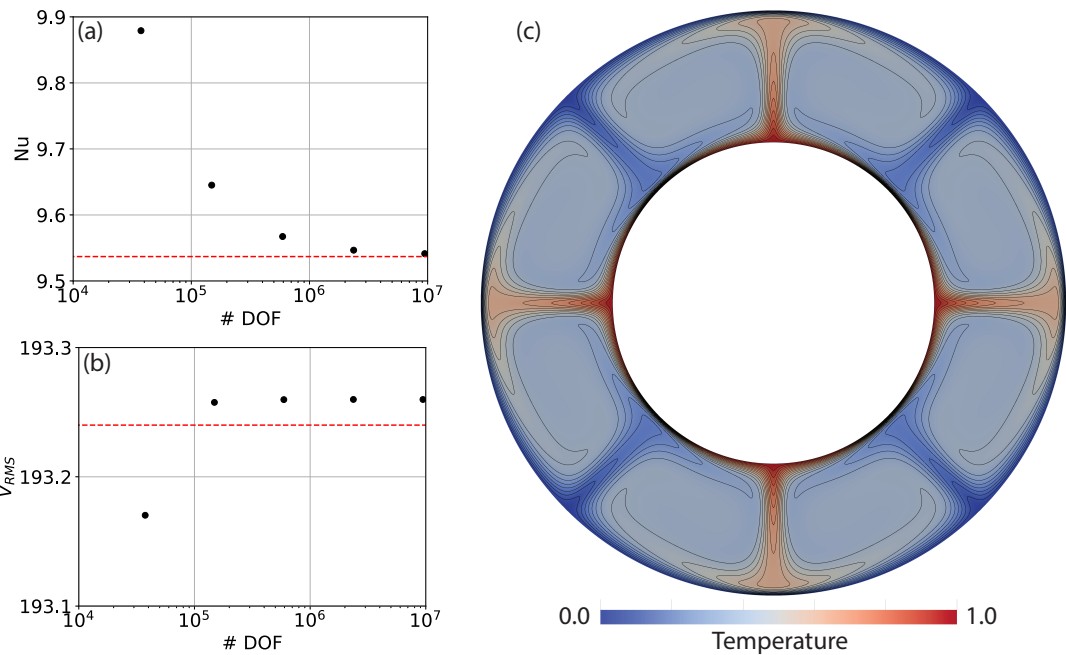

**Figure 6.** (a)/(b) Nusselt number/RMS velocity vs. number of pressure and velocity DOF, at $Ra = 1 \times 10^5$, for a series of uniform, structured meshes in a 2-D cylindrical domain. High-resolution, adaptive mesh, results from the Fluidity computational modelling framework are delineated by dashed red lines; (c) final steady-state temperature field, with contours spanning temperatures of 0 to 1, at intervals of 0.05.

over the top or bottom surface of the mesh, respectively). `FacetArea` and `CellVolume` return, respectively, $A_f$ and $V_{cf}$ required by Equation 17. Given that velocity boundary conditions are handled weakly through UFL, they are no longer passed to the Stokes problem as a separate option (line 46).

4. We define the rotational nullspace for velocity and combine this with the pressure nullspace in the mixed finite element
620    space $Z$ (lines 30-35). Constant and rotational near-nullspaces, utilised by our GAMG preconditioner, are also defined on lines 37-42, with this information passed to the solver on line 47. Note that iterative solver parameters identical to those presented in the previous example are used (see Section 5.3.1).

Our predicted Nusselt numbers converge towards those of Fluidity with increasing resolution (Figure 6), demonstrating the accuracy of our approach. Predicted RMS velocities exceed those of Fluidity, albeit only by $\sim 0.1\%$, but lie within the bounds
625    set by other codes for this case (Wilson, Pers. Comm., using TerraFERMA: Wilson et al., 2017). To further assess the validity of our setup, we have confirmed the accuracy of our solutions to the Stokes system in this 2-D cylindrical geometry, through comparisons with analytical solutions from Kramer et al. (2021a), for both zero-slip and free-slip boundary conditions. These provide a suite of solutions based upon a smooth forcing term, at a range of wave-numbers $n$, with radial dependence formed by a polynomial of arbitrary order $k$. We study the convergence of our Q2-Q1 discretisation with respect to these solutions.



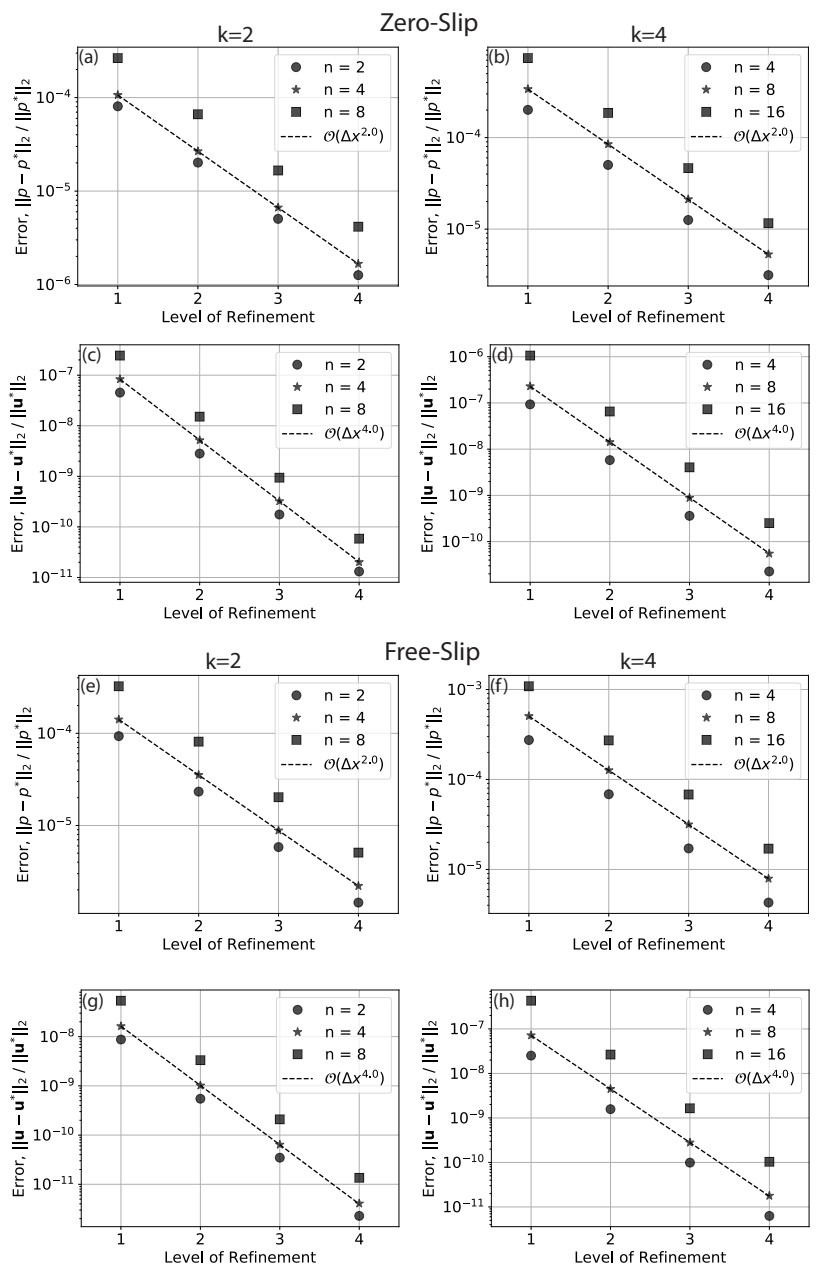

**Figure 7.** Convergence for 2-D cylindrical cases with zero-slip (a-d) and free-slip (e-h) boundary conditions, driven by smooth forcing at a series of different wave-numbers, $n$, and different polynomial orders of the radial dependence, $k$, as indicated in the legend (see Kramer et al., 2021a, for further details). Convergence rate is indicated by dashed lines, with the order of convergence provided in the legend. For the cases plotted, the series of meshes start at refinement level 1, where the mesh consists of 1024 divisions in the tangential direction and 64 radial layers. At each subsequent level the mesh is refined by doubling resolution in both directions.



```
# Mesh Generation:
rmin, rmax, ref_level, nlayers  = 1.22, 2.22, 4, 16
mesh2d = CubedSphereMesh(rmin, refinement_level=ref_level, degree=2)
mesh = ExtrudedMesh(mesh2d, layers=nlayers, extrusion_type='radial')
--------------------------------------------------------------------------------
# Nullspaces and near-nullspaces:
x_rotV = Function(V).interpolate(as_vector((0, X[2], -X[1])))
y_rotV = Function(V).interpolate(as_vector((-X[2], 0, X[0])))
z_rotV = Function(V).interpolate(as_vector((-X[1], X[0], 0)))
V_nullspace = VectorSpaceBasis([x_rotV, y_rotV, z_rotV])
V_nullspace.orthonormalize()
p_nullspace = VectorSpaceBasis(constant=True) # Constant nullspace for pressure
Z_nullspace = MixedVectorSpaceBasis(Z, [V_nullspace, p_nullspace]) # Setting mixed nullspace
nns_x = Function(V).interpolate(Constant([1., 0., 0.]))
nns_y = Function(V).interpolate(Constant([0., 1., 0.]))
nns_z = Function(V).interpolate(Constant([0., 0., 1.]))
V_near_nullspace = VectorSpaceBasis([nns_x, nns_y, nns_z, x_rotV, y_rotV, z_rotV])
V_near_nullspace.orthonormalize()
Z_near_nullspace = MixedVectorSpaceBasis(Z, [V_near_nullspace, Z.sub(1)])
```

**Listing 6.** Difference in Firedrake code required to reproduce 3-D spherical benchmark cases from Zhong et al. (2008).

Convergence plots are illustrated in Figure 7. We observe super-convergence for the Q2-Q1 element pair at fourth- and second-order, for velocity and pressure, respectively, with both zero-slip and free-slip boundary conditions, which is higher than the theoretical (minimum) expected order of convergence of three for velocity and two for pressure (we note that super-convergence was also observed in Zhong et al., 2008; Kramer et al., 2021a). Cases with lower wave-number, $n$, show smaller relative error than those at higher $n$, as expected. The same observation holds for lower and higher polynomial order, $k = 2$ and $k = 4$, for

the radial density profile. Python scripts for these analytical comparisons can be found in the repository accompanying this paper.

### 5.3.3   3-D Spherical Shell Domain

We next move into a 3-D spherical shell geometry, which is required to simulate global mantle convection. We examine a well-known isoviscous community benchmark case (e.g. Bercovici et al., 1989; Ratcliff et al., 1996; Zhong et al., 2008; Davies et al.,

2013), at a Rayleigh number of $Ra = 7 \times 10^3$, with free-slip velocity boundary conditions. Temperature boundary conditions are set to 1 at the base of the domain ($r_{min} = 1.22$) and 0 at the surface ($r_{max} = 2.22$), with the initial temperature distribution approximating a conductive profile with superimposed perturbations triggering tetrahedral symmetry at spherical harmonic degree $l = 3$ and order $m = 2$ (see Zhong et al., 2008, for further details).

As illustrated in Listing 6, when compared to the 2-D cylindrical case examined in Section 5.3.2, the most notable change

required to simulate this 3-D case is the generation of the underlying mesh. We use Firedrake's built-in CubedSphereMesh and extrude it radially through 16 layers, forming hexahedral elements. As with our cylindrical example, we approximate the curved cylindrical domain quadratically, using the optional keyword argument degree= 2. Further required changes, highlighted in Listing 6, relate to 3-D extensions of the velocity nullspace, and the near-nullspaces required by the GAMG preconditioner, all of which are simple. We do not show the changes associated with extending the radial unit vector to 3-D, or the initial condition

for temperature, given that they are straightforward, although, as with all examples, a complete Python script for this case can be found in the repository accompanying this paper.



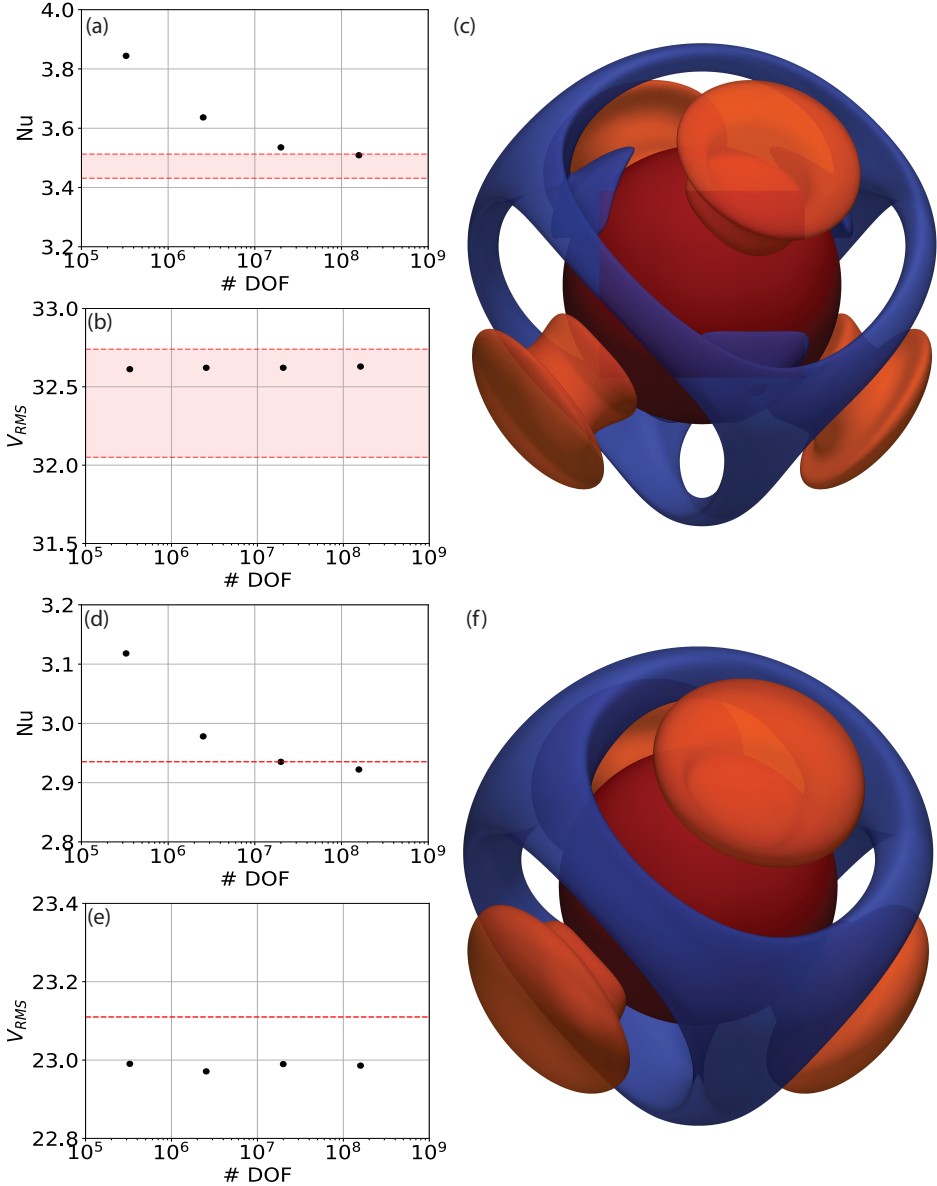

**Figure 8.** (a)/(b) Nusselt number/RMS velocity vs. number of pressure and velocity DOF, designed to match an isoviscous 3-D spherical benchmark case at $Ra = 7 \times 10^3$, for a series of uniform, structured meshes. The range of solutions predicted in previous studies are bounded by dashed red lines (Bercovici et al., 1989; Ratcliff et al., 1996; Yoshida and Kageyama, 2004; Stemmer et al., 2006; Choblet et al., 2007; Tackley, 2008; Zhong et al., 2008; Davies et al., 2013); (c) final steady-state temperature field highlighted through isosurfaces at temperature anomalies (i.e. away from radial average) of $T = -0.15$ (blue) and $T = 0.15$ (orange), with the core-mantle-boundary at the base of the spherical shell marked by a red surface; (d-f) as in a-c, but for a temperature-dependent-viscosity case, with thermally induced viscosity contrasts of $10^2$. Fewer codes have published predictions for this case, but results of Zhong et al. (2008) are marked by dashed red lines.



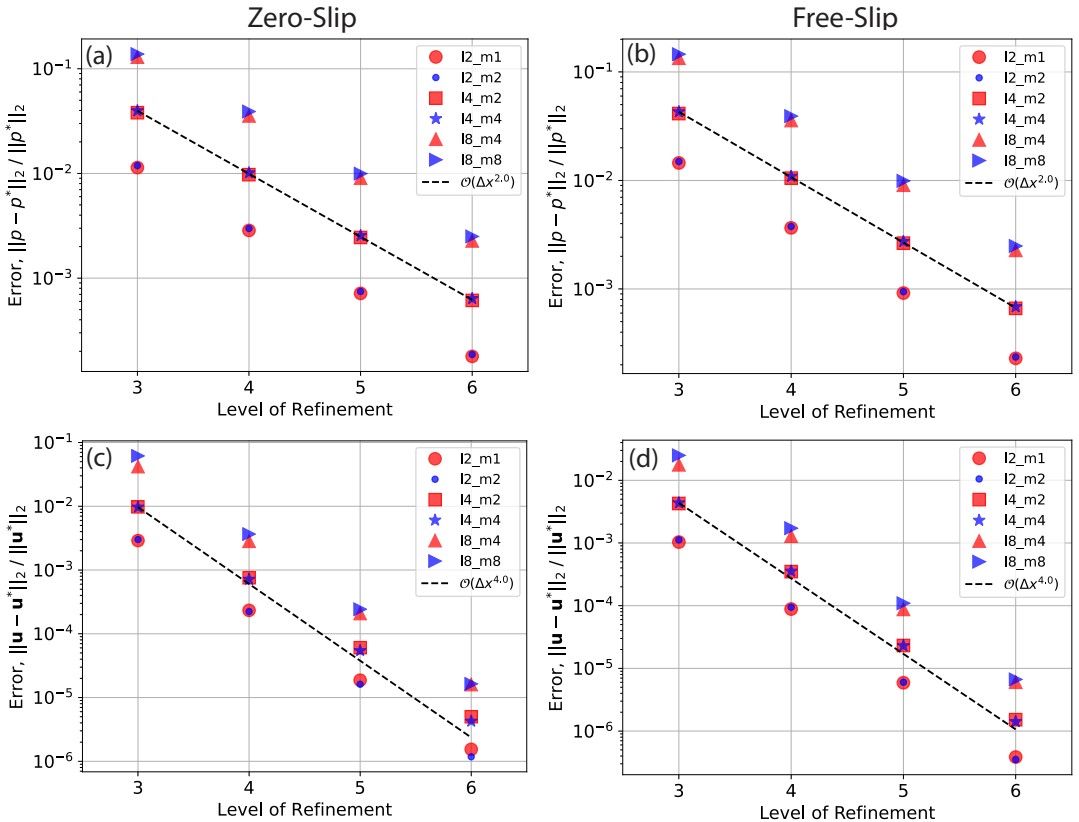

**Figure 9.** Convergence of velocity and pressure for 3-D spherical cases with zero-slip and free-slip boundary conditions, for perturbations at a range of spherical harmonic degrees $l$ and orders $m$. Note that all cases with a smooth forcing are run at $k = l + 1$. Refinement level 3 corresponds to the level specified for our cubed sphere mesh, comprising 386 elements in the tangential direction, which is extruded radially to 8 layers. Resolution is doubled in all directions at subsequent refinement levels.

Despite the simplicity of our setup, the accuracy of our approach is confirmed via comparison of both Nusselt numbers and RMS velocities with those of previous studies (e.g. Bercovici et al., 1989; Ratcliff et al., 1996; Yoshida and Kageyama, 2004; Stemmer et al., 2006; Choblet et al., 2007; Tackley, 2008; Zhong et al., 2008; Davies et al., 2013). For completeness,

the final steady-state temperature field is illustrated in Figure 8(c). Furthermore, in line with our 2-D cases, we have confirmed the accuracy of our Stokes solver for both zero-slip and free-slip boundary conditions in a 3-D spherical geometry, through comparisons with analytical solutions from Kramer et al. (2021a), which provide solutions based upon a smooth forcing term at a range of spherical harmonic degrees, $l$, and orders, $m$, with radial dependence formed by a polynomial of arbitrary order $k$. As with our 2-D cases, we observe super-convergence for the Q2-Q1 element pair at fourth- and second-order, for velocity

and pressure, respectively, with both zero-slip and free-slip boundary conditions (Figure 9).

This section has allowed us to highlight a number of Firedrake's benefits over other codes: (i) the ease at which simulations can be examined in different geometries, with minimal changes to the Python code, facilitated by Firedrake's built-in mesh





generation utilities and extrusion functionality; (ii) the ease at which iterative solver configurations and preconditioners can
be updated and tested, including scenarios incorporating multiple nullspaces, facilitated by Firedrake's fully-programmable
solver interface, alongside its customisable preconditioner interface, both of which are seamlessly coupled to PETSc; and (iii)
the convergence properties of our finite element system, in geometries that are representative of Earth's mantle. Taken together,
these confirm Firedrake's suitability for simulations of global mantle dynamics, as will be further highlighted in Section 7.

## 6  Parallel Scaling

We assess parallel scalability using a 3-D spherical case similar to that presented in Section 5.3.3, albeit incorporating a
temperature-dependent viscosity, following the relation:

$$\mu = \exp[E(0.5 - T)], \tag{39}$$

where E is a parameter that controls the temperature dependence of viscosity. In the example considered — Case A4 from
Zhong et al. (2008) — we set $E = \ln(100)$, leading to thermally induced viscosity contrasts of $10^2$ across the computational
domain. For completeness, our steady-state results, highlighting the consistency of our results for this case with the predictions
of Zhong et al. (2008), are displayed in Figure 8, although for the purposes of parallel scaling analyses, we run simulations for
20 time-steps only.

We focus on weak scaling, where the problem size and the number of processing cores are simultaneously increased. Cases
are examined on 24, 192, 1536 and 12288 cores, maintaining 4096 elements per core and ensuring a constant element aspect
ratio across all resolutions examined. Simulations were examined on the Gadi supercomputer at the National Computational
Infrastructure (NCI) in Australia, using compute nodes with $2 \times 24$ core Intel Xeon Platinum 8274 (Cascade Lake) 3.2 GHz
CPUs, and 192 GB RAM, per node. Linking the nodes is the latest generation HDR InfiniBand technology in a Dragonfly+
topology, capable of transferring data at up to 200 Gb/s.

The most challenging aspect of weak parallel scaling is solver performance as the problem size increases. Whilst the amount
of computation in equation assembly typically scales linearly with the number of DOFs – before taking parallel aspects such
as communication into account – solver scaling is generally worse. In the case of iterative solvers, this is due to a deterioration
in the conditioning of the matrix, driving an increase in the number of iterations required for convergence. As a result, even if
the cost per iteration scales linearly, the overall cost will not. This implies that for weak scaling, the amount of work per core
may increase rapidly, despite the number of DOFs per core remaining consistent.

The deterioration in conditioning is intimately related to the fact that an increase in resolution increases the ratio between
smallest and largest resolvable length-scales. For elliptic operators, like the viscosity matrix K, the condition number scales
with the square of that ratio (e.g. Kramer et al., 2010). Multigrid approaches, which separate smaller and larger length scales on
a hierarchy of fine to coarse meshes, are commonly used to address this problem, which motivates the choice of the algebraic
multigrid preconditioner, GAMG, used here. Such approaches aim to maintain a constant, or only slowly increasing number of
iterations and, thus, a near-linear scaling of the overall cost, as the problem size increases. This can be a challenge however as,



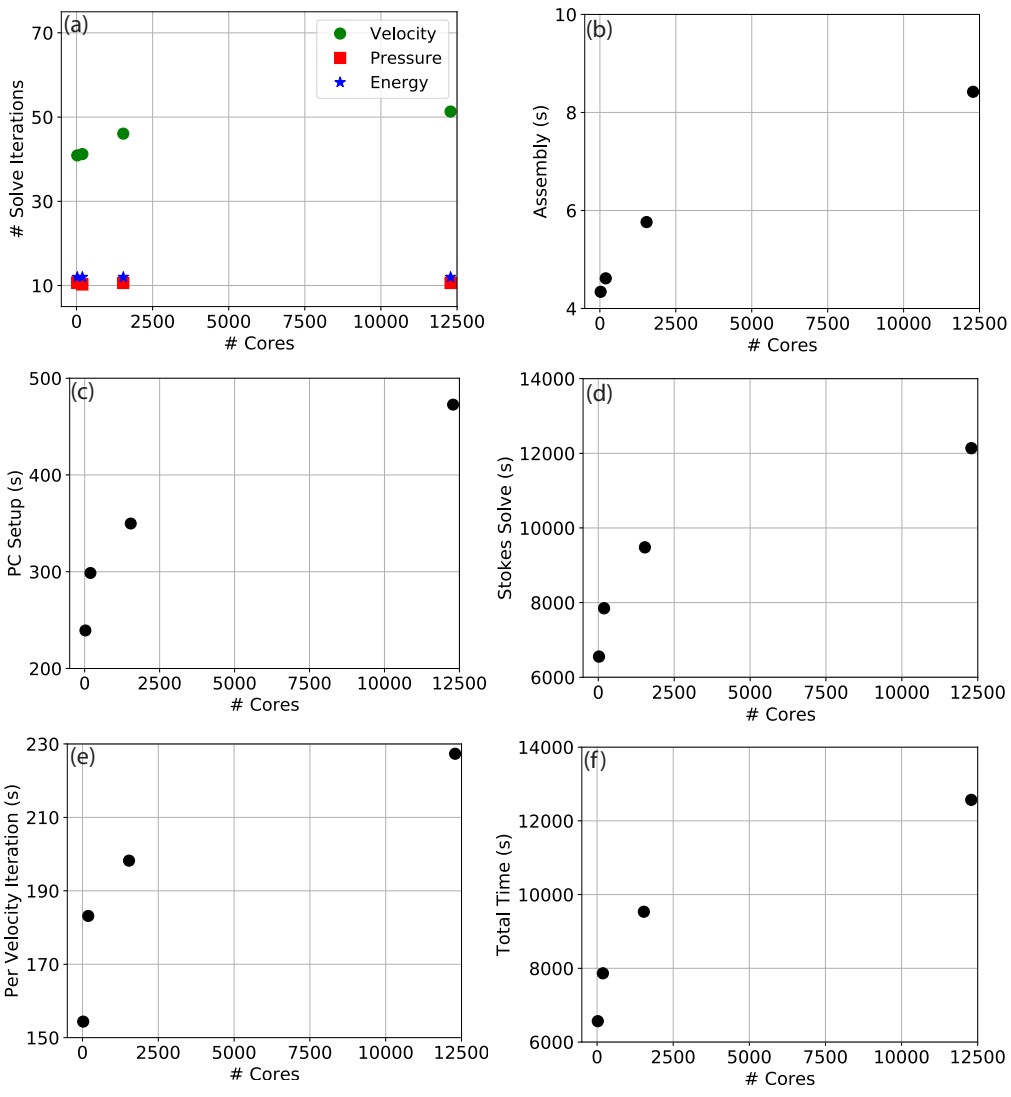

**Figure 10.** Weak scaling analyses for a 20 time-step, temperature-dependent viscosity, spherical shell simulation with free-slip boundary conditions: (a) mean number of iterations per time-step, for energy (blue), pressure (red) and velocity (green) solves, respectively; (b) time spent in assembly of finite element systems; (c) time spent setting up algebraic multgrid preconditioner; (d) time spent solving the Schur complement (Stokes) system; (e) cost per velocity solve iterations; (f) total simulation time, which closely mimics the Schur complement time.

for instance, an increase in resolution will require more multigrid levels, which will lead to an increased setup time and cost per iteration. In practice, when configuring the multigrid method, a compromise needs to be found between the effectiveness of multigrid in limiting the number of iterations, and not allowing the setup and costs per iteration to grow too rapidly. The





two options, `gamg_threshold` and `gamg_square_graph`, specified in our solver setup, ensure a balance between multigrid effectiveness an coarse grid complexity.

A breakdown of key parallel scaling results are presented in Figure 10. Panel a displays the average number of iterations per solve over the 20 timesteps. We find that the number of pressure (the Schur complement solve: `fieldsplit_1`) and energy solve iterations remains flat (12 and $\sim 10.5$, respectively), whilst the number of velocity solve iterations (inversion of the matrix K, using the GAMG preconditioner: `fieldsplit_0`) increases only slowly, from $\sim 41$ to $\sim 51$, over a greater than three-orders-of-magnitude increase in problem size and number of processor cores. This demonstrates algorithmic scalability on up to 12288 cores and $\sim 50 \times 10^6$ elements (which corresponds to $\sim 1.25 \times 10^9$ velocity and pressure degrees of freedom).

Parallel scalability can also be assessed by analysing the growth in CPU-time of the dominant components of our problem: assembly of finite element systems (Figure 10b), setup of the algebraic multigrid (GAMG) preconditioner (Figure 10c), and time spent solving the Schur complement system (Figure 10d). We find that the assembly time is a negligible fraction of this problem. The setup time for our GAMG preconditioner grows from $\sim 240$s on 24 cores to $\sim 470$s on 12288 cores. This is understandable, given the large communication costs associated with setting up various multigrid levels, particularly for problems incorporating nullspaces and near-nullspaces, as is the case here. We note, however, that this is not a concern: as a fraction of the entire solution time for the Schur complement solve (Figure 10d), GAMG setup remains small. We do observe an increase in time required for solution of the Schur Complement (Stokes solve), from $\sim 6500$s on 24 cores to $\sim 12100$s on 12288 cores. This results primarily from the minor increase in the number of velocity solve iterations and the increased cost per iteration (Figure 10e), which rises from 155s on 24 cores to 225s on 12288 cores, reflecting costs associated with increasing the number of multigrid levels for higher-resolution problems. The total time spent in running this problem mirrors the time spent in solving the Schur complement system (Figure 10f), indicating where future optimisation efforts should be directed. We note that the change in gradient, apparent in panels b-f when moving from 24-192 and 192-12288 cores arises due to a transition from running simulations on a single compute node to multiple nodes.

## 7 Realistic Application in 3-D spherical geometry: Global Mantle Convection

In this section, we demonstrate application of Firedrake to a time-dependent simulation of global mantle convection in a 3-D spherical shell geometry, at a realistic Rayleigh number. As with the examples provided above, calculations are performed using a hexahedral trilinear Q2-Q1 element pair for velocity and pressure. We use a Q2 discretisation for temperature and, given the increased importance of advection at higher Rayleigh numbers, incorporate stabilisation through a streamline upwinding scheme, following Donea and Huerta (2003). Our solution strategy for the Stokes and energy equations is otherwise identical to the spherical examples presented above.

For simplicity, we assume an incompressible mantle and a linear temperature- and depth-dependent rheology, following the relation,

$$\mu = \mu_0 \exp\left[E(0.5 - T)\right]. \tag{40}$$





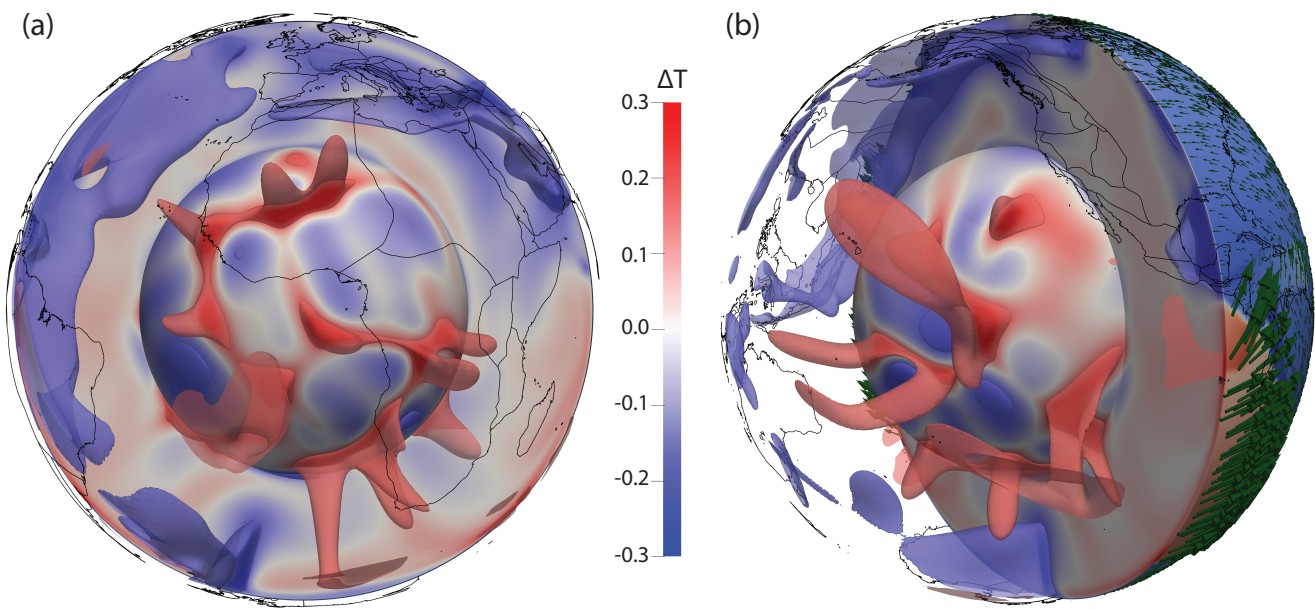

**Figure 11.** Present-day thermal structure, predicted from our global mantle convection simulation where the geographic distribution of heterogeneity is dictated by 230 Myr of imposed plate motion history (Muller et al., 2016). Each image includes a radial surface at $r = 1.25$ (i.e. immediately above the core-mantle boundary), a cross-section, and transparent isosurfaces at temperature anomalies (i.e. away from the radial average) of $T = -0.15$ (blue) and $T = 0.15$ (red), highlighting the location of downwelling slabs and upwelling mantle plumes (below $r = 2.19$), respectively. Continental boundaries provide geographic reference. Panel a provides an Africa-centered view, with panel b centered on the Pacific Ocean, and including glyphs at the surface highlighting the imposed plate velocities.

Here $\mu_0$ is a reference viscosity that increases by a factor of 40 below the mantle transition zone, and $E = \ln(1000)$ controls the sensitivity of viscosity to temperature. We specify a reference (basally heated) Rayleigh number of $2 \times 10^7$, which is comparable to estimates of Earth's mantle (e.g. Davies, 1999), and also include internal heating at a non-dimensional heating rate of 10. The simulation is spun-up with free-slip and isothermal ($T = 0$ at base; $T = 1$ at top) boundaries at both surfaces. After the model reaches a quasi-steady state (i.e. when the surface and basal Nusselt numbers both change by less than 0.1%

over 10 consecutive time-steps), surface velocities are assimilated through a kinematic boundary condition, according to 230 Myr of plate motion histories (Muller et al., 2016), using the Python interface to GPlates (e.g. Gurnis et al., 2012; Muller et al., 2018). Our simulation then runs forward towards the present-day. This case is therefore analogous to the simulations examined when addressing questions from the very frontiers of geodynamical research (e.g. Schuberth et al., 2009; Davies and Davies, 2009; Davies et al., 2012; Bower et al., 2013; Hassan et al., 2015; Nerlich et al., 2016; Rubey et al., 2017; Koelemeijer et al.,

2018; Ghelichkhan et al., 2018).

     Our results are illustrated in Figure 11. We find that the present-day upper mantle convective planform is dominated by strong downwellings in regions of plate convergence. In the mid-mantle, cold downwellings are prominent beneath North America and South-East Asia, whilst remnants of older subduction are visible above the core-mantle-boundary. The location





of hot upwelling material is strongly modulated by these downwellings, with upwelling plumes concentrating in two clusters

beneath the African continent and the central Pacific ocean (i.e. away from regions that have experienced subduction over the past 150 Myr or so). The cluster of plumes in the Pacific is reasonably circular, whilst those beneath Africa extend in a NW-SE trending structure, which to the north curves eastward under Europe and to the south extends into the Indian Ocean.

Further analysis of this proof-of-concept simulation is beyond the scope of this study. However, when combined with the benchmark and parallel scaling analyses presented-above, our model predictions, which are consistent with those from a num-

ber of previous studies (e.g. Bunge et al., 2002; Davies et al., 2012; Bower et al., 2013; Davies et al., 2015a), confirm Firedrake's applicability for realistic, time-dependent, global mantle dynamics simulations of this nature.

## 8    Discussion

Firedrake is a next-generation automated system for solving variational problems using the finite element method (e.g. Rathgeber et al., 2016; Gibson et al., 2019). It has a number of features that are ideally suited to simulating geophysical fluid dynamics

problems, as exemplified by its use in application areas such as coastal ocean modelling (Kärnä et al., 2018), numerical weather prediction (Shipton et al., 2018), and glacier flow modelling (Shapero et al., 2021). The focus of this manuscript has been to demonstrate Firedrake's applicability for geodynamical simulation, with an emphasis on global mantle dynamics. To do so, we presented, analysed and validated Firedrake against a number of benchmark and analytical cases, of systematically increasing complexity, building towards a realistic time-dependent global simulation.

In order to introduce the core components and illustrate the elegance of setting up and validating a geodynamical model in Firedrake, we started with a simple, incompressible, isoviscous case in an enclosed 2-D Cartesian box. Setting up this problem was straightforward, requiring only a weak formulation of the governing equations for specification in UFL, together with a mesh, initial and boundary conditions, and appropriate discrete function spaces. By utilising Firedrake's built-in meshing functionality and default direct solver options, we were able to demonstrate the framework's accuracy for simulations of this

nature: in less than 70 lines of Python, we reproduced results from the well-established benchmark study of Blankenbach et al. (1989),

Representative simulations of mantle and lithosphere dynamics, however, incorporate more complicated physics. To demonstrate Firedrake's applicability in such scenarios, we next set up 2-D simulations that accounted for compressibility, through the Anelastic Liquid Approximation (Schubert et al., 2001), and a nonlinear viscosity that depends upon temperature, depth

and strain-rate. Our results were validated through comparison with the benchmark studies of King et al. (2009) and Tosi et al. (2015), respectively. For compressible cases, despite the governing equations differing appreciably from their incompressible counterparts, the modifications required to our setup were minimal, with the most notable change being the UFL describing the relevant PDEs. For the viscoplastic rheology case, where viscosity varied by several orders of magnitude across the domain, an appropriate solution strategy was required to deal with nonlinear coupling between strain-rate and viscosity: Firedrake's

fully-programmable solver interface and seamless coupling to PETSc facilitated the straightforward use of PETSc's Scalable Nonlinear Equation Solvers (SNES) (Kirby and Mitchell, 2018). Taken together, these examples highlight one of Firedrake's





key benefits: by leveraging UFL (Alnes et al., 2014), associated strategies for automatic assembly of finite element systems, and PETSc (Balay et al., 1997, 2021a, b), the framework is easily extensible, allowing for straightforward application to problems involving different physical approximations, even when they require distinct solution strategies.

This is further highlighted with the transition from 2-D to 3-D. With modifications to only a few lines of Python, the basic 2-D Cartesian case described above was easily extended to 3-D, allowing for comparison and validation against the well-established benchmark results of Busse et al. (1994). However, the direct solvers used for our 2-D cases quickly become computationally intractable in 3-D, necessitating the use of an iterative approach. Firedrake's programmable solver interface facilitates the straightforward inclusion of Python dictionaries that define iterative solver parameters for the Stokes and energy systems. A

number of different schemes have been advocated by the geodynamical modelling community (e.g. May and Moresi, 2008; Burstedde et al., 2013), but in all 3-D simulations examined herein, the Schur complement approach was utilised for solution of our Stokes system, exploiting the fieldsplit preconditioner type to apply preconditioners, including algebraic multigrid, to different blocks of the system. A Crank-Nicholson scheme was utilised for temporal discretisation of the energy equation, with a standard GMRES Krylov method with SOR preconditioning used for solution. We have demonstrated that such solution

strategies are effective and scalable, with algorithmic scalability confirmed on up to 12288 cores.

    Cartesian simulations offer a means to better understand the physical mechanisms controlling mantle convection, but a 3-D spherical shell geometry is required to simulate global mantle dynamics. We have demonstrated how Firedrake's built-in meshing and extrusion functionality facilitates the effortless transition to such geometries (in addition to comparable 2-D cylindrical shell geometries), whilst its Python user-interface allows for the simple inclusion of a radial gravity direction and boundary

conditions that are not aligned with Cartesian directions. The convergence properties and accuracy of our simulations in a 3-D spherical geometry have been demonstrated through comparison with the extensive set of analytical solutions introduced by Kramer et al. (2021a) and a series of low Rayleigh number isoviscous and temperature-dependent viscosity simulations, from Zhong et al. (2008). We observed super-convergence for the Q2-Q1 element pair at fourth- and second-order, for velocity and pressure, respectively.

Having validated Firedrake against this broad suite of cases, we finally applied the framework to a realistic simulation of global mantle convection. For simplicity, we assumed an incompressible mantle and a linear temperature- and depth-dependent rheology, assimilating 230 Myr of plate motion histories (Muller et al., 2016) through a kinematic surface boundary condition. These prescribed plate velocities organize underlying mantle flow, such that the predicted present-day convective planform is dominated by cold downwellings in regions of plate convergence, with upwellings concentrating elsewhere, particularly

beneath the African and Pacific domains. Our model predictions, which reproduce first-order characteristics of the structure of Earth's mantle imaged through seismology (e.g. Ritsema et al.; French and Romanowicz, 2015), the geographical distribution of mantle plumes (e.g. Austermann et al., 2014; Davies et al., 2015b), and are consistent with those from a number of previous studies and the (e.g. Bunge et al., 2002; Davies et al., 2012; Bower et al., 2013; Davies et al., 2015a), serve as a proof-of-concept, confirming Firedrake's applicability for realistic, time-dependent, global simulations of this nature and, accordingly,

its suitability for addressing research problems from the very frontiers of geodynamical research.





Despite this, several components of Firedrake have not been fully examined in this paper. Many of these will likely be useful for geodynamical simulation and, accordingly, will be examined in the future. These include:

1. A range of finite elements: in all examples considered herein, we utilised a continuous Q2-Q1 element pair for velocity and pressure with a Q2 discretisation for temperature (with the exception of one set of examples in Section 5.1, where we demonstrated the use of a Q1 temperature discretisation). Accordingly, we have not demonstrated Firedrake's support for a wide-range of finite elements, including continuous, discontinuous, H(div) and H(curl) discretisations, and elements with continuous derivatives such as the Argyris and Bell elements (see Kirby and Mitchell, 2019, for an overview). Some of these could offer major advantages for geodynamical simulation. For example, a number of studies now advocate the use of Discontinuous Galerkin (DG) schemes for solution of the energy equation (e.g. Vynnytska et al., 2013; He et al., 2017). Importantly, Firedrake's simple API allows a user to escape the UFL abstraction, and implement common operations that fall outside of pure variational formulations, such as flux limiters, which are central to DG schemes. Firedrake also provides the necessary infrastructure for hybridisation strategies (Gibson et al., 2019), which allow for a reduction of the many extra degrees of freedom introduced by DG schemes in the global system to a smaller subset, defined on element interfaces through so-called trace elements. This offers the prospect of arriving at more efficient ways of solving the Stokes system (e.g. Cockburn and Shi, 2014). Such possibilities will be explored in future work, noting that Firedrake's existing support for these elements will facilitate rapid and efficient testing and validation.

2. Fully coupled nonlinear systems: in all examples considered herein, we solve for velocity and pressure in a separate step to temperature, largely owing to our familiarity with this approach from previous work (e.g. Davies et al., 2011; Kramer et al., 2021a). However, a number of studies advocate solving for these fields simultaneously (e.g. Wilson et al., 2017), particularly for strongly coupled, highly-nonlinear, multi-physics problems. By leveraging UFL, in combination with PETSc's fieldsplit preconditioning approach, future work to configure and test such coupled schemes within Firedrake will be relatively straightforward.

3. Preconditioners: a major benefit of Firedrake for the problems considered herein is access to the wide variety of solution algorithms and preconditioning strategies provided by the PETSc library, which can be flexibly configured through the solver parameters dictionary, allowing one to test and apply different strategies with ease. The development of preconditioners for the Stokes problem is an active area of research (e.g. May and Moresi, 2008; Burstedde et al., 2013; Shih et al., 2021). As noted above, Firedrake supports a powerful programmable preconditioner interface which, in turn, connects with the Python preconditioner interface of PETSc, and allows users to specify their own linear operator in UFL, thus enabling preconditioning techniques with bespoke operator approximations. We note that in addition to the complete range of algebraic solvers offered by PETSc, Firedrake also provides access to multilevel solvers with geometric hierarchies, opening up the possibility of exploring geometric multigrid approaches in the future.

We note that the automated approach underpinning Firedrake has the potential to revolutionize the use of adjoints and other inverse schemes in geodynamics. Adjoint models have made an enormous impact in fields such as meteorology and oceanography. However, despite significant progress (e.g. Bunge et al., 2003; Liu et al., 2008; Li et al., 2017; Colli et al., 2018;





Ghelichkhan and Bunge, 2018; Ghelichkhan et al., 2020), their use in other scientific fields, including geodynamics, has been hampered by the practical difficulty of their derivation and implementation. In contrast to developing a model directly in Fortran or C++, high-level systems, such as Firedrake, allow the developer to express the variational problems to be solved in near-mathematical notation through UFL. As such, these systems have a key advantage: since the mathematical structure of the problem is preserved, they are more amenable to automated analysis and manipulation, which can be exploited to automate

the derivation of adjoints (e.g. Farrell et al., 2013; Mitush et al., 2019) and the generation of the low-level code for the derived models. Exploring the use of such an approach in geodynamics will be an important avenue for future research.

Finally, given that the importance of reproducibility in the computational geosciences is increasingly being recognized, we note that Firedrake integrates with Zenodo and GitHub to provide users with the ability to generate a set of DOIs corresponding to the exact set of Firedrake components used to conduct a particular set of simulations. In providing our input scripts and a

DOI for the version of Firedrake used herein, we ensure traceable provenance of model data, in full compliance with FAIR (Findable, Accessible, Interoperable, Reusable) principles.

## 9   Conclusions

Firedrake is a next-generation system for solving variational problems using the finite element method (e.g. Rathgeber et al., 2016; Gibson et al., 2019). It treats finite element problems as a composition of several abstract processes, using separate and

open-source software components for each. Firedrake's overarching goal is to save users from manually writing low-level code for assembling the systems of equations that discretize their model physics. It is written completely in Python, and exploits automatic code-generation techniques to apply sophisticated performance optimisations.

In this manuscript, we have confirmed Firedrake's applicability for geodynamical simulation, by configuring and validating model predictions against a series of benchmark and analytical cases, of systematically increasing complexity. In all cases,

Firedrake has been shown to be *accurate* and *efficient*, and we have also demonstrated that that it is *flexible* and easily *extensible*: by leveraging UFL and PETSc, it can be effortlessly applied to problems involving different physical approximations (e.g. incompressible and compressible flow; isoviscous and more complex nonlinear rheologies), even if they require distinct solution strategies. We have illustrated how Firedrake's built-in mesh generation utilities and extrusion functionality provide a straightforward mechanism for examining problems in different geometries (2-D and 3-D Cartesian, 2-D cylindrical and

3-D spherical), and how its fully-programmable solver dictionary and customisable preconditioner interface, both of which are seamlessly coupled to PETSc, facilitate straightforward configuration of different solution approaches. Parallel *scalability* has been demonstrated, on up to 12288 compute cores. Finally, using a realistic simulation of global mantle dynamics, where the distribution of heterogeneity is governed by imposed plate motion histories (Muller et al., 2016), we have confirmed Firedrake's suitability for tackling challenges from the very forefront of geodynamical research. We note that all simulation data

presented herein has traceable provenance: in providing our input scripts and a DOI for the exact set of Firedrake components employed, Firedrake facilitates *transparency* and *reproducibility*, in full compliance with FAIR principles.





*Code and data availability.* Minor adjustments to the Firedrake code base required to successfully run the cases in this paper have been merged into the open-source software associated with the Firedrake Project: https://www.firedrakeproject.org/. For the specific components of the Firedrake project used in this paper, see https://zenodo.org/record/5599102. For the input files of all examples and benchmarks presented,
see https://zenodo.org/record/5644392#.YYNqGL1BxR4.

## Appendix A: Governing Equations under the Anelastic Liquid Approximation

Density changes across Earth's mantle result primarily from hydrostatic compression, with density increasing by $\approx 65\%$ from surface to core-mantle-boundary (CMB) (e.g. Schubert et al., 2001). Variations in density associated with local temperature and pressure perturbations are small in comparison to the spherically averaged density. For a chemically homogeneous mantle,
it is therefore appropriate to assume a linearized equation of state, of the form:

$$
\begin{aligned}
\rho &= \bar{\rho}(\bar{T}, \bar{p}) + \rho', \\
&= \bar{\rho}(\bar{T}, \bar{p}) + \bar{\rho}(\bar{\chi}_T p' - \bar{\alpha} T').
\end{aligned}
\tag{A1}
$$

Here $\rho$, $p$, $T$, $\chi_T$ and $\alpha$ denote density, pressure, temperature, isothermal compressibility and the coefficient of thermal expansion, respectively, whilst overbars refer to a reference state and primes to departures from it:

$$T = \bar{T} + T', \qquad p = \bar{p} + p'. \tag{A2}$$

It is convenient to take the reference state as motionless and steady. Accordingly, for the purposes of the compressible case examined herein, we will assume that the reference state varies as a function of depth, $z$, only. The reference state pressure thus satisfies the hydrostatic approximation:

$$\frac{\partial \bar{p}}{\partial z} = \bar{\rho} g \cdot \hat{\boldsymbol{k}}, \tag{A3}$$

where $\mathbf{g}$ is the acceleration of gravity and $\hat{\boldsymbol{k}}$ is the unit vector in the direction opposite to gravity. On Earth, $\mathbf{g}$ is a function of position, however, for simplicity, it will be assumed constant for the compressible case examined herein. Following King et al. (2009), the reference density and reference temperature are described through an adiabatic Adams–Williamson equation of state (Birch, 1952), where:

$$\bar{\rho}(z) = \rho_0 \exp\left(\frac{\alpha_0 g_0}{\gamma_0 c_{p_0}} z\right) \tag{A4}$$

and:

$$\bar{T}(z) = T_s \exp\left(\frac{\alpha_0 g_0}{c_{p_0}} z\right). \tag{A5}$$

Here, $c_p$ and $T_s$ represent the specific heat capacity at constant pressure and surface temperature, respectively, whilst $\gamma_0$ denotes the Grüneisen parameter, given by:

$$\gamma_0 = \frac{\alpha_0}{\rho_0 c_{v_0} \chi_{T_0}}, \tag{A6}$$





where $c_v$ denotes the specific heat capacity at constant volume. Variables with a sub-script 0 are constants, used in defining the reference state. Here, they are defined at the domain's upper surface.

Assuming a linearised equation of state (Eq. A1), the dimensionless form of the conservation of mass equation under the Anelastic Liquid Approximation (ALA) can be expressed as (e.g., Schubert et al., 2001):

$$\nabla \cdot (\bar{\rho}\mathbf{u}) = 0, \tag{A7}$$

where $\mathbf{u}$ is the velocity. Neglecting inertial terms, the force balance equation becomes:

$$\nabla \cdot \left[\mu\left(\nabla\mathbf{u} + \nabla\mathbf{u}^T - \frac{2}{3}\nabla\cdot\mathbf{u}\mathbf{I}\right)\right] - \nabla p' - Ra\bar{\rho}\hat{\boldsymbol{k}}\bar{\alpha}T' - \frac{Di}{\gamma_0}\frac{c_{p_0}}{c_{v_0}}\bar{\rho}\hat{\boldsymbol{k}}\bar{\chi}_T p' = 0, \tag{A8}$$

where $\mu$ denotes the dynamic viscosity, $\mathbf{I}$ the identity tensor, $Ra$ the Rayleigh number, and $Di$ the dissipation number given by, respectively:

$$Ra = \frac{\rho_0\alpha_0\Delta T g_0 d^3}{\mu_0\kappa_0}; \quad Di = \frac{\alpha_0 g_0 d}{c_{p_0}}, \tag{A9}$$

with $\kappa$ denoting the thermal diffusivity, $d$ the length scale and $\Delta T$ the temperature scale. Note that the final term in Eq. A8 is expressed in terms of the temperature perturbation, $T'$ (sometimes called the potential temperature). Finally, in the absence of internal heating, conservation of energy is expressed as:

$$\bar{\rho}\bar{c}_p\left(\frac{\partial T'}{\partial t} + \boldsymbol{u}\cdot\nabla T'\right) - \nabla\cdot\left[\bar{k}\nabla(\bar{T} + T')\right] + Di\bar{\alpha}\bar{\rho}g\cdot\mathbf{u}T' - \frac{Di}{Ra}\Phi = 0, \tag{A10}$$

where $k$ is the thermal conductivity and $\Phi$ denotes viscous dissipation.

*Author contributions.* DRD and SCK conceived this study, with all authors having significant input on the design, development and validation of the examples and cases presented. All authors contributed towards writing the manuscript.

*Competing interests.* Authors declare that they have no conflict of interest.

*Acknowledgements.* All authors acknowledge support from the Australian Research Data Commons (ARDC: https://ardc.edu.au/, under the G-Adopt platform grant: PL031), AuScope, Geosciences Australia and the National Computational Infrastructure (NCI). DRD and SCK
acknowledge support from the Australian Research Council, under grant no. DP170100058. Numerical simulations were undertaken at the NCI National Facility in Canberra, Australia, which is supported by the Australian Commonwealth Government. Authors are grateful to the Firedrake development team, particularly David Ham, for support and advice at various points of this research.



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
