# Peer review of "Automating Finite Element Methods for Geodynamics via Firedrake"

_Geoscientific Model Development, 2021_

## Referee Comment (RC3)

**Comments on the manuscript 'Automating Finite Element Methods for Geodynamics via Firedrake' by Davies, Kramer, Ghelichkhan and Gibson**

**General Comments**

I have read the paper with great interest and find it well-written and also well-structured. The approach of the authors to take the reader from the most simple isoviscous 2D Cartesian box problem through a sequence of more and more challenging applications up to a full 3D simulation of mantle convection is nice. That these are all (mostly) standard community benchmarks makes them easily relatable. The embedded code pieces are well explained and easy to follow.

Overall I think the paper demonstrates that there is a great potential in the approach taken by Firedrake for the Geodynamics community. And it is interesting to see how that approach of automated finite element analysis has developed and been extended since the 2013 Vynnytska paper on FEniCS and mantle convection simulation.

I fully recommend publishing the manuscript once the details mentioned here and by the two other reviewers have been taken care of.

**Specific Comments**

- Listings 4, 5, 6: w.r.t. the ExtrudedMesh() function it might be helpful to the reader to mention that the default value for the layer\_height parameter is 1 / layers, which explains why e.g.  $r_{\max}$  does not factor into the call in your script.
- line 571: 'Specification of the matrix type matfree (line 16) for the combined system ensures that we do not explicitly assemble its associated sparse matrix, instead computing the matrix-vector multiplications required by the Krylov iterations as they arise.'

I am not sure that I fully understand that part. Does this imply that the full matrix (27) of the Stokes system is not assembled, but that the matvecs needed in the Schur complement approach are computed involving e.g. assembled G? Or is it a fully matrix-free implementation. Besides the need to assemble K for the AMG, that is. Especially if it means the latter case, that would be a feat to showcase more clearly and add a reference how that is handled in Firedrake.

- Listing 5, line 17: 'maximal polynomial degree of gradient of velocity' Not sure why that is 2? For a bi-quadratic Q2 element that should hold w.r.t. the gradient on the reference element, but wouldn't that change when you map to the actual element using a quadratic polynomial (iso-parametric), since the Jacobian of that would be of degree 1? Or is that unimportant for the Nitsche coefficient, or included in the fudge factor sort of?
- I have one question concerning the 3D examples with time-varying viscosity. Especially to the discussion in Section 6. As viscosity changes, is the matrix K reassembled in each time-step and does that then imply that the AMG preconditioner is re-computed? Or do you keep the initial AMG hierarchy? In the latter case can you give any idea on how large a change in viscosity needs to be before that would lead to a deterioration in the convergence?
- Section 7: As this is the sort of simulation a lot of geodynamicists would like to run, I'd appreciated more technical details here. Like what was the mesh resolution and on how many cores was it run. What was the time-to-solution? Was it the same architecture as in Section 6? I know that runtimes are hard to extrapolate from one machine to another on, but that would at least give some impression. Maybe in a future publication one could compare Firedrake and maybe Aspect? Just to demonstrate that its competitive, which I'd expect.
- line 296+297: Eqns (22) and (23) IMHO the '= 0' is incorrect and (23) should read  $b_j = -F_{\text{energy}}$ '?
- line 342: 'indefinite matrices'? Should that not read 'singular matrices', as far as I know, indefinite implies positive and negative (and potentially also zero) eigenvalues? So the additional nullspace would not affect the indefiniteness of the matrix.
- Figure 1: Just for my understanding, the different benchmark results reported in Blankenbach et al. for Case 1a/1c were very close together, only 1-2% apart, so is that why there is only one line visible? As opposed to e.g. Figure 3, where one has a visible range for reported results?

- Figure 10: Given the mentioned 'change in gradient' it would be nice to have another datapoint inbetween 1536 and 12288 cores. But that's probable not possible given the 4096 elements per core and fixed aspect ratio requirement? Is it possible to present these results in a maybe more appealing fashion?
- I concur with Cedric Thieulot in that the treatment of the continuity equation is relatively uncommon. The standard approach being to keep the derivative with the velocity and shift it over from pressure to the velocity test-function in the momentum equation, not only, but also for symmetry. In a Taylor-Hood approach the divergence of a function from the velocity space would be inside the pressure space. Can you comment on your choice? Is your approach maybe motivated by the fact that you then do not get a derivative of  $\bar{\rho}$  in the compressible formulation?

**Suggestions**

- line 213: Wouldn't 'Ritz-Galerkin' be more appropriate here?
- line 484: Maybe insert \left ... \right for scaling of parentheses?
- line 318+319: Should the derivative not be w.r.t. the expansion coefficient  $u_j$ ' (non-bold u)?  $u_j$ ' (bold u) seems to indicate the j-th component of the vector field u?
- line 701: Maybe write 'Panel (a)'?
- Personally I found the 3 pages discussion plus the conclusion a little bit lengthy as some aspects are repeatedly mentioned. Maybe that could be shortened and the extensibility of Firedrake to aspects that are not directly FEM be give more space? Like how could one combine it with e.g. a Eulerian-Lagrangian type approach for the energy equation or particle tracing for chemical species advection, .... Sec. 2 mentions 'handwritten C-kernels' and the discussion 'Firedrake's simple API', but that remains a little vague.

**Technical Corrections**

- line 192: 'constituative'  $\rightarrow$  'constitutive'
- line 269: ' $F_{\text{Stokes}}$ ' seems to be only introduced later in (24)
- line 346: 'all'  $\rightarrow$  'any'?
- line 382: 'unit vector  $(\mathbf{k})' \rightarrow$  'unit vector  $(\hat{\mathbf{k}})'$  ?
- line 423: 'and surface'  $\rightarrow$  'and surface'
- line 497: 'of of'  $\rightarrow$  'of'
- line 723: 'trilinear'  $\rightarrow$  remove
- line 915: 'the final term'  $\rightarrow$  to me that looks like the 'last but one term'?
- line 1056: King 2009, was probably available online 2009, but the volume it is in is from 2010
- line 1121: Maybe you could fix the capitalisation for Nitsche. That would be 'Über ein Variationsprinzip zur Lösung von Dirichlet-Problemen bei Verwendung von Teilräumen, die keinen Randbedingungen unterworfen sind.' p 9-15, despite its age it even has a DOI: 10.1007/BF02995904
- line 1132: 'ACT'  $\rightarrow$  'ACM' and its pages 1-27 (24 was the article number)
- line 1145: 'Discussions'  $\rightarrow$  remove
- Figure 10: 'multgird'  $\rightarrow$  'multigrid'

---

## Author Comment (AC1)

We would like to take this opportunity to thank all 7 reviewers for constructive and careful comments that allowed us to improve the paper. We respond to all reviewer comments in this document. Reviewer comments are in black, with our responses in blue.

**Reviewer 1:**

I suggest to publish this article in GMD journal after minor revisions.

**General Remarks**:
The manuscript is well written and also well constructed, it is then easy to follow the various cases tested by the authors, from simple 2D objects to more complex objects. The main objective of this manuscript is to demonstrate Firedrake's applicability for geodynamical simulation. The first part of the manuscript focuses on reproducing already approved analytical test cases and well known benchmarks. The authors must also be congratulated for the efforts they have put in describing the previous work done on the topic, which later help understanding the conducted experiments and results. The choice made by the authors to present their Python code by section, displaying each time only differences compared to the initial test case must be acknowledge, as it is very helpful in understanding the paper, and a great asset in reproducibility.

We thank the reviewer for their positive comments on the manuscript and, in particular, their explicit acknowledgment that: (i) our summary of previous worked helped to place our contribution in context of existing modelling efforts; (ii) the benefit of stepping through cases with systematically increasing complexity, with accompanying Python code.

**Specific Comments**:

1. Although the paper is well constructed and authors guideline conducting their test cases is relatively easy to understand, I believe the paper readability could be enhanced with the addition of a global table introducing the various cases (from the 2-D to a realistic application in 3-D spherical geometry), their differences and their interests.

   We agree that this would be a nice addition and have added to our revised manuscript – see Table 1.

2. Some improvement could be brought to figures, in many cases the Y axes could homogeneous to ease reader analysis of the results (cf. Figure 1, 7, 8...).

   We do not believe that this would work, given the large range of values predicted across all panels. For example, in Figure 1, all values for $Nu$ at $Ra = 10^4$ lie in the range $\sim 4.87 - 5.02$. The variability in these numbers, and their convergence towards the benchmark solution, would not be visible if they were plotted on an axis that also incorporated values of $> 23$, as at $Ra = 10^6$. With that in mind, we have decided to leave our y-axes unchanged on the figures that the reviewer refers to. Nonetheless, where possible, we have further improved figure quality.

**Reviewer 2 - Cedric Thieulot:**

**General Remarks**:
I have read with great interest the paper by Davies et al. on Firedrake in geodynamics. I found the paper well written and well structured. I also really appreciate that the authors explain how the codes work almost line by line while presenting results of standard benchmarks in the community.

We thank the reviewer for their positive comments on the manuscript and, once again, their explicit acknowledgment that explaining things carefully, almost line by line, for standard community benchmarks, was appreciated.

**Specific Comments**:

1. Section 3 is called 'Governing equations'. As such, I would expect that the generic form of the Stokes equations and the energy equation are then introduced. However equations 1 and 2 are dimensionless and those of an incompressible fluid under the Boussinesq approximation. These are of course the relevant equations and the necessary ones for the Blankenbach et al benchmark, but maybe Section 3 should start with a foreword that the paper only deals with buoyancy driven flow of a single fluid so that the relevant set of equations are Eqs 1 & 2.

   We now explicitly mention that we are dealing solely with single-material problems in this manuscript. We return to this in our discussion on future work. Our choice to introduce the non-dimensional equations, in their incompressible form, was motivated by our desire to start with the most straightforward case, and systematically add complexity. We did include the compressible equations in an Appendix, for completeness, and that remains the case.

2. Given how Firedrake works (i.e. the user must input the weak form of the equations directly into the code), I appreciate that the authors have decided to quickly re-derive the weak form of the equations because these find their way in Listing 1. However I am rather puzzled by their approach for the weak form of the continuity equation (Eq. 9). Technically this equation is correct, especially when Eq. 13 is taken into account, but no justification is given as to why div(u)=0 is integrated by parts. This is highly uncommon, so much so that I was not able to find a reference/textbook in which this is carried out. At the very least I believe the authors should justify why this uncommon approach is taken. Conversely, would the 'regular' way of not integrating the continuity equation by parts also work in Firedrake? Also, if this form of the weak form (Eq. 9) is used this begs the question as to what the spaces U,W are but these are not specified beyond "we select appropriate function spaces" at line 203.

   This is a point a number of reviewers have picked up on, and we agree that the standard formulation where the pressure gradient term is integrated by parts, and the divergence term is not, would have been a better choice. However, it should be pointed out that for all cases presented in the first submission of our paper, in which both the velocity and pressure discretisations are continuous, both formulations are identical (assuming exact integration) and this is just a matter of presentation.

   Before discretisation, the 'standard formulation' is indeed required to ensure existence and uniqueness for velocity and pressure solutions in respectively H1 and L2, and from that perspective alone it would make more sense to present it in this way. After discretisation, the choice between the two formulations is typically guided by the regularity of the chosen discrete function spaces (see, for example, Cotter et al., 2009, for an example with discontinuous velocities where our original formulation is used). For the Stokes equations, however, discontinuous pressure discretisations are far more common (Thieulot and Bangerth, 2022), so again from this perspective the standard formulation would have been a better choice.

   Given the above, in our revised submission, we have changed the formulation to follow the standard approach. We have rerun all cases and verified that for the Q2-Q1 element pair the results are, as expected, identical. In addition, based upon comments from other reviewers who were keen to see how easy it was to change the finite element pair in Firedrake, we now run a number of cases with the $Q2P_{1DG}$ element pair for which the 'standard formulation' is required. Our decision to showcase Firedrake with this additional discontinuous pressure finite element pair was motivated by Thieulot and Bangerth (2022).

3. At Line 211, I am not sure to understand what the authors mean by stating that the weak forms are 'a more general mathematically *rigorous* representation of the continuous PDEs'. In what sense?

   We dropped the words 'mathematically rigorous' here, in line with the authors comment.

4. In Section 4.2 the authors state: 'To simplify we will solve for velocity and pressure, u and p, in a separate step before solving for the new temperature $T_{n+1}$.' It might be beneficial to recall first that Eqs. 8-10 are coupled and how, and that one can solve Stokes followed by the energy equations but that this process should be iterated out.

   This is a nice suggestion. We now explicitly describe the coupling and our algorithm for solving these equations towards the beginning of Section 4.2.

5. Line 504: MUMPS is always capitalised in the MUMPS literature.

   Corrected throughout. Thanks for highlighting.

6. Line 524, line 625. For information, this setup is already implemented in ASPECT (see manual). You could use ASPECT results as a point of comparison too.

   We have added the ASPECT results as a point of comparison, as suggested.

7. Section 5.3.1: although it does not matter for the steady state, there is no mention of the *erf* function in Case 1 of Busse et al. (1994). Also, maybe already mention that advection would need a stabilisation term for high Peclet numbers (which is discussed later) but in this specific case it is not needed? Finally, this problem only contains a single pressure nullspace since Dirichlet BC are prescribed all around, so I am not following point 5 at line 580, but it probably comes from my lack of understanding what a near-nullspace is. It may be worth quickly explaining what these are (and how they differ from standard nullspaces) in Section 4.3 and why they matter so much (effectively improving the last paragraph of the section by a sentence or two). Could it also be an idea to run one model without this near-nullspaces information passed to the GAMG preconditioner and show how much it matters ? (in an appendix?)

   There are a few points here which we address in turn:

- Initial condition: we have found that the benchmark results of Busse et al. (1994) are highly sensitive to the initial condition, not only with Firedrake, but also Fluidity. This is true for many other studies and codes as well (for example, it is explicitly mentioned in Wilson et al. (2017). Adding the *erf* function to our initial condition allowed us to consistently drive the solution towards the benchmark values of Busse et al. (1994), regardless of mesh resolution. We now explicitly mention this in the text. As the reviewer notes, this is not really relevant for steady-state solutions, but hopefully our choice of initial condition will prove useful to the community when trying to reproduce these cases.

- Stabilisation: the Busse benchmark considered is not really advection dominated. Indeed the Rayleigh number is $3 \times 10^4$, which is less than the cylindrical case examined earlier in the paper (with Peclet numbers scaling accordingly). With that in mind, we found no requirement for stabilisation in this case. We see not reason to mention stabilisation until the latter (higher Ra) cases where it is required.

- Near null-spaces are already described in Section 4, starting on Line 346. Nonetheless, we now explicitly refer back to this in point 5, to help the reader. We see no motivation for comparing results with and without near-nullspaces in this study. There are a vast number of solver options available to us and a comparison between each is not the focus or goal of this manuscript.

8. Section 5.3.3: This benchmark was run with ASPECT in Liu & King, doi:10.1093/gji/ggz036.

   We have added a citation to this study in the caption. However, their results already fell within the bounds of those shown and, hence, no modifications were required to the figure.

9. General comment about $Nu$ and $V_{RMS}$ calculations throughout the paper: may be show/discuss how these are computed and how it translates into Firedrake code?

   Explicit computation of these (and other) diagnostics were provided in the scripts that accompanied the original submission (see https://zenodo.org/record/5644392#.YYNqGL1BxR4). Nonetheless, for clarity, we have now added the $Nu$ and $V_{RMS}$ calculations to Listing 1, alongside a description.

10. Fig 10: I would have expected the scale on the x-axis of these plots to be logarithmic.

    These plots have been updated to include logarithmic x-axes, as suggested.

11. Line 723: the 'trilinear' adjective alone can't be used to describe 'Q2-Q1'.

    We removed the word 'trilinear' here, noting that this point was also raised by Reviewer 3 - Marcus Mohr.

12. Eq. 40: I find that this is a bit of a missed opportunity. Given the type of modelling in Section 7, why not use the radial viscosity profile of Bull et al (EPSL 2014, http://dx.doi.org/10.1016/j.epsl.2014.06.008) who also drive the system with plate velocities and an identical temperature dependence and briefly compare/discuss your results and theirs ? (although I understand that the focus of the paper is Firedrake, not plate tectonics).

    Many studies have examined global simulations with imposed plate motion histories (including our own). The goal of this paper is not a direct comparison with these studies. We use this merely as an opportunity to showcase Firedrake's ability to run this class of model. Moreover, a direct comparison with the study of Bull et al. (2014) would not be possible. Whilst the rheological approximation of the 'realistic' simulation in our original submission may be similar, the $Ra$ is different and, most importantly, different plate motion reconstructions are imposed as surface velocity boundary conditions. As a result, the predictions from these models will not be directly comparable, given the dominant role of imposed plate velocities in organising underlying mantle flow. In light of comments by Reviewer 4 - Wolfgang Bangerth - we have also updated our final case to be compressible, with a non-linear rheology. We are not aware of any comparable case within the geodynamics community.

13. Section 7: I think it would be nice to report on how long that simulation took on how many cores.

    This information has been added to the manuscript, as requested.

14. After reading the paper I am impressed and convinced that Firedrake has a role to play in computational geodynamics in the coming years. However, the numerical experiments here, despite being useful as benchmarks and learning cases, are limited: they do not showcase features which have now become very common in state-of-the-art such as free surface, elasto-visco-plastic rhelogies, particle-in-cell or level set or compositional fields, multiphysics with operator splitting, etc ... I think the authors should address this point in the discussion by explaining whether they think that Firedrake as it is now is capable to generate a code with the above features, what would/could be the problems along the way, etc ... For instance the use of compositional fields requires to solve an advection equation

per field, which obviously Firedrake could do. But what about particle-in-cell ? Which limitations (if any) will Firedrake impose on what can and cannot be implemented?

The reviewer is correct that we have not showcased many of these features in our paper – there were already several components that we needed to cover before even getting to these! Our goal is for this paper to provide solid foundations for future work in Firedrake that we, and others in the geodynamical modelling community, can build upon. We feel that the paper achieves that goal. Nonetheless, to address the questions raised by the reviewer on specific functionalities and whether or not they can be incorporated within Firedrake: (i) free-surface – although we have not demonstrated how this is done in Firedrake, the TerraFERMA framework of Wilson et al. (2017), which is built on FEniCS, has been able to match the free-surface benchmarks of Kramer et al. (2012) – we see no reason why Firedrake would be unable to do the same; (ii) multi-material schemes – the reviewer is correct that solving an advection equation, for example with a DG scheme and appropriate limiters, would be straightforward in Firedrake. Whilst particle-in-cell schemes have been successfully developed and tested with FEniCS (Maljaars et al., 2021), we have not yet explored the use of this library (or similar) in Firedrake. Nonetheless, we see no fundamental reason that such an approach cannot be incorporated, facilitating multi-material schemes that are consistent with those used elsewhere in the community. We are also exploring the use of level-set methods, which have shown some early promise; (iii) multi-physics simulations - as mentioned in our original submission, this is one of the areas where Firedrake should really excel. By leveraging UFL, in combination with PETSc's fieldsplit preconditioning approach, future work to configure and test such coupled schemes within Firedrake should be relatively straightforward; (iv) elasto-visco-plastic rheologies – again, we see no fundamental reason that such a functionality cannot be incorporated within Firedrake. In light of the reviewer's comment, we have updated our discussion and now explicitly mention each of these aspects.

15. Finally, on the basis of the benchmarks presented I would argue that the authors have demonstrated that Firedrake is suitable for single fluid convection modelling, not for lithospheric/crustal modeling. May be the title should reflect this and the word 'geodynamics' be replaced by Mantle convection?

We have updated the title to: 'Towards Automatic Finite Element Methods for Geodynamics via Firedrake' (which implies that there is still work to do!).

16. Could the authors also add an appendix with $V_{RMS}$, $Nu$ values in tables?

Added, as requested. See Table A1 in revised submission, which includes the highest resolution results from all benchmark cases examined.

**Reviewer 3 - Marcus Mohr:**

***General remarks***:
I have read the paper with great interest and find it well-written and also well-structured. The approach of the authors to take the reader from the most simple isoviscous 2D Cartesian box problem through a sequence of more and more challenging applications up to a full 3D simulation of mantle convection is nice. That these are all (mostly) standard community benchmarks makes them easily relatable. The embedded code pieces are well explained and easy to follow. Overall I think the paper demonstrates that there is a great potential in the approach taken by Firedrake for the Geodynamics community. And it is interesting to see how that approach of automated finite element analysis has developed and been extended since the 2013 Vynnytska paper on FEniCS and mantle convection simulation.

I fully recommend publishing the manuscript once the details mentioned here and by the two other reviewers have been taken care of.

We thank the reviewer for their positive comments on the manuscript and their constructive suggestions, which we address below.

***Specific Comments***:

1. Listings 4, 5, 6: w.r.t. the ExtrudedMesh() function it might be helpful to the reader to mention that the default value for the layer height parameter is 1 / layers, which explains why e.g. rmax does not factor into the call in your script.

We have added this, as suggested, when the ExtrudedMesh functionality is first used (i.e. associated with Listing 4).

2. Line 571: 'Specification of the matrix type matfree (line 16) for the combined system ensures that we do not explicitly assemble its associated sparse matrix, instead computing the matrix-vector multiplications required by the Krylov iterations as they arise.' I am not sure that I fully understand that part. Does this imply that the full matrix (27) of the Stokes system is not assembled, but that the matvecs needed in the Schur complement approach are computed involving e.g. assembled G? Or is it a fully matrix-free implementation. Besides the need to assemble K for the AMG, that is. Especially if it means the latter case, that would be a feat to showcase more clearly and add a reference how that is handled in Firedrake.

   We have clarified that this indeed means the system is fully matrix-free (except when explicitly specified otherwise, in the case of the $K$-matrix), and how the resulting matrix-vector product is assembled directly from the symbolic expression. This matrix free capability of Firedrake is indeed a powerful feature that has large potential benefits, e.g. in combination with geometric multigrid approaches. Here, however the choice for matrix-free in the combined Stokes saddle point system is a practical one: extracting sub-matrices from the combined matrix is slow, but we do not expect to gain much from the fact that we are also matrix-free at the sub-matrix level instead of assembling into sub-matrices directly, as the solution time is dominated by the velocity sub-system which needs to be assembled anyway. We therefore chose to not further expand on the potential benefits of fully matrix-free approaches.

3. Listing 5, Line 17: 'maximal polynomial degree of gradient of velocity' Not sure why that is 2? For a bi-quadratic Q2 element that should hold w.r.t. the gradient on the reference element, but wouldn't that change when you map to the actual element using a quadratic polynomial (iso-parametric), since the Jacobian of that would be of degree 1? Or is that unimportant for the Nitsche coefficient, or included in the fudge factor sort of?

   You are correct that the polynomial degree of the gradient of Q2 element in physical space is not simply 2. As we have indicated in our response to Reviewer 4 – Wolfgang Bangerth – the correct choice of penalty parameters is a bit of a dark art. It would indeed make more sense to use a value of p=3, but it is unclear to us whether the theoretical lower-bound guarantees derived in the Hillewaert (2013) study simply extend in that way. Instead we have chosen to compensate for element curvature by incorporating it into the value of $C_{\text{ip}}$ (the 'fudge factor').

4. I have one question concerning the 3D examples with time-varying viscosity. Especially to the discussion in Section 6. As viscosity changes, is the matrix K reassembled in each time-step and does that then imply that the AMG preconditioner is re-computed? Or do you keep the initial AMG hierarchy? In the latter case can you give any idea on how large a change in viscosity needs to be before that would lead to a deterioration in the convergence?

   The matrix $K$ is re-assembled every time the value of viscosity changes. The question regarding the potential reuse of the AMG hierarchy even when the values of the matrix change is a very interesting one. In the setups in the paper we use the default behaviour which reassembles the entire hierarchy in all cases, but there is indeed potential for further speed ups by reusing it some of the time. This can be switched on using the PETSc option `pc_gamg_reuse_interpolation`. Within a Newton solve with strain-rate dependent viscosity, this would do the right thing, assuming the initial guess is sufficiently accurate. Between timesteps you would presumably need to insert some logic to reset the cached prolongators, stored on the preconditioner object which can be accessed through `petsc4py`, when the viscosity pattern has changed more than a certain tolerance. We have not yet experimented with this.

5. Section 7: As this is the sort of simulation a lot of geodynamicists would like to run, I'd appreciated more technical details here. Like what was the mesh resolution and on how many cores was it run. What was the time-to-solution? Was it the same architecture as in Section 6? I know that runtimes are hard to extrapolate from one machine to another on, but that would at least give some impression. Maybe in a future publication one could compare Firedrake and maybe Aspect? Just to demonstrate that its competitive, which I'd expect.

   This information has been added to the manuscript, as requested. A future comparison with ASPECT could potentially be valuable and of interest, although beyond the scope of the current study.

6. Line 296+297: Eqns (22) and (23) IMHO the '= 0' is incorrect and (23) should read 'bj = -Fenergy'?

   The reviewer is correct here, on both counts, and we have updated the equations accordingly.

7. Line 342: 'indefinite matrices'? Should that not read 'singular matrices', as far as I know, indefinite implies positive and negative (and potentially also zero) eigenvalues? So the additional nullspace would not affect the indefiniteness of the matrix.

   The reviewer is correct here once again, and we have updated the text accordingly.

8. Figure 1: Just for my understanding, the different benchmark results reported in Blankenbach et al. for Case 1a/1c were very close together, only 1-2% apart, so is that why there is only one line visible? As opposed to e.g. Figure 3, where one has a visible range for reported results?

   The Blankenbach study derives a 'best estimate' based upon the predictions from a suite of codes: this is why we are able to plot one value. The studies of King et al. (2009) and Tosi et al. (2015), however, do not do so and, hence, when comparing to these studies we plot the range of reported values instead.

9. Figure 10: Given the mentioned 'change in gradient' it would be nice to have another data point in between 1536 and 12288 cores. But that's probable not possible given the 4096 elements per core and fixed aspect ratio requirement? Is it possible to present these results in a maybe more appealing fashion?

   Indeed, if we are to maintain a fixed aspect ratio, which is important to ensure the problems are directly comparable at different levels, there is no way to have a data point in between 1536 and 12288 cores. We have made our x-axes logarithmic, in response to comments from Reviewers 2 and 4 – Cedric Thielout and Wolfgang Bangerth.

10. I concur with Cedric Thieulot in that the treatment of the continuity equation is relatively uncommon. The standard approach being to keep the derivative with the velocity and shift it over from pressure to the velocity test-function in the momentum equation, not only, but also for symmetry. In a Taylor-Hood approach the divergence of a function from the velocity space would be inside the pressure space. Can you comment on your choice? Is your approach maybe motivated by the fact that you then do not get a derivative of $\bar{\rho}$ in the compressible formulation?

    Please refer to our response to Reviewer 2 – Cedric Thieulot - Point 2.

**Suggestions**:

1. Line 213: Wouldn't 'Ritz-Galerkin' be more appropriate here?

   Although it is true that the specific Galerkin approach used in our discretisation follows the Ritz method, the statement in that sentence only refers to restriction to finite dimensional approximation spaces which is true of the Galerkin method in general. In that light however, the addition of 'Galerkin' to 'finite element discretisation' is indeed superfluous, so we have now removed it altogether.

2. Line 484: Maybe insert `$\left ... \right$` for scaling of parentheses?

   Corrected.

3. Line 318+319: Should the derivative not be w.r.t. the expansion coefficient 'uj' (non-bold u)? 'uj ' (bold u) seems to indicate the j-th component of the vector field u?

   That should be the $j$-th coefficient of the expansion, stored in the discrete vector $\underline{u}$, indeed. Corrected.

4. Line 701: Maybe write 'Panel (a)'?

   Corrected.

5. Personally I found the 3 pages discussion plus the conclusion a little bit lengthy as some aspects are repeatedly mentioned. Maybe that could be shortened and the extensibility of Firedrake to aspects that are not directly FEM be give more space? Like how could one combine it with e.g. a Eulerian-Lagrangian type approach for the energy equation or particle tracing for chemical species advection, . . . . Sec. 2 mentions 'handwritten C-kernels' and the discussion 'Firedrake's simple API', but that remains a little vague.

   This aligns with a comment from Cedric Thieulout. We have update the discussion accordingly, avoiding repetition where possible and discussing functionalities not incorporated in our manuscript at more length.

**Technical Corrections**:

1. Line 192: 'constituative' – 'constitutive'.

   Corrected.

2. Line 269: 'FStokes' seems to be only introduced later in (24).

   We have added a forward reference to aid the reader.

3. Line 346: 'all' – 'any' ?

   Corrected.

4. Line 382: 'unit vector (k)' – 'unit vector (kˆ)' ?

   Corrected.

5. Line 423: 'andsurface' – 'and surface'.

   Corrected.

6. Line 497: 'of of' – 'of'.

   Corrected.

7. Line 723: 'trilinear' – remove

   Removed.

8. Line 915: 'the final term' – to me that looks like the 'last but one term'?

   Corrected.

9. Line 1056: 'King 2009, was probably available online 2009, but the volume it is in is from 2010.

   Corrected.

10. Line 1121: Maybe you could fix the capitalisation for Nitsche. That would be 'Uber ein Variationsprinzip ¨ zur L¨osung von Dirichlet-Problemen bei Verwendung von Teilr¨aumen, die keinen Randbedingungen unterworfen sind.' p 9-15, despite its age it even has a DOI: 10.1007/BF02995904

    Corrected.

11. Line 1132: 'ACT' – 'ACM' and its pages 1-27 (24 was the article number).

    Corrected.

12. Line 1145: 'Discussions' – remove

    Corrected.

13. Figure 10: 'multgird' – 'multigrid'

    Corrected.

**Reviewer 4 - Wolfgang Bangerth:**

***General Remarks***:
I've read the paper about using Firedrake for mantle convection with great interest. It is clear that Firedrake is an excellent system for writing such simulators in quite a small amount of code, something I think is just really impressive! That said, there are two big issues I would like to raise (and on which I will then let the editor make a decision):

1. What is it actually that is new/what is the *message* of the paper?

2. The paper is quite long.

We thank the reviewer for taking time to provide comments on the manuscript. Many of the concerns raised are at odds with comments from other reviewers, who appreciate the aspects of the paper that this reviewer seems to dislike. That said, the reviewer makes some valid and important points, which we address in blue below.

***Specific Comments***:

1. It hasn't quite become clear to me what the *message* of the paper is or should be. Most of the paper is devoted to things that are quite standard for those who have used the finite element method, and to showing numerical results. But neither of that is new: We know how the finite element method works. And how we turn the weak form into a linear system, and one would expect that the authors have checked that their code produces correct answers – the fact that it does could have been stated in a couple of sentences of the form 'We have verified that our code produces correct answers for benchmarks X, Y, and Z, to within the errors that have previously been reported when using other codes (ref 1, ref 2, ...)'. We don't need to see many pages of tables and figures to believe this; this is particularly true because these figures do not actually compare against other codes, and so all we can infer from

them is that the code is correct, but not whether it is better or worse in any regard than existing codes. So, then, one is left with the question what actually *is* the message of the paper, seeing that the authors do not actually use the space well to make their point. I *suspect* that the authors think that the message should be 'Using Firedrake, we can write these codes in 200 lines that using other software systems takes 2000 or 20,000'. If this were the message, I think it would make a fine paper, though then I would try to remove much of the extraneous stuff mentioned above (see also my point 2 below).

I would, however, like to point out that that by itself is not as impressive as it may sound like. Being able to do what others have been able to do for a long time is not actually particularly interesting, even if the new system can do it quicker and with less code. The question that *should* have been answered is what one can do with this system that others cannot do – what does Firedrake make possible that would otherwise not have been possible (including because it would take too much work in other systems)? Unfortunately, the paper does not answer this question, and I'm not sure whether there really *is* a good answer. The fundamental issue is that systems like Firedrake are really good at *making simple things easy*; the paper illustrates this: all of the codes shown are very nice and concise, but they fundamentally all solve very simple problems that the community has been able to solve for a long time. But the other codes that are out there can *also* solve much more complicated things, with moving boundaries, much more complex material laws, adaptive meshes, adaptive time step choices, and particles. It is not a stretch to speculate that doing all of these things would not actually be much easier with Firedrake either – the majority of the code in ASPECT, for example, is after all not in the description of the finite element shape functions, or the description of linear solvers, but in material models, post-processors, time step control, dealing with how compositional fields affect this that and the other, etc. It wouldn't surprise me if that could be done with 5 times less code in Firedrake than in C++, but that would still take several 10,000 lines of code. (I will add as a side note that the model of writing many small codes for each benchmark is also not sustainable in that it encourages the fragmentation mentioned early on in the paper where everyone has their own variation of the code for a specific problem; codes like ASPECT spend many thousands of lines of code on run-time parameter handling precisely so that we can have a single code base and only need to exchange input files.)

In other words, I would appreciate if the authors could sharpen their message: Being able to do what others have already done, and illustrating this with a long list of examples is just not very interesting in itself.

There are a few points here which we address in turn:

- What is the message of the paper? As stated in our introduction, the purpose of this manuscript is to demonstrate the applicability of Firedrake for geodynamical simulation, whilst also highlighting its advantages over existing geodynamical research software. We feel that the former was clearly evident in our original submission. However, to better highlight the advantages of Firedrake over existing codes, and link these advantages back to our examples, we have rejigged the discussion slightly, to make this clear. To summarise, the key advantages of Firedrake over existing codes are:

  - Firedrake provides a complete problem solving environment, in Python: there is no longer a need to be the master of a low-level programming language like C or Fortran to develop, modify, validate and use a state-of-the-art geodynamical model. Firedrake creates a separation of concerns between employing the finite element method and implementing it - this is a game-changer, as it opens up these problems to an entire new class of user and developer.

  - Owing to the automatic code generation techniques, Firedrake applies sophisticated performance optimisations that very few developers and users would have the skill and resources to code by hand.

  - Firedrake allows a user to set up a range of problems, with differing levels of complexity, in only a tens or hundreds of lines of Python. The reviewer claims that this is not particularly impressive – we disagree, although we acknowledge that this is subjective! Building on this, the reviewer states that *Firedrake makes simple things easy* – on this we agree. The reviewer then goes on to imply that frameworks like Firedrake break down on solving more complex things – on this, we strongly disagree. In our revised submission, we have updated our 'realistic' case, to include compressibility and a non-linear rheology. This class of global model is at the very frontiers of geodynamical research (and could be further enhanced, as required). Furthermore, in our revised submission, the example that the reviewer uses to highlight the need for hand-written code – a specific class of complex rheology – has been implemented in only a few lines of Python (we return to this below). To put it simply, we have shown that the 'complex' problem explicitly highlighted by the reviewer, from Fraters et al. (2019b), as 'falling into the category of things that just require a decent amount of code' does not. We cannot demonstrate everything in this paper, and neither should we be expected to, but hopefully these changes go at least some way towards appeasing the reviewers' concerns about the applicability of Firedrake for more complex and realistic problems. At this stage, we have not shown that Firedrake can do *everything* that ASPECT, or other community codes, can do: neither was that

our goal. However, we see no fundamental reason why any of the components mentioned by the reviewer, or other reviewers, would not be straightforward to include within the Firedrake framework.

- By leveraging UFL (which allows for expressive specification of any PDE), associated strategies for automatic and efficient assembly of different finite element systems (across a vast array of finite element spaces), and PETSc for sophisticated and programmable solution of the resulting systems (with customisable preconditioners), the framework is easily extensible, allowing for straightforward application to problems involving different physical approximations, even when they require distinct solution strategies, as highlighted in our previous point and in our original submission.

- There is no need to interface with any external mesh generation packages to run any of the problems addressed in our manuscript: problems can be run in all geometries relevant to geodynamics, with ease.

- Parallelisation is seamless and problems scale well to thousands of cores.

- The reviewer claims that 'figures do not actually compare against other codes' – this is simply not the case. As noted in our response to Reviewer 2 – Marcus Mohr – in Figures 1, 3, 4, 5, 6 and 8, we are either: (i) comparing to benchmark studies where the study derives a 'best estimate' based upon the predictions from a suite of codes (Blankenbach et al., 1989; Busse et al., 1994); (ii) comparing to the range of reported values in benchmark studies where best estimates are not derived (e.g. King et al., 2010; Tosi et al., 2015); or (iii) include comparisons with the available published results from a suite of papers and models (e.g. Zhong et al., 2008; Davies et al., 2013; Bangerth et al., 2020). As well as comparing to benchmark papers, we compare carefully to analytical cases from Kramer et al. (2021), which allows us to examine the convergence properties of our system.

- The reviewer seems to suggest that these scripts will balloon if we add other components such as adaptive time-stepping and post-processing. Again - this is not the case. In our revised submission, we have added adaptive time-step control to Listing 1 (3 lines), in addition to the computation of two common diagnostics (Nu, VRMS): output in vtk format that can be visualised in paraview (or similar) was already incorporated. As is evident from the full scripts that accompanied our original submission (see https://zenodo.org/record/5644392#.YYNqGL1BxR4), output of diagnostics and vtk files for visualisation in paraview amounts to only a few additional lines.

2. Space: The paper is quite long, at 50 pages. That's because in many regards it reads like a *manual*: It is addressed to people who don't already have the finite element background to understand the basics that underlie the codes, and to people who need to be convinced with a long list of detailed examples that the codes produce the correct answer. But the audience for a manual is different from the audience of a research paper. The authors might want to think about who they are writing this paper for, and then think about every section and every figure, and re-evaluate whether that section or figure is necessary for that audience. For example, I thought that many of the figures carry no more meaning than one could equally express with the sentence "We have verified that our simulations converge to the same values as reported in X and Y (1997)". It would not greatly surprise me if the paper could be shrunk to 30 or fewer pages without actually losing any of the message.

We are surprised by this statement. Verification and validation of codes is an essential prerequisite for their successful application: it is our view that *all* users and readers *need to be convinced that these codes produce the correct answer*. Within the geodynamical community, there is a long history of papers that introduce a new approach/modelling framework, the results of which are subsequently validated against analytical and benchmark solutions (e.g. Zhong et al., 2008; Davies et al., 2011; Kronbichler et al., 2012; Wilson et al., 2017): the reviewer has himself been involved in some of these. Whilst comparisons with benchmark studies may not be the most exciting thing, they are amongst the most important aspects of introducing a new modelling framework, providing the foundations for everything that follows. We note that such comparisons are also explicitly sought by GMD through *Model Evaluation Papers*.

The novelty of Firedrake is indeed the automatic code generation, and how it allows us to examine a range of different physical approximations and geometries with ease. However, saying it can do this is one thing: demonstrating that it can do so accurately, by comparing with analytical and benchmark solutions of systematically increasing complexity, is far more powerful. Can we really introduce Firedrake to the community and claim it is accurate without presenting these tests? Would it really be scientifically acceptable just to say *we have verified that our simulations converge to the same values as reported in X and Y*, as suggested? The validity of models and approaches needs to be carefully demonstrated through evidence, rather than assertion. We note that many other reviewers comment that they appreciate the structure of the manuscript and its content, with their comments on these aspects listed below:

- Reviewer 1 – 'The choice made by the authors to present their Python code by section, displaying each time only differences compared to the initial test case must be acknowledged, as it is very helpful in understanding the paper, and a great asset in reproducibility'.

- Reviewer 2 – Cedric Thielout – 'I found the paper well written and well structured. I also really appreciate that the authors explain how the codes work almost line by line while presenting results of standard benchmarks in the community.'

- Reviewer 3 – Marcus Mohr – 'I have read the paper with great interest and find it well-written and also well-structured. The approach of the authors to take the reader from the most simple isoviscous 2D Cartesian box problem through a sequence of more and more challenging applications up to a full 3D simulation of mantle convection is nice. That these are all (mostly) standard community benchmarks makes them easily relatable. The embedded code pieces are well explained and easy to follow.'

- Reviewer 7 – Carsten Burstedde - 'This is a comprehensive and detailed paper on implementing mantle convection simulations in Firedrake. I am in favor of publishing it, due to the attention to reproducibility, the careful matching of benchmarks, and its general model and tutorial value.'

With this in mind, we chose to keep the format of the paper consistent with our original submission.

**Other Comments**

1. Line 206: This is a rather unusual formulation, with not integrating by parts the 'grad p' term, but integrating by parts the 'div u' term. The only comparable formulation I know of is what is done when comparing the primal and dual mixed formulations of the mixed Laplace equation – there it makes sense based on whether one wants the velocity or the pressure to be in a space that requires derivatives. But for the Stokes equations, the velocity *always* needs derivatives because the viscous friction term already has derivatives on it. The usual approach is to get the derivative off the pressure and let it be in L2, but the authors here gratuitously require the pressure to be in H1. This seems unnecessary and at least requires explanation.

   Please refer to our response to Reviewer 2 – Cedric Thieulot - Point 2.

2. Line 277: Choosing $C_{ip} = 100$ seems like a sledgehammer. It will for sure affect the condition number of the matrix. Is there not a smarter approach to choosing it.

   For simplicity and consistency, we have chosen the same prefactor across all cylindrical and spherical shell cases examined, although this could have been chosen lower for most cases. The trouble with the literature on this subject is that it typically makes the following assumptions: constant viscosity, uniform meshes with a poorly defined edge length $h$, and linear (straight) elements. Using the results from the Shahbazi and Hillewaert references used in the paper, we derive lower bounds for $C_{ip}$ that are also valid in non-uniform meshes, but these still do not take into account variations in viscosity and the curvature of elements. As Reviewer 3 – Marcus Mohr – points out, the polynomial degree for Q2 in an isoparameteric setting more likely needs to be set to 3 (giving a factor of $(p+1)^2 = 16$ instead of 9), although it is unclear whether the theoretical derivation from Hillewaert (2013) simply extends in that way. In addition, the chosen value of $C_{ip}$ also takes into account the rapid variation of viscosity in some of the cases examined. It is true that setting a very high value leads to a deterioration of the conditioning of the matrix, but we have not seen any issues associated with this for the value chosen here. Furthermore, convergence against the analytical solutions of Kramer et al. (2021) is consistent with theoretical expectations, for a range of different element pairs, adding confidence to the veracity of results when setting $C_{ip} = 100$.

3. Line 284: Choosing Crank-Nicolson isn't stupid, but it is also not particularly good choice. Why not use something a bit more accurate? Or, if you want to use something simple for expository reasons, at least say so. Separately, all codes use a fixed time step size; this too is not sufficient in practice.

   The reviewer is partially correct that we chose a Crank Nicolson scheme for ease of illustration. Nonetheless, the scheme is sufficiently accurate for the benchmark cases presented herein, which implies that it is a reasonable choice. Saying that, other schemes could have been used, as suggested, and one of the benefits to Firedrake is that such approaches can be tested and validated with ease, as outlined in our revised discussion. In our revised manuscript an adaptive timestep is now used.

4. Line 323: For more complex rheologies – specifically if the viscosity also depends on the pressure – the Newton matrix will have additions also in other blocks. But there are other complications one has to address for nonlinear rheologies as well; a straight up Newton scheme does not always work for strain-weakening rheologies as are common in geodynamics (see Fraters et al., GJI, 2019). In fact, this falls in the category of things that just require a decent amount of code: One has to choose a Newton matrix that is *not* just the derivative of the residual, and that has to be programmed because the auto-differentiation of the residual isn't going to produce the matrix.

   As indicated above we disagree with the conclusion that 'this falls in the category of things that just require a decent amount of code'. In fact, we are grateful for this opportunity to further demonstrate the strength of the

automated code generation approach for such cases. The construction of the Jacobian for more complex rheologies indeed bring various additional challenges such as additions in other blocks, which in traditional, hand-written code may require significant restructuring of code and data structures, as stated by the reviewer. In Firedrake this is all handled automatically, and importantly requires no additional effort when switching between various formulations one may want to investigate. The fact that the full Jacobian is derived in a fully automatic way does not imply that modifications to the Jacobian, in cases where pure Newton does not perform adequately, cannot also be implemented effortlessly. In the revised manuscript, we have included the Spiegelman et al. (2016) benchmark with pressure-dependent Drucker-Prager rheology and implemented the stabilisation advocated in (Fraters et al., 2019a). Although the Jacobian modifications are non-trivial, they are readily expressed symbolically in UFL, lifted straight from the mathematical formulae in Fraters et al. (2019a), after which Firedrake automatically derives efficient code that evaluates these at the Gauss points and adds their contributions to the integrals that form the Jacobian matrix. These Jacobian modifications were implemented in only 7 lines of Python. The same setup also highlights the ease of switching between various solution strategies, such as intertwining Picard and Newton iterations, and various other tweaks as advocated in Spiegelman et al. (2016) and Fraters et al. (2019a). As a point of interest, and to preempt further speculation about what the framework may or may not be able to do, Firedrake also provides a useful interface to implement nonlinear operations that are not readily expressed in symbolic functional expressions, e.g. rheologies that themselves require point-wise Newton solves to be evaluated, or are heavily tied into external data, where it allows the user to also provide the gradient (or a suitable approximation) of such operations (Bouziani and Ham, 2021).

5. p. 13: For this and all other listings: Align comments vertically to make the code easier to read. If you can align a few '=' signs, then that's worth doing as well.

   We follow the PEP8 conventions, which discourage 'more than one space around an assignment (or other) operator to align it with another'. This extends to the alignment of inline comments as well.

6. Line 497: 'of of' – 'of'.

   Corrected.

7. Line 683, and the following paragraph about solver scaling: This does not actually sound true. It is simply a deficiency of the preconditioner the authors choose. There are plenty of papers that show that with good preconditioners, one can achieve almost perfect weak scaling, and in particular achieve a more or less constant number of linear iterations regardless of problem size. The statement as given is a cop-out :-)

   In the paragraphs highlighted by the reviewer, we merely discuss the challenges associated with weak scaling for problems of this nature. We are not sure what exactly the reviewer means when they use the term 'cop-out'. We can only assume that they find our weak scaling results to be poor. We agree that that there are a number of schemes that improve weak scaling, and will try to explore some of these in the future. Nonetheless, our scaling results are comparable to those from other community codes, such as Burstedde et al. (2013).

8. Fig. 10: Please use a logarithmic x-axis to make these graphs easier to read.

   These plots have been updated, as also suggested by Reviwer 2.

9. Also Fig. 10: It would have been nice if there was a comparison to how long these sorts of computations take with other software systems. For example, how long would ten time steps with ASPECT take? This goes in the same direction as the 'mission' of the paper I mentioned above. Just knowing *that* your code can solve a problem others have been able to solve for a long time is not so interesting. If you can do it *as fast as others* with 1/100th the lines of code they need, then that *is* interesting.

   We have no experience with running ASPECT and a direct comparison is beyond the scope of this paper. However, we would be amenable to such a comparison in the future, if the reviewer were happy to work with us on setting up directly comparable problems.

10. Section 7 (Line 720): It doesn't sound quite right to call this section 'realistic applications'. The application uses an incompressible mantle, a linear temperature and depth dependence for the viscosity, and has no special provisions to deal with the crust any different than the rest of the mantle. This is neither realistic, nor new. 'Realistic' models have been far more complex than this for a very long time already.

   What the reviewer is essentially saying here is that 'realistic' is a subjective term. In our eyes, the class of model shown is realistic, in that these are the type of global mantle dynamics model that we use and are familiar with, in addition to many others in the community (an exemplar citation list is provided in the paper – this is also acknowledged by reviewer Mohr). However, it is true that the model presented in our original submission was

incompressible with a simple depth- and temperature-dependent rheology. In light of this comment, we have run a new compressible (ALA) simulation, with a non-linear rheology, whilst keeping imposed plate motion histories as a boundary condition. To our knowledge, no mantle circulation models (i.e. mantle convection models with imposed plate motion histories) have been run with this level of complexity. The fundamental message of this section of the paper remains the same however: Firedrake is not just a tool for toy problems, it can be used to examine questions from the frontiers of geodynamical research. In light of the reviewer's comment we have dropped the word 'realistic' from the section title.

11. Line 733: Are T=0 and T=1 mixed up here?

   Indeed they were. Good spot!

**Reviewer 5:**

This is a well-written manuscript. I would recommend it to be published with some minor revisions.

***Main Comments***:

1. I agree with RC4 by Wolfang Bangerth that the main message of the manuscript should be clarified. If the target audience is geodynamicists who are testing different approximations of governing equations or the performance of different solver algorithms, this manuscript is an excellent introduction and demonstration and can be published with some minor revisions in the introduction and conclusion. If the target audience is geodynamicists who are working on real-world geophysical problems, this manuscript falls short in several aspects. It is probably beyond the scope of this manuscript to address these aspects, but I would list them below nonetheless.

   We have clarified the message of the paper in response to comments by Reviewer 4 – Bangerth. We feel that the manuscript is of interest, and of benefit, to both classes of reader described this reviewer (cf. comments from other reveiwers). The range of problems presented, culminating in an (updated) global spherical case, demonstrates that Firedrake is applicable to the breadth of problems listed by the reviewer.

2. Since solving the governing equation is only a small part of a mantle convection code, some part of the generated C kernel must be extended. I would like to know how easy is it to modify and extend the generated C kernel? Can it be done in Python or must be done in C? For example, to add dynamic time stepping, one has to calculate $\min(v/h)$ over all elements, where h is the element size. Can the contents of v and h be accessed in Python, or only in C?

   Unfortunately, we do not entirely follow the first part of the reviewer's comment. A C kernel in this context is the local assembly code that deals with the assembly of local vectors and matrices associated with the degrees of freedom of a single element and its neighbours. Such kernels are scheduled to be executed inside a loop over (subsets) of the elements in the mesh with the accompanying copying between local and global structures, as usual in all finite element code. This additional code again is automatically generated, but the high-level scheduling in PyOP2 (e.g. to start this code to loop over the halo region), is all driven from Python. Similarly firedrake itself is completely written in Python. At runtime it is this Python code that ensures the necessary bits of C code are generated just in time, and executes these by passing it the necessary vectors and sparse matrices, which are also fully accessible from Python as numpy arrays and PETSc matrices (through petsc4py). There is, as such, not a single monolithic piece of generated C code that *'needs to be modified and extended'*. All diagnostics and any additional functionality required to obtain the results in this paper are included in the full Python scripts in the repository that accompanies the paper (see https://zenodo.org/record/5644392#.YYNqGL1BxR4 for repository accompanying our original submission), in all cases requiring only a few lines of extra Python code. To give the reader a more complete picture of how this is done, we have added computation of Nusselt Number and RMS velocity diagnostics, in addition to adaptive timestepping, to Listing 1. The diagnostics amount to 1 line of Python code each, with adaptive timestepping requiring 3 lines.

   More generally, if additional diagnostics need to be computed, or other pre or post-processing steps performed, this can be done in Python, for example by accessing the coefficients of the solution fields directly as numpy arrays. We should however keep in mind that treating finite element solutions as just a bunch of values defined in a number of locations, goes against the core finite element paradigm that solutions are really functions defined everywhere in the domain. It is therefore often better to define your diagnostics in that same paradigm, involving finite element integrals and appropriate interpolations and projections, where Firedrake offers the full flexibility to express and efficiently perform any such operations as required. Again the CFL-based timestep calculation is a good example of why thinking in terms of individual velocity values and grid distances $h$ as distances between nodes may lead to incorrect results in non-uniform and bendy meshes, where it is better to think of the CFL condition in terms of a transformation of the physical velocity back to a velocity in reference coordinates. For operations that are not

easily expressed in a finite element manner (e.g. in our final example we need to interpolate the forcing from a GPlates-defined structured grid onto our solution grid), efficient implementation is often available through numpy operations. For time critical operations directly on the solution grid using the local stencil, Firedrake also provides the functionality to provide your own local C kernel, expressed as a Python string, to be scheduled for execution in a loop over the mesh entities, however we have so far not needed this functionality in any of our setups.

3. Most legacy mantle convection codes are using domain decomposition to parallelize. But Firedrake seems to parallelize by distributing the matrices and vectors. How will this affect the extension of the code, for example, adding markers?

   Firedrake has an efficient local parallelisation through PyOP2, but there is an additional layer of domain decomposition through PETSc's DMPlex layer. This means that the loop-generating code doesn't need to know about the exact structure of the problem, but this structure is maintained at a higher level to achieve communication-aware computational efficiency. We see no reason why this would complicate parallelisation of markers/particles, for example, and note that it has already been achieved in FEniCS, which uses comparable parallelisation approaches (Maljaars et al., 2021).

***Minor Suggestions***:

1. In the code listings, it will be better if some function arguments are passed as keywords, instead of just as numbers. For example, line 8 in listing 1 becomes: W = FunctionSpace(mesh, family='CG', degree=1).

   Good suggestion, the keyword args have been added to FunctionSpace and DirichletBC in Listing 1.

2. Line 40-41 listing 1, the boundary tags 1, 2, 3 ,4 are better to named as left_id, right_id, bottom_id, top_id, respectively. Related, Listing 2, bottom_id and top_id are used but not defined.

   Updated.

3. The solver parameters for energy solver in Listing 1 and as described in Line 405 appear to be inconsistent with those described in Line 300. Also, the solver parameters for Stokes solver in Line 336 are inconsistent with those in later code listings.

   The reviewer is partially correct here. On line 300 of our original submission, we stated that 'For the 2-D cases presented in this paper, this asymmetric linear system is solved with a direct solver' - which is the exact setup used is listing 1. For the Stokes system, the options specified on line 335 are also consistent with the listing. However we failed to mention the use of the FGMRES scheme for the outer solve, on line 335, which is potentially what was confusing the reviewer. We have now clarified this point.

4. In the code, the buoyancy term is using Ttheta, rather than Told or Tnew. Why?

   This is from the Crank-Nicolson scheme, and defined in Equation 20. On line 19 of Listing 1, Ttheta is weighted between Told and Tnew.

**Reviewer 6:**

I agree with the comments RC4 (by Wolfgang Bangerth) and RC5 that the overall purpose of the paper needs to be clarified. In general the authors should try to answer the following question: What can we do with this code now that was impossible with other approaches before? I'll try to provide some suggestions for the authors in order to answer this questions and improve the manuscript to a point where I can recommend it for publication.

***Major Suggestions***:

1. The authors actually give some suggestions themselves in Section 8 of the paper which have not been followed up but would have made the manuscript a clear candidate for publication. Furthermore, I would suggest to restructure the paper such that we have a section presenting the benchmark implementations using UFL and a part discussing improvements over other existing packages and approaches.

   - Benchmark cases implemented using UFL: It would be a valuable contribution to provide a ready-to-use collection of benchmark cases (as presented in the paper) described in UFL to be used with multiple packages that can make use of UFL. An example here is dolfin_dg, which provides UFL forms for discontinuous Galerkin discretizations for compressible flow and can be used by FEniCS, Firedrake, or even DUNE-FEM. Right now

it is not entirely clear whether the presented examples could be used with FEniCS or TerraFERMA. It would also be in line with the much advertised separation of concerns.

Providing the set of cases present is an essential aspect of reproducibility and we did so in our original submission (see https://zenodo.org/record/5644392#.YYNqGL1BxR4 for repository accompanying our original submission). The UFL for each case is a part of this. With that in mind, the reviewer's first point was already satisfied. Could the presented examples be used in FEniCS or TerraFERMA? The answer is almost definitely not. TerraFERMA, for example, has a required syntax for its UFL so copying and pasting would be impossible. FEniCS also has a different API. However, because the code presented is so readable (high-level code, rather than 1000s of lines of Fortran/C) then it is straightforward to translate across to different frameworks.

- Improvements over other existing approaches: In the second part of the paper I suggest to better highlight the strength of using the benchmark cases with Firedrake over other available packages. For example:
  - Could we easily define our own preconditioner, for example, making use of UFL again? And if so, an example should be added.

    This was already mentioned in the discussion (lines 833-841 of the original submission). We have now also added an example code, as suggested, to show how a UFL based preconditioner can be simply defined by the user, reimplementing the MassInvPC preconditioner already used in the paper.
  - The presented finite element discretization (Q2-Q1 on Cartesian grids) is standard. How easy and feasible is it to use other (maybe more appropriate) discretiztions or other grid element types?

    It is straightforward to use a different element pair in Firedrake, as indicated in our original submission. A recent study highlighted the Q2-Q1 element pair as one the most robust and reliable choices for geo-dynamical simulations (Thieulot and Bangerth, 2022), so we find the reviewer's comment 'maybe more appropriate' difficult to understand. Nonetheless, to illustrate the ease at which the element pair can be changed, we also run one of our basic benchmark cases and our analytical comparisons against cylindrical shell cases from Kramer et al. (2021) with a $Q2P_{1DG}$ finite element pair (i.e. incorporating a discontinuous pressure). As demonstrated in our revised submission, the changes required are minimal. We have updated our discussion in light of these changes.
  - Could we easily do a space-time discretization since UFL should make it actually very easy to write this down?

    There are a large range of approaches that can be classified as space-time discretisations. A subset in which space-time is still discretised as a tensor product between a spatial mesh and a 1D time mesh, can be realized indeed in a relatively straightforward manner in Firedrake. A current limitation is that UFL currently only allows for one, two or three dimensional domains, limiting this approach to two spatial dimensions. Local DG timestepping (local subcycling) approaches could be implemented in much the same way as in other codes but require a fair amount of bookkeeping. More generic, adaptive, frontal space-time discretisation and solver techniques would of course require a substantial amount of code development in the Firedrake framework. Although there are many interesting avenues for research in these areas, we do not immediately see mantle convection modelling to be a prime candidate for application of these techniques.
  - As stated by the authors, 'the automated approach underpinning Firedrake has the potential to revolutionize the use of adjoints and other inverse schemes in geodynamics'. This would have been a clear major improvement over the existing state of the art. Could this at least be demonstrated for a simplified example? It would clearly help the community to have a starting point in this direction.
    Including adjoint results is beyond the scope of this paper. We simply cannot do everything and some reviewers already comment that the manuscript is long. This will be the topic of future work.
  - Scalability: Would it be possible to also see at least one strong scaling result for one of the presented cases?
    Strong scaling applies to CPU-bound problems, and is about finding a good balance of parallel efficiency to walltime. Efficient weak scaling enables larger problem sizes, allowing for novel investigation of large-domain or high-resolution processes. The latter is the current focus of geodynamical simulation and we see no clear motivation for including strong-scaling analyses.
- Limitations: What are the limitations of the presented approach? Existing approaches already cover a wide range of features. For example, could this code be used to replicate the results presented in 'Large-scale adaptive mantle convection simulation' by Burstedde et al. (2013) at the same grid resolution?

The excellent paper by Burstedde et al. (2013) includes many results and it is difficult to know which set of results the reviewer is referring to. We already include comparisons with analytical solutions from Kramer et al. (2021) (which are a more comprehensive test than those analysed in Figures 7 and 8 of Burstedde et al. (2013)). We also include 3-D spherical models that are directly equivalent to Figures 9 and 10 of Burstedde et al. (2013) and parallel scaling analyses similar to those presented in Table 5 of Burstedde et al. (2013). With that in mind, many of our results are already directly comparable to those of Burstedde et al. (2013).

- Besides that, the length of the paper in it's current form is ok, since for this journal the resulting published paper will be much shorter due to the two-column format used. I'm just wondering how the code representation would look like in that format. And then, addressing my comments from above will probably mean to shorten the current presentation of the test cases.

  We see no motivation to shorten the presentation of test cases: they systematically increase in complexity and other reviewers have commented that they enjoy the presentation. How will these look in two-column style? That is something we will work on with the typesetters, who will have experience with this.

**Minor Remarks:**

1. Throughout the paper: Crank-Nicholson should be Crank-Nicolson.

   Corrected throughout. Thanks for pointing this out.

2. Line 418: The citation of Homolya et al. is in my opinion wrong here, because the cited paper in the end only shows compilation times of the TSFC but does not make any statement about the efficiency of the produced code.

   Homolya et al. (2018) developed TSFC. We feel that the citation is fair as is.

**Reviewer 7: Carsten Burstedde**

This is a comprehensive and detailed paper on implementing mantle convection simulations in Firedrake. I am in favor of publishing it, due to the attention to reproducibility, the careful matching of benchmarks, and its general model and tutorial value. I have only minor suggestions.

**Minor Suggestions:**

1. The Newton terms are omitted, the Nitsche terms are omitted from the discussion of system solution. Yet these would be rather interesting since these take some effort to derive and many will not have taken the trouble to do so before, especially in light of concrete benchmarks. Conversely, what is presented in terms of equations in the paper is rather well known. Thus my question, would it be practical to add these equations/derivations? Or is it rather that due to the automatic differentiation available in Python, there is no need to present them in detail?

   The reviewer is correct that, in our original submission, we did not explicitly write out the additional Newton terms that arise in nonlinear cases, and that we did not perhaps sufficiently highlight the effort these may take to derive in traditional codes. We have expanded our discussion of additional terms required for nonlinear rheologies but without explicitly working out the $\partial \eta / \partial u$ term. This, of course, depends on the specific rheological law that is being applied, and it is indeed the strength of the approach advocated in our paper that it allows us to easily switch between various formulations with Firedrake automatically deriving their derivatives symbolically and generating efficient code that implements them. This point is now highlighted more clearly in the discussion section and we have referred readers to the appendix of Fraters et al. (2019a) to get an impression of what the viscosity derivative terms may look like for a number of commonly used rheologies.

2. Figure 6b is the first that disagrees with established values. Is it understood why? Maybe the results obtained here are actually better?

   In response to comments from Reviewer 2 – Cedric Thieulot – ASPECT results have been added to Figure 6b. The predictions from Firedrake sit in between those of Fluidity and ASPECT, albeit closer to predictions from Fluidity. A broader community benchmark effort (which has been ongoing for some time) is needed to better understand the reasons behind these differences – they could, for example, simply be because the ASPECT results included in their manual at a coarser resolution than the cases examined here.

3. Chapter 7, realistic convection simulation: what is the spatial and temporal resolution?

   This information has been added in our revised submission, alongside the CPU time required for solution, as requested by other reviewers.

4. Line 220, it is also required that finite element of different spaces pairings are stable, which depends crucially on the choice of these spaces. The paper omits this point entirely, maybe some comments can be added. Related, discontinuous pressure discretizations may be worth attention.

   In our original submission, we only mentioned that this choice should be suitable without being any more specific. We have now added the following lines: 'It should be noted however that in practice this choice is guided by numerical stability considerations in relation to the specific equations that are being solved. In particular, the choice

of velocity and pressure function spaces used in the Stokes system is restricted by the LBB condition (see Thieulot and Bangerth, 2022, for an overview of common choices for geodynamical flow)'. In addition to the Q2Q1 finite element pair we have now also included a number of cases with the $Q2P_{1DG}$ finite element pair with discontinuous pressure.

5. Line 255: The (imposed) Tinhom values produce a term linear in q that moves to the right hand side of (10). How is this handled?

Effectively, pretty much as you describe: that term moves to the right-hand side (i.e. it is lifted). In terms of implementation details this happens automatically because Firedrake always solves the equations in defect form, even in the linear case, which means that after we have applied the inhomogenous boundary condition to the initial guess, the right hand side is formed by the initial residual which includes the contribution you mention. In the Jacobian matrix the degrees of freedom that would normally allow deviation from the imposed inhomogenous value are completely decoupled from the equation by zero-ing out the corresponding row and column entries

6. Line 290, 298: It is not clearly described how the temperature equation could possibly become nonlinear in T. Maybe the authors are referring to the fully coupled systems, where the parameter u depends itself on T?

We have now mentioned potential non-linearities in the energy equation, for example, through a temperature dependent thermal diffusivity.

7. Line 550: Using PETSc fieldsplit is more advanced and even more interesting than what is lined out earlier in the paper. It would be valuable to discuss the corresponding numerical approach and technique in more detail, which may help readers to move in this direction themselves.

PETSc fieldsplit is indeed very powerful. We have given a short description of potential use cases in geodynamics. We also discuss these in Section 8 of the manuscript.

8. Line 836, may mention preconditioning work by Bangerth used in the Aspect code.

Citation added to Kronbichler et al. (2012) where ASPECT's preconditioners are discussed.

**References**

Bangerth, W., Dannberg, J., Gassmoeller, R., and Heister, T.: ASPECT v2.2.0, https://doi.org/10.5281/zenodo.3924604, 2020.

Blankenbach, B., Busse, F., Christensen, U., Cserepes, L., Gunkel, D., Hansen, U., Harder, H., Jarvis, G., Koch, M., Marquart, G., Moore, D., Olson, P., Schmeling, H., and Schnaubelt, T.: A benchmark comparison for mantle convection codes, Geophys. J. Int., 98, 23–38, https://doi.org/10.1111/j.1365-246X.1989.tb05511.x, 1989.

Bouziani, N. and Ham, D. A.: Escaping the abstraction: a foreign function interface for the Unified Form Language [UFL], arXiv preprint arXiv:2111.00945, 2021.

Burstedde, C., Stadler, G., Alisic, L., Wilcox, L. C., Tan, E., Gurnis, M., and Ghattas, O.: Large-scale adaptive mantle convection simulation, Geophys. J. Int., 192, 889–906, https://doi.org/10.1093/gji/ggs070, 2013.

Busse, F. H., Christensen, U., Clever, R., Cserepes, L., Gable, C., Giannandrea, E., Guillou, L., Houseman, G., Nataf, H. C., Ogawa, M., Parmentier, M., Sotin, C., and Travis, B.: 3D convection at infinite Prandtl number in Cartesian geometry - a benchmark comparison, Geophys. Astrophys. Fluid Dyn., 75, 39–59, https://doi.org/10.1080/03091929408203646, 1994.

Cotter, C. J., Ham, D. A., Pain, C. C., and Reich, S.: LBB stability of a mixed Galerkin finite element pair for fluid flow simulations, Journal of Computational Physics, 228, 336–348, 2009.

Davies, D. R., Wilson, C. R., and Kramer, S. C.: Fluidity: a fully unstructured anisotropic adaptive mesh computational modeling framework for geodynamics, Geochem. Geophys. Geosys., 120, Q06 001, https://doi.org/10.1029/2011GC003551, 2011.

Davies, D. R., Davies, J. H., Bollada, P. C., Hassan, O., Morgan, K., and Nithiarasu, P.: A hierarchical mesh refinement technique for global 3D spherical mantle convection modelling, Geosci. Mod. Dev., 6, 1095–1107, https://doi.org/10.5194/gmd-6-1095-2013, 2013.

Fraters, M. R., Bangerth, W., Thieulot, C., Glerum, A., and Spakman, W.: Efficient and practical Newton solvers for non-linear Stokes systems in geodynamic problems, Geophysical Journal International, 218, 873–894, 2019a.

Fraters, M. R., Bangerth, W., Thieulot, C., Glerum, A., and Spakman, W.: Efficient and practical Newton solvers for non-linear Stokes systems in geodynamic problems, Geophysical Journal International, 218, 873–894, 2019b.

Hillewaert, K.: Development of the discontinuous Galerkin method for high-resolution, large scale CFD and acoustics in industrial geometries, PhD Thesis, Université de Louvain, 2013.

Homolya, M., Mitchell, L., Luporini, F., and Ham., D.: Tsfc: a structure-preserving form compiler, SIAM J. Sci. Comput., 40, 401–428, https://doi.org/10.1137/17M1130642, 2018.

King, S. D., Lee, C., van Keken, P. E., Leng, W., Zhong, S., Tan, E., Tosi, N., and Kameyama, M. C.: A community benchmark for 2-D Cartesian compressible convection in Earth's mantle, Geophys. J. Int., 179, 1–11, 2010.

Kramer, S. C., Wilson, C. R., and Davies, D. R.: An implicit free-surface algorithm for geodynamical simulations, Phys. Earth Planet. Int., 194, 25–37, https://doi.org/10.1016/j.pepi.2012.01.001, 2012.

Kramer, S. C., Davies, D. R., and Wilson, C. R.: Analytical solutions for mantle flow in cylindrical and spherical shells, Geosci. Model Dev., 14, 1899–1919, https://doi.org/10.5194/gmd-14-1899-2021, 2021.

Kronbichler, M., Heister, T., and Bangerth, W.: High accuracy mantle convection simulation through modern numerical methods, Geophys. J. Int., 191, 12–29, https://doi.org/10.1111/j.1365-246X.2012.05609.x, 2012.

Maljaars, J. M., Richardson, C. N., and Sime, N.: LEoPart: A particle library for FEniCS, Comp. Math. App., 81, 289–315, 2021.

Spiegelman, M., May, D. A., and Wilson, C. R.: On the solvability of incompressible Stokes with viscoplastic rheologies in geodynamics, Geochemistry, Geophysics, Geosystems, 17, 2213–2238, https://doi.org/10.1002/2015GC006228, 2016.

Thieulot, C. and Bangerth, W.: On the choice of finite element for applications in geodynamics, Solid Earth, 13, 229–249, 2022.

Tosi, N., Stein, C., Noack, L., Hüttig, C., Maierová, P., Samuel, H., Davies, D. R., Wilson, C. R., Kramer, S. C., Thieulot, C., Glerum, A., Fraters, M., Spakman, W., Rozel, A., and Tackley, P. J.: A community benchmark for viscoplastic thermal convection in a 2-D square box, Geochemistry, Geophysics, Geosystems, 16, 2175–2196, https://doi.org/10.1002/2015GC005807, 2015.

Wilson, C. R., Spiegelman, M., and van Keken, P. E.: TerraFERMA: The Transparent Finite Element Rapid Model Assembler for multiphysics problems in Earth sciences, Geochemistry, Geophysics, Geosystems, 18, 769–810, https://doi.org/10.1002/2016GC006702, 2017.

Zhong, S., McNamara, A., Tan, E., Moresi, L., and Gurnis, M.: A benchmark study on mantle convection in a 3-D spherical shell using CitcomS, Geochem. Geophys. Geosys., 9, Q10017, https://doi.org/10.1029/2008GC002048, 2008.